# How Much Can Transfer? BRIDGE: Bounded Multi-Domain Graph Foundation Model with Generalization Guarantees

Haonan Yuan[1]  Qingyun Sun[1]  Junhua Shi[1]  Xingcheng Fu[2]  Bryan Hooi[3]  Jianxin Li[1]  Philip S. Yu[4]

## Abstract

Graph Foundation Models hold significant potential for advancing multi-domain graph learning, yet their full capabilities remain largely untapped. Existing works show promising task performance with the "pretrain-then-prompt" paradigm, which lacks theoretical foundations to understand why it works and how much knowledge can be transferred from source domains to the target. In this paper, we introduce **BRIDGE**, a **B**ounded g**R**aph foundat**I**on model pre-trained on multi-**D**omains with **G**eneralization guarant**E**es. To learn discriminative source knowledge, we align multi-domain graph features with domain-invariant aligners during pre-training. Then, a lightweight Mixture of Experts (MoE) network is proposed to facilitate downstream prompting through self-supervised selective knowledge assembly and transfer. Further, to determine the maximum amount of transferable knowledge, we derive an optimizable generalization error upper bound from a graph spectral perspective given the Lipschitz continuity. Extensive experiments demonstrate the superiority of BRIDGE on both node and graph classification compared with 15 state-of-the-art baselines.

## 1. Introduction

Graph-structured data is pervasive across a wide range of domains, including social network analysis (Fan et al., 2019), recommendation systems (Wu et al., 2019), drug discovery (Lin et al., 2021), and bioinformatics (Gligorijević et al., 2021), *etc.* Graphs are capable of representing complex relationships and attributed knowledge between entities, making

them a powerful tool for solving real-world problems. As large models like GPTs and DeepSeek have shown remarkable success in handling multi-modal Euclidean data (Chang et al., 2024; Guo et al., 2025a), they struggle with graphs. The inherent non-Euclidean structures, combined with heterogeneity and task diversity, makes it hard for large models to process effectively (Mao et al., 2024a; Jin et al., 2024).

Within this context, Graph Foundation Models (GFMs) have garnered significant attention in very recent studies (Mao et al., 2024b; Shi et al., 2024a; Zi et al., 2024). A GFM is typically defined as a graph learning model pre-trained on extensive multi-domain graph data and adapted for diverse downstream graph tasks (Bommasani et al., 2021). Such a "pretrain-then-finetune" paradigm has validated its feasibility of constructing GFMs (Tang et al., 2024a; He & Hooi, 2024; Lachi et al., 2024). Beyond that, graph prompt learning is a promising technique to speed up test-time fine-tuning, demonstrating the updated "pretrain-then-prompt" paradigm enabling generalization capabilities to any target domains and diverse tasks, showing a potential path to GFMs.

However, despite the initial success, "pretrain-then-prompt" frameworks still face notable limitations: **(1) Over-reliance on text-attributed graphs.** Several methods rely heavily on aligning features with textual descriptions, leveraging language models to interpret domain-specific semantics (Tang et al., 2024a; Li et al., 2024a; Wang et al., 2024; Tan et al., 2024). Yet, many open graph datasets lack explicit text attributes, making such approaches impractical for extensive real-world applications. Furthermore, obtaining textual annotations for non-text graphs can be costly, time-consuming, and prone to domain biases induced by language model hallucinations (Ribeiro et al., 2021), ultimately hindering generalization. **(2) Insufficient theoretical analysis.** While graph prompts have empirically demonstrated strong performance in transferring knowledge across graph domains (Sun et al., 2023; Yu et al., 2024b), theoretical insights remain sparse. Beyond discussing learnable parameters and fine-tuning methods, there is limited theoretical understanding of the mechanisms behind effective prompt-based transfer, especially in few-shot scenarios. Consequently, foundational theoretical bounds guiding graph prompt generalization remain unclear.

---

[1]SKLCCSE, School of Computer Science and Engineering, Beihang University [2]Key Lab of Education Blockchain and Intelligent Technology, Ministry of Education, Guangxi Normal University [3]School of Computing, National University of Singapore [4]Department of Computer Science, University of Illinois, Chicago. Correspondence to: Qingyun Sun <sunqy@buaa.edu.cn>.

*Proceedings of the $42^{nd}$ International Conference on Machine Learning*, Vancouver, Canada. PMLR 267, 2025. Copyright 2025 by the author(s).

The limitations are non-trivial due to **three key challenges,**

**(1) Pre-training: How to align multi-domain features?** Graph features across different domains show varying distributions, often characterized by both distinct dimensions and semantics. Traditional feature alignment techniques like SVD (Stewart, 1993; Yu et al., 2024c) or non-linear mapping with MLPs (Yin et al., 2022; Hou et al., 2024), often fail to capture domain-specific knowledge. The challenge lies in how to simultaneously and effectively align the heterogeneous features at both dimension- and semantics-level while preserving shared and domain-invariant knowledge.

**(2) Down-prompting: How to initialize prompts for efficient adaptation?** Graph prompt learning aims for efficient adaptation with limited target domain data, which implicitly requires fast fine-tuning on restricted learnable parameters within limited epochs instead of re-training. Existing methods suffer from inefficient random initialization of graph prompts, which results in slow convergence in fine-tuning, particularly in few-shot settings where the available labeled data is scarce. The challenge lies in how to initialize graph prompts in a self-supervised way enabling source knowledge to actively and adaptively support target adaptation.

**(3) Theoretical analysis: How much knowledge can be transferred?** The ultimate goal of the GFM is to serve as graph intelligence, taking in massive graph knowledge and enabling zero-shot transfer to any target domain. Despite the success of the aforementioned "pretrain-then-prompt" paradigm, the challenge lies in the lack of understanding regarding the lower bound of transferable knowledge, *i.e.*, a thorough theoretical analysis of the generalization error upper bound is crucial to quantify the limits of knowledge transfer, which can also play as regularizers constraining both the pre-training and prompting optimization.

To address these challenges above, we introduce **BRIDGE**, a novel **B**ounded g**R**aph foundat**I**on model pre-trained on multi-**D**omains with **G**eneralization guarant**E**es. To capture discriminative source domain knowledge, we align multi-domain graph features with a set of domain-invariant aligners. We then introduce a lightweight MoE network to enhance prompting by enabling self-supervised and selective knowledge transfer. To quantify the maximum transferable knowledge, we derive an optimizable upper bound on the generalization error, grounded in graph spectral theories, leveraging the Lipschitz continuity. **Our contributions are,**

- We propose a novel multi-domain graph foundation model named BRIDGE with a set of domain-invariant feature aligners and efficient graph prompts initializers.

- We analyze generalization error for prompt fine-tuning, and derive an optimizable upper bound for the first time.

- Extensive experiments have demonstrated the superiority of BRIDGE on downstream node and graph classification compared with 15 state-of-the-art baselines.

## 2. Related Work

### 2.1. Multi-Domain Graph Pre-training (Appendix H.1)

Pre-training on graphs (Liu et al., 2022; Hu et al., 2020a; Xie et al., 2022) leverages self-supervised learning to capture intrinsic semantics, but often assumes high similarity between domains, limiting generalization. Cross-domain learning (Zhao et al., 2024; Liu et al., 2024b) aims to transfer knowledge between domains but is often domain-specific and task-tailored. Multi-domain methods (Liu et al., 2024b; Tang et al., 2024b) seek to align domain features, yet specific graph types constrain many or fail to address conflicts.

### 2.2. Graph Fine-tuning and Prompting (Appendix H.2)

Fine-tuning pre-trained graph models for downstream tasks is computationally expensive (Zhang et al., 2022; Sun et al., 2024; Zhili et al., 2024). Parameter-efficient fine-tuning (PEFT) (Zhu et al., 2024a; Tian et al., 2024) mitigates these issues but struggles in complex multi-domain scenarios due to the domain biases and task heterogeneity. Prompt learning (Liu et al., 2023a; Zhou et al., 2022b) offers a more efficient way of guiding models with minimal updates but is limited to single-domain tasks and lacks scalability.

### 2.3. Towards Graph Foundation Models (Appendix H.3)

Graph Foundation Models (GFMs) extend language and vision foundation models to graphs, where pre-training on multi-domain data and transfer to downstream by parameter-efficient fine-tuning and prompt learning provides a promising way. A growing line of inquiry is into the existence of scaling laws in GFMs (Ma et al., 2024; Tang et al., 2024c; Chen et al., 2024a; Liu et al., 2024c), which hypothesize increasing parameter scale may yield emergence, such as enhanced transferability and generalization across tasks. However, its theoretical foundations remain under-developed.

## 3. Preliminary

**Notation.** A graph is denoted as $G = (\mathcal{V}, \mathcal{E})$, where $\mathcal{V}$ is the node set and $\mathcal{E}$ is the edge set. Let $\mathbf{A} \in \{0, 1\}^{N \times N}$ be the adjacency matrix and $\mathbf{X} \in \mathbb{R}^{N \times d}$ be the node features, where $N = |\mathcal{V}|$ is node number and $d$ for input dimension.

**Pre-training and Few-shot Fine-tuning.** Given $n$ source graphs $\{G^{\mathcal{S}}\} \in \mathcal{G}^{\mathcal{S}}$ from multi-domains $\{D^{\mathcal{S}}\} \in \mathcal{D}^{\mathcal{S}}$ with their labels $\{Y^{\mathcal{S}}\} \in \mathcal{Y}^{\mathcal{S}}$, the pre-training goal is to train a graph learner $h = g(f_{\Theta}(\cdot))$ on multi-domain graph datasets, after which the pre-training parameter $\Theta^{\star}$ is frozen. During down-prompting, given a set of target graphs $\{G^{\mathcal{T}}\} \in \mathcal{G}^{\mathcal{T}}$ from target domain $\{D^{\mathcal{T}}\} \in \mathcal{D}^{\mathcal{T}}$ (seen or unseen) with $m$ accessible labels $\{Y^{\mathcal{T}}\} \in \mathcal{Y}^{\mathcal{T}}$ under $m$-shot setting ($m \ll n$), the goal is to fine-tune the pre-trained $\Theta^{\star}$ to adapt to predict the labels of the remaining unlabeled samples correctly.

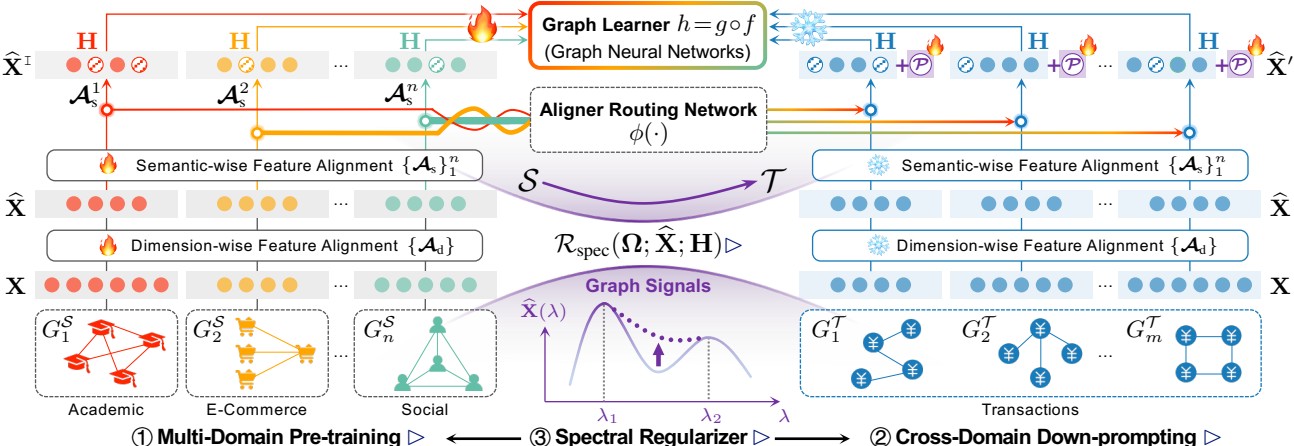

*Figure 1.* Framework of the proposed "pretrain-then-prompt"-based BRIDGE. ❶ **Multi-Domain Pre-training:** unifies multi-domain graph features with domain-invariant dimension- and semantic-wise aligners. ❷ **Cross-Domain Down-prompting:** facilitates feature adaptation with a lightweight MoE network. ❸ **Spectral Regularizer:** determines the maximum amount of transferable knowledge.

## 4. Proposed Method: BRIDGE

In this section, we elaborate on the proposed "pretrain-then-prompt"-based BRIDGE with its framework shown in Figure 1. During pre-training, we align multi-domain graph features with domain-invariant aligners. Then, a lightweight Mixture of Experts (MoE) network is proposed to facilitate downstream prompting. We derive an upper bound for the generalization error from the graph spectral perspective to determine the maximum amount of transferable knowledge.

### 4.1. Pre-training with Domain-Invariant Aligner

The primary goal of multi-domain graph pre-training is to learn domain-specific knowledge and transfer it to the target domain via parameters. However, the multi-domain graph features are often associated with inconsistent dimensions and intrinsic noise, which may hinder knowledge extraction. This motivates us to propose the domain-invariant aligner, which aligns features both dimensionally and semantically.

**Domain-Invariant Aligner.** We denote $\mathcal{A}_d$ as the aligner functioned for **d**imension-wise feature alignment, and $\mathcal{A}_s$ designed for **s**emantic-wise alignment. Specifically, given a graph $G_i = (\mathcal{V}_i, \mathcal{E}_i)$ $(i \leq n)$ with its feature matrix $\mathbf{X}_i^{\mathcal{S}} \in \mathbb{R}^{N_i \times d_i}$ from the source domain, $\mathcal{A}_d$ first acts dimension-wise alignment for every graph from each source domain,

$$\widehat{\mathbf{X}}_i^{\mathcal{S}} \in \mathbb{R}^{N_i \times d^{\mathcal{S}}} = \mathcal{A}_d(\mathbf{X}_i^{\mathcal{S}}), \quad \text{for all } G_i \in \{G^{\mathcal{S}}\}, \quad (1)$$

where $d^{\mathcal{S}}$ denotes the unified dimension for source graphs. There exist multiple implementations for the $\mathcal{A}_d$, where we combine SVD and an MLP, consequently. To further align semantic-wise features, encouraging pre-training model relying on domain-specific knowledge that better generalizes to unseen domains, we make following assumption *w.r.t.* Independent Causal Mechanism (ICM) (Pearl & Judea, 2009).

**Assumption 4.1** (**Domain Invariance**). *For each source domain $D_i \in \{D^{\mathcal{S}}\}$, there exist a domain-specific invariant aligner $\mathcal{A}_s^{\mathbb{I}}$ and variant aligner $\mathcal{A}_s^{\mathbb{V}}$ that lead to generalized graph representation under any target domains. Domain-invariant features $\widehat{\mathbf{X}}_i^{\mathbb{I}} = \mathcal{A}_s^{\mathbb{I}} \odot \widehat{\mathbf{X}}_i^{\mathcal{S}}$ should satisfy,*

- **Invariance Property:** *for any target domain $D_j \in \{D^{\mathcal{S}}\} \cup \{D^{\mathcal{T}}\}$, the equality $\mathbb{P}(Y_i \mid \widehat{\mathbf{X}}_i^{\mathbb{I}}, D_j) = \mathbb{P}(Y_i \mid \widehat{\mathbf{X}}_i^{\mathbb{I}})$ holds.*

- **Sufficient Condition:** *given any independent and random noise $\epsilon$, $Y_i = g(f_{\Theta}(\widehat{\mathbf{X}}_i^{\mathbb{I}}, \mathbf{A}_i)) + \epsilon$, i.e., $Y_i \perp \mathcal{A}_s^{\mathbb{V}} \mid \mathcal{A}_s^{\mathbb{I}}$, where $f_{\Theta}(\cdot)$ is the pre-trained graph encoder, $g(\cdot)$ is a discriminator. The logic structure syntaxes $\perp$ and $\mid$ represents label $Y_i$ is independent of $\mathcal{A}_s^{\mathbb{V}}$ but rely on $\mathcal{A}_s^{\mathbb{I}}$.*

Assumption 4.1 suggests the aligned invariant features $\widehat{\mathbf{X}}_i^{\mathbb{I}}$ governed by $\mathcal{A}_s^{\mathbb{I}}$ contain domain-specific causal correlations to their ground truth label $Y_i$, which is instrumental in facilitating the generalizability of the trained encoder $f_{\Theta}(\cdot)$ when adapting to unseen target domains. The Invariance Property ensures the aligner $\mathcal{A}_s^{\mathbb{I}}$ can always filter out discriminative features for each domain by eliminating intrinsic noise dimensions and spurious correlations. The Sufficient Condition guarantees the aligned $\widehat{\mathbf{X}}_i^{\mathbb{I}}$ is sufficiently robust for forecasting outcomes under the error budget $\epsilon$. We initialize $\mathcal{A}_s^{\mathbb{I}}$ as random dimension-wise 0-1 mask ($\mathcal{A}_s^{\mathbb{V}}$ is the complementary of $\mathcal{A}_s^{\mathbb{I}}$) projected by some learnable matrix.

**Generalizable Pre-training Objective.** Inspired by prior studies (Liu et al., 2023b), we adopt link prediction as the pre-training task, as there exist a vast number of links on the massive multi-domain graphs without requiring annotations, enabling the pre-training phase conducted in a self-supervised manner. We leverage a universal task template based on the node similarity, which solves task heterogeneity by incorporating classification and link prediction tasks.

A series of quadruples $(u, v_+, v_-, Y_u)$ are consistently and non-redundantly sampled from the source domain graphs $\{G^{\mathcal{S}}\}$, where $v_+$ is the direct neighbor of $u$, $v_-$ is the negative neighbor does not link to $u$, and $Y_u$ is the binary label indicates the link existence, given by the pair-wise node similarity discriminator $g = \text{MLP}(\text{sim}(\cdot, \cdot))$. The pre-training goal is to increase the semantic similarity between the node embedding $\mathbf{h}_u^{\mathcal{S}}$ and $\mathbf{h}_{v_+}^{\mathcal{S}}$ while decreasing that between $\mathbf{h}_u^{\mathcal{S}}$ and $\mathbf{h}_{v_-}^{\mathcal{S}}$. We formalize the task objective as,

$$\mathcal{L}_{\text{pre}}(\boldsymbol{\Theta}; \mathbf{H}^{\mathcal{S}}) = -\sum_{(u,v_+,v_-)} \ln \frac{\exp(g(\mathbf{h}_u^{\mathcal{S}}, \mathbf{h}_{v_+}^{\mathcal{S}})/\tau)}{\sum_{v \in \{v_+, v_-\}} \exp(g(\mathbf{h}_u^{\mathcal{S}}, \mathbf{h}_v^{\mathcal{S}})/\tau)}, \quad (2)$$

where $\tau$ is the temperature, $\mathbf{H}^{\mathcal{S}} = \text{stack}[f_{\boldsymbol{\Theta}}(\widehat{\mathbf{X}}_i^{\mathbb{I}}, \mathbf{A}_i)]_{i=1}^n$ is the invariant embedding obtained by the graph encoder.

**Alignment Risk Regularization.** It could be expected that pre-training $f_{\boldsymbol{\Theta}}$ via Eq. (2) has limited generalization capabilities to target domains as the joint distribution of graphs and their labels being conditioned on domain intrinsic distributions, *i.e.*, $\mathbb{P}(\mathcal{G}, \mathcal{Y} \mid \mathcal{D}) = \mathbb{P}(\mathcal{G} \mid \mathcal{D})\mathbb{P}(\mathcal{Y} \mid \mathcal{G}, \mathcal{D})$. To improve knowledge transferability of the pre-trained model, Eq. (2) is enhanced into a min-max dual optimization strategy on the multi-domain semantic alignment risk, *i.e.*,

$$\min_{\boldsymbol{\Theta}} \max_{D \in \{D^{\mathcal{S}}\} \cup \{D^{\mathcal{T}}\}} \mathbb{E}_{(G,Y) \sim \mathbb{P}(\mathcal{G}, \mathcal{Y}|\mathcal{D})} \big[ \mathcal{L}_{\text{pre}}(\boldsymbol{\Theta}; \mathbf{H}^{\mathcal{S}}) | D \big]. \quad (3)$$

As $\mathbb{P}(D)$ is unknown, directly optimize Eq. (3) is infeasible. Leveraged by the aligner $\mathcal{A}_{\text{s}}$, we transform Eq. (3) into,

$$\min_{\boldsymbol{\Theta}} \mathbb{E}\mathcal{L}_{\text{pre}}(\boldsymbol{\Theta}; \mathbf{H}^{\mathcal{S}}) + \alpha \mathbb{V}([\mathbb{E}\mathcal{L}_{\text{pre}}(\boldsymbol{\Theta}; \widetilde{\mathbf{H}}_k^{\mathcal{S}})]_{k=1}^K), \quad (4)$$

where, $\widetilde{\mathbf{H}}_k^{\mathcal{S}} = f_{\boldsymbol{\Theta}}(\mathcal{A}_{\text{s}}^{\mathbb{I}} \odot \widehat{\mathbf{X}}^{\mathcal{S}} + (\mathcal{A}_{\text{s}}^{\mathbb{V}} \odot \widehat{\mathbf{X}}^{\mathcal{S}})\mathbf{W}_k, \mathbf{A}). \quad (5)$

Note that, $\alpha$ and $K$ are hyperparameters, $\mathbb{V}$ denotes variance. Parameter of GNN $\boldsymbol{\Theta}^{\star}$ is fixed once pre-training converges.

**Proposition 4.2 (Achievable Assumption).** *Minimizing Eq. (4) encourages pre-trained $f_{\boldsymbol{\Theta}}^{\star}$ to satisfy Assumption 4.1.*

**Proposition 4.3 (Equivalent Optimization).** *Optimizing Eq. (4) is equivalent to minimizing the upper bound of multi-domain semantic alignment risk in Eq. (3).*

Proposition 4.2 avoids overly strong assumption, and Proposition 4.3 guarantees the transformation from Eq. (3) to Eq. (2) is equivalent. This highlights the role of aligners, which not only align dimension- and semantic-wise multi-domain graph features, but also enhance the generalization potential and model robustness during the in-context pre-training. We provide detailed proof in Appendix C.

### 4.2. Down-prompting via Self-Supervised MoE

To bridge the gap between pre-training and downstream tasks (Liu et al., 2023b), our focus shifts towards transferring multi-domain knowledge acquired during pre-training to downstream domains to overcome domain misalignment.

**Dimension-wise Adaptation.** Similarly, given any sample (node or graph) $G_i = (\mathcal{V}_i, \mathcal{E}_i)$ $(i \leqslant m)$ with feature $\mathbf{X}_i^{\mathcal{T}} \in \mathbb{R}^{N_i \times d_i}$ from target domain $D_j \in \{D^{\mathcal{T}}\}$, we then utilize the same dimension-wise aligner $\mathcal{A}_{\text{d}}$ to unify dimensionality,

$$\widehat{\mathbf{X}}_i^{\mathcal{T}} \in \mathbb{R}^{N_i \times d^{\mathcal{T}}} = \mathcal{A}_{\text{d}}(\mathbf{X}_i^{\mathcal{T}}), \quad \text{for all } G_i \in \{G^{\mathcal{T}}\}, \quad (6)$$

where dimension $d^{\mathcal{T}} = d^{\mathcal{S}} = d$ unless otherwise specified.

**Selective Knowledge Assembly.** However, performing further semantic-level feature alignment in the target domain proves challenging, primarily as no target domain knowledge is available during the pre-training. Building on the assumption that domain experts share common knowledge, we introduce a lightweight MoE network that selectively assembles source domain aligners (experts) to facilitate feature adaptation in the target domain,

$$\widehat{\mathbf{X}}_i^{\mathcal{T}\prime} = (\mathbf{S}^{\top} \mathcal{A}_{\text{s}}^{\mathbb{I}}) \odot \widehat{\mathbf{X}}_i^{\mathcal{T}}, \mathbf{S} \in \mathbb{R}^n = \text{Softmax}(\phi(\mathbf{X}^{\mathcal{T}})), \quad (7)$$

where $\mathbf{S}$ is the assignment weight vector, $\phi(\cdot)$ is a learnable routing network initialized by $\mathbf{X}^{\mathcal{T}}$. Intrinsic source-domain knowledge carried in $\mathcal{A}_{\text{s}}$ are assembled and transferred to target domains aiding feature adaptation. Assembly MoE is constructed in a lightweight scheme as it requires no extra training for source-domain experts, and is optimized in a self-supervised way by minimizing its assembly uncertainty,

$$\mathcal{L}_{\text{MoE}}(\mathcal{A}_{\text{s}}; \mathbf{S}) = H(\mathbf{S}) = -\sum_{i=1}^n \mathbf{S}_i \log(\mathbf{S}_i). \quad (8)$$

**Fine-tuning with Prompt Learning.** The unification of pre-training and downstream tasks enhances knowledge transfer by the universal task template based on subgraph similarity. However, it remains crucial to differentiate domain-specific characteristics. Inspired by prompts in natural language processing (Brown et al., 2020), where they are traditionally handcrafted instructions that steer downstream learning with task-specific cues, we propose to recall potential knowledge with learnable prompts as few-shot graph learners, *i.e.*,

$$\mathbf{H}_i^{\mathcal{T}} = f_{\boldsymbol{\Theta}}^{\star}(\mathcal{P}_{\boldsymbol{\Omega}} \odot \widehat{\mathbf{X}}_i^{\mathcal{T}\prime}, \mathbf{A}_i), \quad (9)$$

where $\mathcal{P}$ is the graph prompt with initialized tunable vector $\boldsymbol{\Omega}$, which is expected to be updated within constrained gradient descent epochs on $m$-shot supporting samples.

**Overall Down-prompting Objective.** As we aim at both the node and graph classification tasks, we formulate the down-prompting objective with the same universal task template as pre-training. Given $m$ $(m \ll n)$ supporting (training) samples $\{(G_i^{\mathcal{T}}, Y_i^{\mathcal{T}})\}_{i=1}^m$, where $\mathbf{H}^{\mathcal{T}}$ denotes the node (for node classification) or ego-graph embeddings (for graph classification), the downstream classification objective is transformed into determining the highest similarity between the query sample and class prototype embedding, *i.e.*,

$$\mathcal{L}_{\text{cls}}(\boldsymbol{\Omega}; \mathbf{H}^{\mathcal{T}}) = -\sum_{(G_i^{\mathcal{T}}, Y_i^{\mathcal{T}})} \ln \frac{\exp(g(\mathbf{H}_i^{\mathcal{T}}, \overline{\mathbf{H}}_{Y_i}^{\mathcal{T}})/\tau)}{\sum_{Y_j \in \{Y^{\mathcal{T}}\}} \exp(g(\mathbf{H}_i^{\mathcal{T}}, \overline{\mathbf{H}}_{Y_j}^{\mathcal{T}})/\tau)}, \quad (10)$$

where $\overline{\mathbf{H}}_{Y_i}^{\mathcal{T}}$ is the mean embedding (prototype) for samples in class $Y_i$. Then, the overall down-prompting objective is,

$$\mathcal{L}_{\text{down}} = \mathcal{L}_{\text{cls}}(\mathbf{\Omega}; \mathbf{H}^{\mathcal{T}}) + \beta \mathcal{L}_{\text{MoE}}(\mathcal{A}_s; \mathbf{S}), \qquad (11)$$

where $\beta$ is the trade-off hyperparameter.

### 4.3. Optimizing by Minimizing Generalization Error Upper Bound During Knowledge Transfer

To address how much knowledge can be transferred from the source to target domains, we reformulate this problem by minimizing the generalization error, which allows us to maximize transferable knowledge during down-prompting.

**Generalization Error Upper Bound.** Theoretically, the amount of transferable knowledge depends on the empirical adaptation errors. To this end, we leverage graph spectral theory and Lipschitz continuity to derive an upper bound that explicitly incorporates these factors.

**Assumption 4.4** (Lipschitz Continuity). *Suppose $f$ and $g$ are Lipschitz continuous with constants $L_f$ and $L_g$, satisfies,*

$$L_f \triangleq \|f\|_{Lip} = \sup_{G_i, G_j} \frac{\|f(\mathbf{X}_i, \mathbf{A}_i) - f(\mathbf{X}_j, \mathbf{A}_j)\|_2}{d_W(G_i, G_j)}, \quad (12)$$

$$L_g \triangleq \|g\|_{Lip} = \sup_{\mathbf{H}_i, \mathbf{H}_j} \frac{\|g(\mathbf{H}_i, \cdot) - g(\mathbf{H}_j, \cdot)\|_2}{d_E(\mathbf{H}_i, \mathbf{H}_j)}, \quad (13)$$

*where $d_W$ denotes the first Wasserstein distance (Villani et al., 2009), and $d_E$ denotes the Euclidean distance.*

The Lipschitz Continuity guarantees the smoothness of $f$ and $g$, laying the groundwork for constraining $h = g \circ f$ with a Lipschitz constant $L_h$, which cooperates with the Wasserstein distance to establish a connection between cross-domain distribution shifts and the generalization error.

**Proposition 4.5** (Generalization Error Upper Bound (Redko et al., 2017; Li et al., 2021; You et al., 2023)). *Let $\mathcal{H} = \{h \colon \mathcal{G} \mapsto \mathcal{Y} | h = g \circ f\}$ represent the hypothesis space of bounded real-valued functions, where $\mathrm{VCdim}(\mathcal{H}) = d$. Suppose $h^{\mathcal{S}} \neq h^{\mathcal{T}}$ with the Lipschitz constant $L_h$ satisfies,*

$$L_h \triangleq \|h\|_{Lip} = \sup_{G_i, G_j} \frac{\|h^D(G_i) - h^D(G_j)\|_2}{d_W(G_i, G_j)} \leqslant L_f L_g, \, (14)$$

*where $D \in \{\mathcal{S}, \mathcal{T}\}$. With probability no less than $1 - \delta$, the following inequality holds for target domain error $\epsilon^{\mathcal{T}}(h^{\mathcal{T}})$,*

$$\epsilon^{\mathcal{T}}(h^{\mathcal{T}}) \leqslant \overset{[a]}{\frac{m}{n+m}} \widehat{\epsilon}^{\mathcal{T}}(h^{\mathcal{T}}) + \overset{[b]}{\frac{n}{n+m}} \widehat{\epsilon}^{\mathcal{S}}(h^{\mathcal{S}})$$

$$+ \overset{[c]}{2 L_f L_g d_W} \left(\mathbb{P}(G^{\mathcal{S}}), \mathbb{P}(G^{\mathcal{T}})\right) + \overset{[d]}{\Delta_h}$$

$$+ \mathcal{O}\left[\left(\frac{1}{m} + \frac{1}{n}\right) \log \frac{1}{\delta} + \frac{d}{n} \log \frac{n}{d} + \frac{d}{m} \log \frac{m}{d}\right]^{\frac{1}{2}}, (15)$$

*where $\widehat{\epsilon}$ indicates empirical error, and $\Delta_h$ satisfies,*

$$\Delta_h = \min\left(\left|\epsilon^{\mathcal{S}}(h^{\mathcal{S}}) - \epsilon^{\mathcal{S}}(h^{\mathcal{T}})\right|, \left|\epsilon^{\mathcal{T}}(h^{\mathcal{S}}) - \epsilon^{\mathcal{T}}(h^{\mathcal{T}})\right|\right). (16)$$

**Bound Implementations.** To instantiate Eq. (15), we decompose the generalization bound into four key components from $[a]$ to $[d]$. Terms $[a]$ and $[b]$ represent the empirical error on target and source domains, which can be equivalently optimized by Eq. (10) and Eq. (2), respectively.

In such ways, the lower bound of the amount of transferable knowledge (the upper bound of generalization error) is constrained by both the domain-divergence (term $[c]$) and the discriminative capability of the graph encoder $f_{\mathbf{\Theta}}$ (term $[d]$), both of which are restricted by the Lipschitz constant $L_f$.

**Proposition 4.6** (The Lower Bound of $L_f$ (Gama et al., 2020; You et al., 2023)). *Rewrite the first Wasserstein distance (optimal transport distance) between $G_i$ and $G_j$,*

$$d_W(G_i, G_j) = \inf_{\mathbf{P} \in \Pi} \left(\|\mathbf{X}_i - \mathbf{P}\mathbf{X}_j\|_F + \|\mathbf{A}_i - \mathbf{P}\mathbf{A}_j\mathbf{P}^\top\|_F\right), (17)$$

*where $\mathbf{P}$ is the optimized permutation matrix by minimizing $d_W(G_i, G_j)$, $\Pi$ is matrix set. Such that, the lower bound of the Lipschitz constant of the encoder $f$ is,*

$$L_f \geqslant L_\lambda \left(1 + \Gamma_{ij}\sqrt{N_\Delta}\right) + \mathcal{O}(\Gamma^2), \qquad (18)$$

$$\Gamma = \|\mathbf{A}_i - \mathbf{P}^\star \mathbf{A}_j \mathbf{P}^{\star\top}\|_F, \Gamma_{ij} = (\|\mathbf{U}_i - \mathbf{U}_j\|_F + 1)^2 - 1, (19)$$

*where $\mathbf{P}^\star$ is the optimal permutation, $\mathbf{A}_i = \mathbf{U}_i \mathbf{\Lambda}_i \mathbf{U}_i^\top$ is the eigenvalue decomposition for both the adjacency matrices. $L_\lambda$ is the spectral Lipschitz constant, defined such that for any eigenvalues $\lambda_1$, $\lambda_2$, the inequality $|r(\lambda_1) - r(\lambda_2)| \leqslant L_\lambda |\lambda_1 - \lambda_2|$, where $r(\cdot)$ is the frequency response. $N_\Delta$ is graph size after padding isolated nodes (Zhu et al., 2021).*

Eq. (18) bounds $L_f$ in a graph spectral perspective by characterizing its smoothness. This suggests that changes in the Lipschitz constant $L_f$ inherently affect the balance between domain divergence (term $[c]$) and discriminability (term $[d]$).

**Bound as Spectral Regularizer.** We transform $L_f$ bound to regularization on both spectral and feature signals by evaluating its smoothness over consecutive spectral domains (You et al., 2023). Considering our "pretrain-then-prompt" settings, the constraints should be restricted to target domains aiming at fine-tuning graph prompt $\mathcal{P}_{\mathbf{\Omega}}$. Given transformed signals $\widetilde{\mathbf{H}}_i^{\mathcal{T}} = \mathbf{U}_i^\top \mathbf{H}_i^{\mathcal{T}}, \widetilde{\mathbf{X}}_i^{\mathcal{T}} = \mathbf{U}_i^\top \widehat{\mathbf{X}}_i^{\mathcal{T}}$, the regularizer is,

$$\mathcal{R}_{\text{spec}}(\mathbf{\Omega}; \widehat{\mathbf{X}}^{\mathcal{T}}; \mathbf{H}^{\mathcal{T}})$$
$$= \frac{1}{m} \sum_{i=1}^{m} \text{sum}\left[\sigma\left(\left|\Delta_r(\widetilde{\mathbf{H}}_i^{\mathcal{T}})\right| - \nu\left|\Delta_r(\mathbf{\Lambda}_i \widetilde{\mathbf{X}}_i^{\mathcal{T}})\right|\right)\right], \quad (20)$$

where $\Delta_r(\mathbf{M}) \leftarrow \mathbf{M}[2:N_\Delta, :] - \mathbf{M}[1:N_\Delta - 1, :]$ is row-wise differencing, reflecting signal deviations, $\nu$ is the threshold, and $\sigma$ is the non-linear activator implemented by $\text{ReLU}(\cdot)$.

**Bounded Down-prompt Fine-tuning.** In summary, we update the overall down-prompting objective in Eq. (11) to,

$$\lceil \mathcal{L}_{\text{down}} \rceil = \mathcal{L}_{\text{down}} + \gamma \, \mathcal{R}_{\text{spec}}(\mathbf{\Omega}; \widehat{\mathbf{X}}^{\mathcal{T}}; \mathbf{H}^{\mathcal{T}}). \qquad (21)$$

Training pipeline in Appendix B, proofs in Appendix C.

# 5. Experiments

In this section, we conduct extensive experiments to evaluate BRIDGE[1]. We focus on the following research questions,

- ***RQ1:*** How effective for few-shot classification?
- ***RQ2:*** Which section contributes most to the performance?
- ***RQ3:*** How time-efficient during down-prompting phase?
- ***RQ4:*** How sensitive to hyperparameters fluctuation?
- ***RQ5:*** Can LLMs enhance knowledge transferrability?

Detailed experimental setups, additional results, and analysis are provided in Appendix D and Appendix E.

## 5.1. Experimental Settings

**Datasets.** To highlight multi-domain pre-training, we select **six** benchmark graph datasets from **three** different domains, which differs from conventional settings that consider any single dataset as one independent domain. Specifically,

- ***Academic Domain:*** three citation networks Cora (McCallum et al., 2000), CiteSeer (Giles et al., 1998), and PubMed (Sen et al., 2008), where nodes represent publications, and edges denote citations.
- ***E-Commerce Domain:*** two Amazon product networks Photo (Shchur et al., 2018) and Computers (Shchur et al., 2018), where nodes correspond to products, and edges indicate frequent and related co-purchases.
- ***Social Network Domain:*** including Reddit (Hamilton et al., 2017), which is a social network constructed from its platform, where nodes represent posts, and edges mean interactions, like comments and replies between posts.

**Baselines.** We compare BRIDGE with **four** primary categories, **15** state-of-the-art baselines. Specifically,

- ***Vanilla Graph Neural Networks:*** including GCN (Kipf & Welling, 2022) and GAT (Velickovic et al., 2017), which are trained in an end-to-end manner without pre-training.
- ***Graph Self-Supervised Pre-training:*** GCC (Qiu et al., 2020), DGI (Veličković et al., 2019), InfoGraph (Sun et al., 2020), GraphCL (You et al., 2020), DSSL (Xiao et al., 2022), and GraphACL (Xiao et al., 2024), which utilize self-supervised objectives to pre-train and finetune.
- ***Graph Prompt Fine-tuning:*** we select pioneering works including GPPT (Sun et al., 2022), GraphPrompt (Liu et al., 2023b), GraphPrompt+ (Yu et al., 2024b), GPF (Fang et al., 2024), and ProNoG (Yu et al., 2025).
- ***Multi-Domain Graph Pre-training:*** we select the most closely related baselines, including GCOPE (Zhao et al., 2024) and MDGPT (Yu et al., 2024c).

[1] https://github.com/RingBDStack/BRIDGE.

**Multi-Domain Pre-training Settings.** We mainly focus on the challenging task of generalizing to unseen datasets and domains in downstream applications, incorporating the distinctions between cross-dataset and cross-domain setups,

- ***Cross-Dataset:*** the goal is to evaluate cross-dataset performance when pre-training and down-prompting shares the **same** multiple domains but **different** datasets. Specifically, we take ***Academic*** (Cora, CiteSeer, PubMed) and ***E-Commerce*** (Photo, Computers) as the shared domains. When each individual dataset is used as the target alternately, the remaining four datasets are collectively used as the pre-training source datasets.
- ***Cross-Domain:*** extends to a more challenging scenario, where the target dataset belongs to an **unseen** domain than the source domain. Specifically, we take **all** datasets in ***Academic*** and ***E-Commerce*** domains as the source during pre-training, while ***Social Network*** (Reddit) as target.

**Few-shot Down-prompting Settings.** We evaluate node and graph classifications as the down-prompting task under the $m$-shot setting, where the $m$ samples (nodes or graphs) from each class are randomly selected as supervision. As each dataset consists of a single graph that is not directly feasible for graph classification, we construct ego-graphs centered on the target nodes within each dataset (Lu et al., 2021; Yu et al., 2024a;c), which are then applied for graph classification, with each ego-graph labeled aligned with its central node. We assess the model performance by accuracy (Acc.) as few-shot is intrinsically a balanced classification.

## 5.2. *RQ1:* Few-Shot Node and Graph Classification

**One-Shot Classification.** We evaluate our BRIDGE against 15 state-of-the-art baselines under the challenging one-shot settings for node and graph classification of its effectiveness by transferring source knowledge in cross-dataset and cross-domain scenarios. Results reported in Table 1 and Table 2 illustrate foundational findings. ❶ Overall, BRIDGE outperforms all 15 baselines in both node and graph classification with an average improvement of 3.41% and 3.25%, respectively, which is a relatively promising advancement with the extreme one-shot supervision. ❷ The most closely related GCOPE and MDGPT fail in both tasks. GCOPE shows suboptimal performance due to its implicit and inadequate domain feature alignment. MDGPT collapses mainly because its shallow SVD dimension-wise feature alignment with the lack of knowledge transferring risk regularization. ❸ Cross-domain adaptation poses a more difficult source knowledge transfer condition compared with the cross-dataset settings, and BRIDGE shows satisfying results when not only the source dataset is unseen, but the domains are also not accessible. ❹ Additional results for **five-shot** setting in Appendix E.1 demonstrate the same trend compared with the above.

*Table 1.* Accuracy (% ± standard deviation for five runs) of **one-shot node** classification. `CR` =Cora, `CS` =CiteSeer, `PM` =PubMed, `Ph` =Photo, `Com` =Computers, `Rdt` =Reddit. The best results are shown in **bold** and the runner-ups are underlined.

| Source (Cross-Dataset \| Cross-Domain) | CS PM Ph Com | CR PM Ph Com | CR CS Ph Com | CR CS PM Com | CR CS PM Ph | CR CS PM Ph Com |
|---|---|---|---|---|---|---|
| **Model / Target** | CR | CS | PM | Ph | Com | Rdt |
| GCN (bb.) (Kipf & Welling, 2022) | 29.36 ± 4.20 | 30.99 ± 4.85 | 40.94 ± 7.05 | 40.05 ± 7.25 | 34.62 ± 8.95 | 52.33 ± 7.26 |
| GAT (Velickovic et al., 2017) | 29.00 ± 5.33 | 29.46 ± 3.32 | 40.09 ± 5.21 | 35.60 ± 6.52 | 33.15 ± 6.59 | 50.70 ± 7.06 |
| GCC (Qiu et al., 2020) | 31.67 ± 5.23 | 32.55 ± 2.69 | 41.66 ± 4.58 | 42.10 ± 5.99 | 35.91 ± 5.68 | 55.67 ± 6.01 |
| DGI (Veličković et al., 2019) | 30.82 ± 4.41 | 31.85 ± 4.36 | 40.08 ± 6.22 | 47.23 ± 6.03 | 37.05 ± 6.40 | 61.95 ± 6.22 |
| GraphCL (You et al., 2020) | 33.28 ± 6.03 | 29.12 ± 4.26 | 39.31 ± 7.05 | 42.98 ± 6.54 | 42.87 ± 5.32 | 60.06 ± 5.29 |
| DSSL (Xiao et al., 2022) | 30.65 ± 5.24 | 31.20 ± 6.33 | 40.89 ± 6.94 | 45.32 ± 7.28 | 38.41 ± 6.54 | 57.23 ± 6.61 |
| GraphACL (Xiao et al., 2024) | 35.26 ± 4.65 | 34.09 ± 6.40 | 43.54 ± 5.58 | 49.20 ± 7.00 | 41.86 ± 6.20 | 60.97 ± 3.39 |
| GPPT (Sun et al., 2022) | 32.18 ± 5.02 | 31.33 ± 4.20 | 41.27 ± 5.60 | 46.98 ± 4.04 | 35.18 ± 8.09 | 55.24 ± 6.87 |
| GraphPrompt (Liu et al., 2023b) | 37.95 ± 6.31 | 34.92 ± 6.75 | 45.85 ± 8.58 | 50.42 ± 8.05 | 42.58 ± 7.21 | 64.71 ± 5.21 |
| GraphPrompt+ (Yu et al., 2024b) | 36.06 ± 6.93 | 33.85 ± 7.62 | 45.08 ± 7.99 | 52.28 ± 7.51 | 43.39 ± 6.90 | 64.12 ± 6.25 |
| GPF (Fang et al., 2024) | 40.26 ± 8.33 | 40.20 ± 7.10 | 47.33 ± 6.63 | 51.48 ± 5.34 | 40.09 ± 6.19 | 63.52 ± 6.37 |
| ProNoG (Yu et al., 2025) | 44.57 ± 6.57 | 39.96 ± 7.99 | 50.48 ± 7.06 | 63.30 ± 4.92 | 50.29 ± 6.32 | 64.87 ± 5.18 |
| GCOPE (Zhao et al., 2024) | 35.29 ± 4.29 | 40.75 ± 4.28 | 44.55 ± 6.07 | 51.55 ± 5.36 | 43.74 ± 7.05 | 64.21 ± 4.22 |
| MDGPT (Yu et al., 2024c) | 42.29 ± 7.75 | 37.32 ± 7.01 | 50.89 ± 7.74 | 63.63 ± 7.23 | 49.78 ± 8.77 | 63.11 ± 3.01 |
| **BRIDGE (ours)** | **46.44 ± 8.01** | **42.18 ± 8.89** | **56.35 ± 7.22** | **67.87 ± 7.45** | **54.04 ± 8.52** | **68.55 ± 3.49** |

*Table 2.* Accuracy (% ± standard deviation for five runs) of **one-shot graph** classification. `CR` =Cora, `CS` =CiteSeer, `PM` =PubMed, `Ph` =Photo, `Com` =Computers, `Rdt` =Reddit. The best results are shown in **bold** and the runner-ups are underlined.

| Source (Cross-Dataset \| Cross-Domain) | CS PM Ph Com | CR PM Ph Com | CR CS Ph Com | CR CS PM Com | CR CS PM Ph | CR CS PM Ph Com |
|---|---|---|---|---|---|---|
| **Model / Target** | CR | CS | PM | Ph | Com | Rdt |
| GCN (bb.) (Kipf & Welling, 2022) | 38.59 ± 6.33 | 29.90 ± 7.20 | 47.26 ± 7.12 | 56.02 ± 5.28 | 39.29 ± 6.25 | 62.77 ± 4.22 |
| GAT (Velickovic et al., 2017) | 35.64 ± 5.09 | 26.78 ± 6.91 | 41.72 ± 6.99 | 50.23 ± 4.98 | 37.25 ± 6.73 | 60.13 ± 4.95 |
| InfoGraph (Sun et al., 2020) | 41.76 ± 4.89 | 31.03 ± 5.07 | 47.92 ± 6.76 | 59.88 ± 5.09 | 41.33 ± 6.67 | 63.28 ± 5.06 |
| GraphCL (You et al., 2020) | 40.23 ± 5.28 | 32.99 ± 5.29 | 49.21 ± 8.16 | 58.02 ± 5.11 | 42.09 ± 5.38 | 64.01 ± 5.17 |
| DSSL (Xiao et al., 2022) | 40.95 ± 6.29 | 32.03 ± 6.80 | 49.32 ± 7.13 | 58.19 ± 6.10 | 41.94 ± 6.23 | 63.90 ± 6.19 |
| GraphACL (Xiao et al., 2024) | 41.08 ± 5.77 | 32.38 ± 5.91 | 49.25 ± 6.22 | 59.66 ± 5.27 | 43.18 ± 5.99 | 64.82 ± 6.22 |
| GraphPrompt (Liu et al., 2023b) | 42.18 ± 4.31 | 38.66 ± 6.12 | 51.28 ± 5.29 | 60.07 ± 5.70 | 47.31 ± 5.11 | 65.56 ± 5.18 |
| GraphPrompt+ (Yu et al., 2024b) | 43.87 ± 5.18 | 40.07 ± 6.21 | 51.83 ± 5.69 | 61.30 ± 5.91 | 47.08 ± 6.32 | 66.19 ± 5.97 |
| GPF (Fang et al., 2024) | 41.82 ± 8.01 | 39.00 ± 7.45 | 47.23 ± 8.90 | 59.21 ± 5.57 | 46.03 ± 9.01 | 66.92 ± 5.23 |
| ProNoG (Yu et al., 2025) | 50.98 ± 7.33 | 47.37 ± 7.08 | 55.35 ± 7.90 | 63.29 ± 6.33 | 50.21 ± 7.29 | 73.09 ± 4.03 |
| GCOPE (Zhao et al., 2024) | 44.29 ± 7.23 | 39.10 ± 8.39 | 50.82 ± 8.10 | 61.83 ± 6.48 | 51.12 ± 7.13 | 70.19 ± 5.12 |
| MDGPT (Yu et al., 2024c) | 49.11 ± 7.63 | 43.56 ± 8.25 | 53.90 ± 9.83 | 63.41 ± 6.70 | 49.56 ± 7.63 | 73.14 ± 6.39 |
| **BRIDGE (ours)** | **53.84 ± 7.52** | **50.49 ± 9.09** | **58.41 ± 8.52** | **68.33 ± 5.65** | **54.06 ± 7.55** | **75.76 ± 4.04** |

**$m$-Shot Classification.** To additionally evaluate BRIDGE's performance under different few-shot settings varied in $m$, we selected three competitive baselines in ***RQ1***, including `ProNoG`, `GCOPE` and `MDGPT` for comparison. Results on both the cross-dataset (`Cora`) and cross-domain (`Reddit`) tasks are shown in Figure 2 and Figure 3. (Additional results are included in Appendix E.2) Results indicate that, ❶ BRIDGE outperforms the other three baselines in nearly all $m$-shot settings. However, in very low-shot scenarios, BRIDGE's performance slightly lags behind the baselines. This is due to the initial stabilization of its graph prompt learning, which depends on the generalization error bound and requires a few iterations to adapt with limited supervision. In contrast, baselines like `ProNoG` have optimized

parameters for low-shot settings, allowing them to outperform BRIDGE initially before its regularization takes effect. ❷ BRIDGE shows a slight advantage when $m$ is small, with its advantage gradually expanding as $m$ grows larger, which we consider as benefiting from the spectral regularizer $\mathcal{R}_{\text{spec}}$ that enables more effective fine-tuning on more sufficient supervision through down prompts with a generalization error upper bound guarantee. ❸ Results in both scenarios show similar trends, highlighting BRIDGE's advantages in knowledge transfer. It demonstrates superior robustness in cross-domain adaptation, particularly when the source and target domains differ, effectively transferring knowledge even without domain-specific data in the target domain, which is crucial for real-world applications.

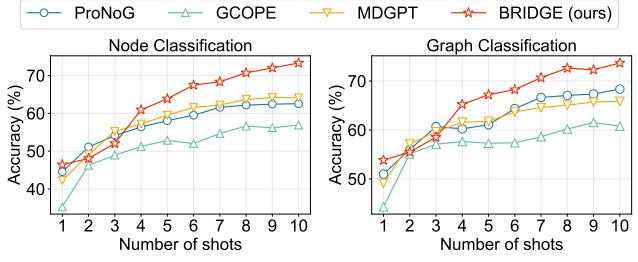

*Figure 2.* $m$-shot evaluation on `Cora` (cross-dataset).

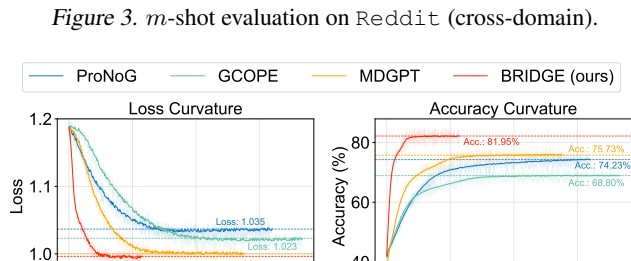

*Figure 3.* $m$-shot evaluation on `Reddit` (cross-domain).

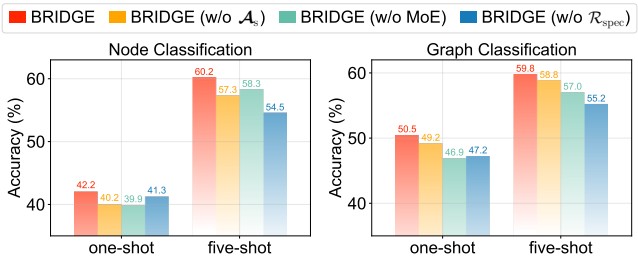

*Figure 4.* Ablation studies on `CiteSeer`.

*Figure 5.* Down-prompt Fine-tuning on `Photo` ($m = 5$).

## 5.3. *RQ2:* Ablation Study

We conduct ablation studies concerning three key sections,

- **BRIDGE** (*w/o* $\mathcal{A}_s$): remove the semantic-wise aligner $\mathcal{A}_s$, and only align multi-domain features with $\mathcal{A}_d$.

- **BRIDGE** (*w/o* MoE): remove the lightweight MoE, and replace the routing network by a simple average routing.

- **BRIDGE** (*w/o* $\mathcal{R}_{spec}$): remove the spectral regularizer $\mathcal{R}_{spec}$ in the bounded down-prompt fine-tuning objective.

Results (Figure 4) show that, ❶ All three sections demonstrate effectiveness in boosting BRIDGE's knowledge transferability in classification performance. ❷ The lightweight MoE has a particularly pronounced impact on the one-shot learning. It selectively assembles expert knowledge from pre-training, which helps compensate for the limited supervision and makes it an essential component for effective adaptation in low-shot scenarios. ❸ The spectral regularizer $\mathcal{R}_{spec}$ becomes increasingly beneficial in the five-shot learning. It helps constrain the generalization error during cross-dataset and cross-domain tasks by leveraging more target supervision. ❹ Removing the semantic-wise aligner $\mathcal{A}_s$ results in a noticeable drop in performance, highlighting the importance of semantic alignment in ensuring effective knowledge transfer across domains, especially when feature semantics vary. ❺ The performance improvement observed when including all components suggests that each plays a complementary role in enhancing BRIDGE's robustness. The MoE and spectral regularizer are especially valuable in scenarios requiring efficient adaptation to unseen data.

## 5.4. *RQ3:* Efficiency Evaluation for Down-prompting

Graph prompt learning necessitates efficient training with a limited number of tunable parameters and epochs. To assess the down-prompting efficiency of BRIDGE, we analyze the loss and accuracy training curves for node classification task on the `Photo` dataset with $m = 5$ shots. Results (Figure 5) reveal that, ❶ BRIDGE outperforms the competitive baselines in both loss reduction and accuracy improvement within fewer epochs. Specifically, BRIDGE converges to a lower loss (0.996) and achieves a higher accuracy (81.95%) much faster compared to other methods. This demonstrates that BRIDGE's training process is more efficient, reaching optimal performance without requiring as many iterations, which is crucial when computational resources and time are limited. ❷ In contrast, other baselines show relatively higher loss values, ranging from 1.0 to 1.1, and achieve weaker performance (accuracies between 68% and 75%). These methods take longer to converge, which increases the overall training time and computational cost. This inefficiency could be problematic in real-world settings where resources are constrained. ❸ These results highlight BRIDGE's efficiency as a graph adapter. The model leverages fewer training iterations to reach higher accuracy and lower loss, which is a significant advantage when performing fine-tuning with limited labeled data or computational power. ❹ The efficiency validation demonstrated here reinforces that BRIDGE is well-suited for real-world applications, especially in scenarios where training time and computational cost are critical factors. This makes BRIDGE a viable option for environments where rapid deployment and scalability are essential.

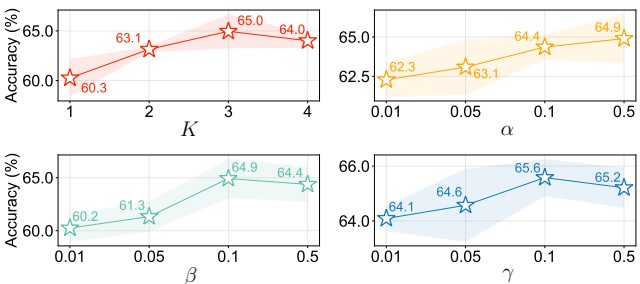

Figure 6. Hyperparameter studies on Computers ($m = 5$).

Table 3. Accuracy (%) of LLM-enhanced few-shot performance targeted on Reddit under the cross-domain setting.

| Source \| Target | | | CR CS PM Ph Com \| Rdt | | | |
|---|---|---|---|---|---|---|
| **Task** | Node Classification | | | Graph Classification | | |
| **Model / $m$** | 0 | 1 | 5 | 0 | 1 | 5 |
| MDGPT | 49.69 | 63.11 | 75.29 | 64.80 | 73.14 | 80.29 |
| MDGPT + LLM | 51.08 | 65.56 | 76.34 | 66.07 | 75.35 | 82.02 |
| BRIDGE | 55.72 | 68.55 | 80.12 | 67.18 | 75.76 | 84.42 |
| BRIDGE + LLM | 57.16 | 70.70 | 82.31 | 69.14 | 77.31 | 86.87 |

### 5.5. *RQ4:* **Hyperparamter Sensitivity Investigation**

We evaluate the sensitvity of important hyperparameters in graph classification on Computers in Figure 6, where,

- **$K$:** controls variant embedding augment times in Eq. (4).
- **$\alpha$:** trades off between expectation and variation in Eq. (4).
- **$\beta$:** balances between $\mathcal{L}_{cls}$ and $\mathcal{L}_{MoE}$ in Eq. (11).
- **$\gamma$:** acts between $\mathcal{L}_{down}$ and $\mathcal{R}_{spec}$ in Eq. (21).

Results (Figure 6) show that, ❶ BRIDGE is sensitive to $K$, with performance peaking at $K = 3$ (65.0%). A moderate number of embedding augmentations improves robustness, while excessive augmentation slightly degrades. ❷ Increasing $\alpha$ leads to steady performance gains, reaching 64.9% at $\alpha = 0.5$. This suggests that emphasizing the variation term in alignment risk enhances generalization. ❸ BRIDGE performs best when $\beta = 0.1$ (64.9%), indicating that moderate MoE regularization benefits expert selection. Higher values may over-constrain the adaptation process. ❹ $\gamma$ shows a clear effect on performance, with the best result (65.6%) at $\gamma = 0.1$. The spectral regularizer helps reduce generalization error, though overly strong constraints may hinder flexibility. ❺ All hyperparameters exhibit stable behavior within reasonable ranges, with identifiable optimal points, demonstrating the robustness and tunability of BRIDGE.

### 5.6. *RQ5:* **LLM-Enhanced Multi-Domain Alignment**

To explore the potential of language models in bridging cross-domain gaps, we leverage pre-trained large language models (LLMs) to generate textual descriptions for nodes based on raw graph data (Liu et al., 2024b; Li et al., 2024b). These generated descriptions serve as auxiliary node features and provide an implicit form of cross-domain alignment, especially for graphs without inherent textual attributes. Results (Table 3) demonstrate that, ❶ Both MDGPT and our BRIDGE perform relatively poorly in the zero-shot setting, which reflects the inherent difficulty of transferring knowledge across domains in the absence of any supervision. This finding is consistent with prior research, where the availability of semantic-rich text information significantly enhances model generalization. ❷ After incorporating LLM-generated descriptions, both models exhibit

noticeable improvements across all few-shot settings. For example, BRIDGE improves from 55.72% to 57.16% in zero-shot node classification, and from 84.42% to 86.87% in five-shot graph classification. This demonstrates that LLM-generated text effectively enhances representation quality even in graphs originally lacking textual attributes. ❷ Compared with MDGPT, BRIDGE benefits more consistently from LLM enhancement. This observation suggests that the architectural components of BRIDGE, such as its semantic aligner and mixture-of-experts routing, are more capable of leveraging the additional semantic cues introduced by LLMs. ❸ The results support the view that integrating LLM-generated features can significantly reduce domain shifts in cross-domain scenarios. Even when textual data is not available natively, generating pseudo-descriptions through language models provides a scalable and generalizable solution to improve knowledge transfer.

We can conclude that, enhancing graph models with LLM-generated features proves to be a promising strategy for improving few-shot performance, particularly under challenging settings where the target domain lacks labeled data.

## 6. Conclusion

We introduce a novel multi-domain graph pre-training and prompting framework BRIDGE with domain-invariant feature aligners and efficient prompt initializers. We analyze knowledge transfer generalization error and derive an optimizable bound as a spectral regularizer. Extensive experiments demonstrate its superiority in classifications compared with 15 state-of-the-art baselines. We also demonstrate compatibility by integrating with LLMs to enhance domain understanding and knowledge transferability.

## Acknowledgments

The corresponding author is Qingyun Sun ✉. This work is supported in part by NSFC under grants No.623B2010 and No.62225202, by NSF under grants III-2106758 and POSE-2346158, and by the Fundamental Research Funds for the Central Universities. We extend our sincere thanks to all authors for their valuable contributions.

## Impact Statement

This paper advances the development of Graph Foundation Models (GFMs) by introducing BRIDGE, a novel GFM pre-trained on multi-domains with bounded generalization guarantees. As GFMs emerge as a cornerstone for leveraging graph-structured data across diverse domains, this work provides a critical step toward robust and scalable knowledge transfer. With applications spanning social networks, e-commerce, and bioinformatics, this framework demonstrates the potential to significantly enhance multi-domain graph learning while maintaining computational efficiency. This work primarily aims to advance the field of machine learning, and we do not identify any specific ethical issues or societal risks requiring further attention.

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

# A. Notations

| Notations | Descriptions |
|---|---|
| $G = (\mathcal{V}, \mathcal{E})$ | Graph $G$ with node set $\mathcal{V}$ and edge set $\mathcal{E}$. |
| $\mathbf{A} \in \{0,1\}^{N \times N}$ | Adjacency matrix $\mathbf{A}$, where $N$ is the number of nodes. |
| $\mathbf{X} \in \mathbb{R}^{N \times d}$ | Node feature matrix $\mathbf{X}$, with $N$ nodes and $d$ feature dimensions. |
| $\{G^{\mathcal{S}}\} \in \mathcal{G}^{\mathcal{S}}, \{G^{\mathcal{T}}\} \in \mathcal{G}^{\mathcal{T}}$ | Graphs sampled from source domain ($\mathcal{S}$) and target domains ($\mathcal{T}$). |
| $\{Y^{\mathcal{S}}\} \in \mathcal{Y}^{\mathcal{S}}, \{Y^{\mathcal{T}}\} \in \mathcal{Y}^{\mathcal{T}}$ | Corresponding graph labels in source domain ($\mathcal{S}$) and target domains ($\mathcal{T}$). |
| $\{D^{\mathcal{S}}\} \in \mathcal{D}^{\mathcal{S}}, \{D^{\mathcal{T}}\} \in \mathcal{D}^{\mathcal{T}}$ | Source domains and targeted domains. |
| $\mathbb{P}(G^{\mathcal{S}}), \mathbb{P}(G^{\mathcal{T}})$ | Probability distributions of the source graph and target graph. |
| $f_{\boldsymbol{\Theta}}(\cdot) : \mathbb{R}^d \mapsto \mathbb{R}^h$ | Graph encoder with learnable parameter $\boldsymbol{\Theta}$. |
| $g(\cdot) : \mathbb{R}^h \times \mathbb{R}^h \mapsto \mathbb{R}^2$ | Graph discriminator composed of $\mathrm{sim}(\cdot, \cdot)$ and a binary MLP. |
| $h = g \circ f$ | The graph learner compound of the encoder $f(\cdot)$ and the binary discriminator $g(\cdot)$ |
| $r(\cdot)$ | Graph signal frequency response function. |
| $\phi(\cdot)$ | Learnable routing network in the MoE. |
| $\sigma(\cdot)$ | Non-linear activation function (ReLU). |
| $n$ | The number of training samples in the source domain. |
| $m$ | The number of supporting samples ($m$-shot setting) in the target domain. |
| $d$ | The aligned feature dimensions. |
| $\mathcal{A}_{\mathrm{d}} : \mathbb{R}^{d_i} \mapsto \mathbb{R}^d$ | Dimension-wise feature aligner (implemented by composing SVD and MLP). |
| $\mathcal{A}_{\mathrm{s}}^{\mathbb{I}}, \mathcal{A}_{\mathrm{s}}^{\mathbb{V}} \in \{0,1\}^{n \times d}$ | Semantic-wise invariant and variant feature aligner (implemented by a pair of masks). |
| $\mathbf{X}^{\mathcal{S}}, \widehat{\mathbf{X}}^{\mathcal{S}}, \widehat{\mathbf{X}}^{\mathbb{I}}$ | Source domain graph features, dimension-aligned features, and semantic-aligned domain-invariant features. |
| $\mathbf{H}^{\mathcal{S}}, \mathbf{h}^{\mathcal{S}}$ | Encoded source domain graph features and node features. |
| $\mathbf{X}^{\mathcal{T}}, \widehat{\mathbf{X}}^{\mathcal{T}}, \widehat{\mathbf{X}}^{\mathcal{T}\prime}$ | Target domain graph features, dimension-aligned features, and adapted target domain features assembled with the lightweight MoE. |
| $\mathbf{H}^{\mathcal{T}}, \mathbf{h}^{\mathcal{T}}$ | Encoded target domain graph features and node features. |
| $\mathbf{S}$ | Assignment weight vector for the MoE network. |
| $\mathbf{W}$ | Learnable weight matrix for projection. |
| $\mathcal{P}_{\boldsymbol{\Omega}}$ | Graph prompt with learnable vector $\boldsymbol{\Omega}$. |
| $\mathcal{H}$ | Hypothesis space, defined as a mapping from graphs to labels. |
| $\mathrm{VCdim}(\cdot)$ | The VC (Vapnik-Chervonenkis) dimension of the hypothesis space. |
| $L_f$ | Lipschitz constant for the graph encoder $f$. |
| $L_g$ | Lipschitz constant for graph discriminator $g$. |
| $L_\lambda$ | Lipschitz constant for graph spectral. |
| $\lambda_i$ | Eigenvalues of the graph spectral signal. |
| $d_{\mathrm{W}}$ | Wasserstein distances. |
| $d_{\mathrm{E}}$ | Euclidean distances. |
| $\epsilon^{\mathcal{S}}, \widehat{\epsilon}^{\mathcal{S}}$ | error and empirical error for generalization in the source domains. |
| $\epsilon^{\mathcal{T}}, \widehat{\epsilon}^{\mathcal{T}}$ | error and empirical error for generalization in the target domains. |
| $\Delta_h$ | Discrepancy between source and target domain errors. |
| $\Gamma, \Gamma_{ij}$ | Terms related to the smoothness of graph features in Eq. (19). |
| $\mathbf{P}, \mathbf{P}^\star$ | Permutation matrix and the optimized permutation matrix. |
| $\mathcal{L}_{\mathrm{pre}}$ | Pre-training loss function. |
| $\mathcal{L}_{\mathrm{MoE}}$ | Loss function for the MoE network. |
| $\mathcal{L}_{\mathrm{cls}}$ | Loss function for the down-prompting classification. |
| $\mathcal{L}_{\mathrm{down}}$ | Overall Loss function for the down-prompting phase few-shot training. |
| $\lceil \mathcal{L}_{\mathrm{down}} \rceil$ | Bounded overall Loss function for the down-prompting phase few-shot training. |
| $\mathcal{R}_{\mathrm{spec}}$ | Graph spectral regularizer for multi-domain generalization. |
| $K$ | Hyperparameter that controls variant embedding augmenting times in Eq. (4). |
| $\alpha$ | Hyperparameter trades-off between expectation and variation term in Eq. (4). |
| $\beta$ | Hyperparameter that balances between $\mathcal{L}_{\mathrm{cls}}$ and $\mathcal{L}_{\mathrm{MoE}}$ in Eq. (11). |
| $\gamma$ | Hyperparameter that acts between $\mathcal{L}_{\mathrm{down}}$ and $\mathcal{R}_{\mathrm{spec}}$ in Eq. (21). |
| $\tau$ | Temperature parameter. |

# B. Algorithm and Complexity Analysis

The pre-training pipeline of BRIDGE is illustrated in Algorithm 1, and the down-prompting (fine-tuning) pipeline is shown in Algorithm 2. Next, we theoretically analyze their computational complexity, respectively.

---

**Algorithm 1** Pre-training pipeline of BRIDGE.

---

**Input:** A set of $n$ source graphs $\{G^{\mathcal{S}}\} \in \mathcal{G}^{\mathcal{S}}$ from multi-domains $\{D^{\mathcal{S}}\} \in \mathcal{D}^{\mathcal{S}}$, where each of the graph $G_i$ is associate with its feature matrix $\mathbf{X}_i \in \mathbb{R}^{N_i \times d_i}$ and adjacency matrix $\mathbf{A} \in \mathbb{R}^{N_i \times N_i}$; Domain-invariant aligners $\mathcal{A}_{\mathrm{d}}$ and $\mathcal{A}_{\mathrm{s}}$; Learnable weight matrix for projection $\mathbf{W}$; Number of pre-training epochs $E_1$; Temperature parameter $\tau$; Number of variation times $K$; Hyperparameter $\alpha$ for trading-off expectation and variation terms.

**Output:** Pre-trained graph encoder and discriminator $g(f_{\Theta}^{\star}(\cdot))$.

Initialize all parameters randomly;

**for** $e = 1, 2, \cdots, E_1$ **do**

  `# Dimension-wise feature alignment`
  Aligned graph feature: $\widehat{\mathbf{X}}_i^{\mathcal{S}} \leftarrow$ Eq. (1) for each $G_i \in \{G^{\mathcal{S}}\}$;

  `# Semantic-wise feature alignment`
  Aligned domain-invariant feature: $\widehat{\mathbf{X}}_i^{\mathtt{I}} = \mathcal{A}_{\mathrm{s}}^{\mathtt{I}} \odot \widehat{\mathbf{X}}_i^{\mathcal{S}}$ with domain-invariant aligner $\mathcal{A}_{\mathrm{s}}^{\mathtt{I}}$;

  `# Generalizable pre-training objective`
  Encode and stack source domain graph embeddings: $\mathbf{H}^{\mathcal{S}} = \mathrm{stack}[f_{\Theta}(\widehat{\mathbf{X}}_i^{\mathtt{I}}, \mathbf{A}_i)]_{i=1}^n$;
  Calculate pre-training loss: $\mathcal{L}_{\mathrm{pre}}(\Theta; \mathbf{H}^{\mathcal{S}}) \leftarrow$ Eq. (2);

  `# Alignment Risk Regularization`
  **for** $k = 1, 2, \cdots, K$ **do**
    Calculate intervened embedding: $\widetilde{\mathbf{H}}_k^{\mathcal{S}} \leftarrow$ Eq. (5);
  **end for**
  Calculate alignment risk regularization by variations: $\mathcal{L}_{\mathrm{pre\_risk}} \leftarrow \mathbb{V}([\mathbb{E}\mathcal{L}_{\mathrm{pre}}(\Theta; \widetilde{\mathbf{H}}_k^{\mathcal{S}})]_{k=1}^K)$;

  `# Optimization`
  Calculate the overall pre-training loss: $\mathcal{L}_{\mathrm{pre\_overall}} \leftarrow \mathbb{E}\mathcal{L}_{\mathrm{pre}} + \alpha \mathcal{L}_{\mathrm{pre\_risk}}$;
  Update $\Theta$ by minimizing $\mathcal{L}_{\mathrm{pre\_overall}}$ and back-propagation.

**end for**

---

**Complexity Analysis of Algorithm 1.** We analyze the computational complexity of each part in Algorithm 1 as follows.

- **Dimension-wise feature alignment:** We implement $\mathcal{A}_{\mathrm{d}}$ by a two-layer MLP ($\mathbb{R}^{d_i} \mapsto \mathbb{R}^{d^{\mathcal{S}}}$). Its computational complexity takes $\mathcal{O}(\sum_{i=1}^n N_i d_i d^{\mathcal{S}})$, where $N_i$ represents node numbers in source graph $G_i$, and $d_i$ is its initial feature dimension. We omit the computational complexity of the preceding SVD computation here.

- **Semantic-wise feature alignment:** We implement $\mathcal{A}_{\mathrm{s}}^{\mathtt{I}}, \mathcal{A}_{\mathrm{s}}^{\mathtt{V}} \in \{0,1\}^{n \times d^{\mathcal{S}}}$ by a pair of 0-1 mask. The mask operations on $\widehat{\mathbf{X}}_i^{\mathcal{S}}$ takes the computational complexity of $\mathcal{O}(\sum_{i=1}^n) N_i d^{\mathcal{S}}$.

- **Feature encoding:** We encode $n$ source graphs by a two-layer GNN, which takes the complexity of $\mathcal{O}(\sum_{i=1}^n (|\mathcal{E}_i|(d^{\mathcal{S}} + d_h) + N_i d_h^2))$, where $|\mathcal{E}_i|$ is the number of edges, $d^{\mathcal{S}}$ is the aligned feature dimension, and $d_h$ is the hidden dimension.

- **Pre-training loss calculation:** Suppose there are $n_1$ direct neighbors, $n_2$ indirect neighbors (nodes that are connected) averaged for each node in the source domain graphs, we could sample $n \times n_1 \times n_2$ quadruples, which is utilized to formalize the self-supervised pre-training contrastive loss. Its computational complexity takes $\mathcal{O}(\sum_{i=1}^n (N_i(n_1 + n_2)d^{\mathcal{S}}))$.

- **Causal intervention on variant dimensions:** We perform $K$ times weighted projection as means of causal intervention, which takes the computational complexity of $\mathcal{O}(K(\sum_{i=1}^n N_i(d^{\mathcal{S}})^2))$.

- **Alignment risk regularization:** $K$ times variant feature encoding by the same two-layer GNN takes this computational complexity of $\mathcal{O}(K \sum_{i=1}^n (|\mathcal{E}_i|(d^{\mathcal{S}} + d_h) + N_i d_h^2))$.

The overall computational complexity for the pre-training phase takes,

$$\mathcal{O}\Big(\sum_{i=1}^n N_i d_i d^{\mathcal{S}}\Big) + \mathcal{O}\Big(\sum_{i=1}^n N_i d^{\mathcal{S}}\Big) + \mathcal{O}\Big(\sum_{i=1}^n (|\mathcal{E}_i|(d^{\mathcal{S}} + d_h) + N_i d_h^2)\Big) + \mathcal{O}\Big(\sum_{i=1}^n (N_i(n_1 + n_2)d^{\mathcal{S}})\Big) +$$

$$\mathcal{O}\Big(K\Big(\sum_{i=1}^n N_i(d^{\mathcal{S}})^2\Big)\Big) + \mathcal{O}\Big(K \sum_{i=1}^n \Big(|\mathcal{E}_i|(d^{\mathcal{S}} + d_h) + N_i d_h^2\Big)\Big). \tag{B.1}$$

Suppose every source graph has an average of $N$ nodes, $E$ edges, with $d_0$ initial feature dimension, the overall computational complexity is simplified to ($d^{\mathcal{S}} = d^{\mathcal{T}} = d$),

$$\mathcal{O}(n(E(d + d_h) + Nd_h^2)) + \mathcal{O}(nNd(d_0 + d + n_1 + n_2), \tag{B.2}$$

where the additional complexity beyond that of a regular two-layer GCN does not lead to a perceptible increase in pre-training time, as $n, n_1, n_2 \ll N$ in most cases. Such that, the pre-training computational complexity of BRIDGE is on par with other state-of-the-art graph multi-domain pre-training models.

---

**Algorithm 2** Down-prompting (fine-tuning) pipeline of BRIDGE.

---

**Input:** $m$ nodes class-balanced sampled from a target graphs $\{G^{\mathcal{T}}\} \in \mathcal{G}^{\mathcal{T}}$ (node classification)
or $m$ target graphs $\{G^{\mathcal{T}}\} \in \mathcal{G}^{\mathcal{T}}$ from target domain $\{D^{\mathcal{T}}\} \in \mathcal{D}^{\mathcal{T}}$ (graph classification),
where each of the graph $G_i$ is associate with feature matrix $\mathbf{X}_i \in \mathbb{R}^{N_i \times d_i}$ and adjacency
matrix $\mathbf{A} \in \mathbb{R}^{N_i \times N_i}$; The pre-trained graph encoder and discriminator $g(f_{\hat{\Theta}}^{\star}(\cdot))$; Domain-
invariant aligners $\mathcal{A}_d$ and $\mathcal{A}_s$; Routing network $\phi(\cdot)$; Assignment weight vector $\mathbf{S}$; Tuna-
ble graph prompts $\mathcal{P}_{\Omega}$; Number of pre-training epochs $E_2$; Temperature parameter $\tau$;
Hyperparameter $\beta$ and $\gamma$ .

**Output:** Fine-tuned graph encoder and discriminator $g(f_{\hat{\Theta}}^{\star}(\cdot))$.

Initialize all parameters randomly;

**for** $e = 1, 2, \cdots, E_2$ **do**

    `# Dimension-wise feature alignment`
    Aligned graph feature: $\widehat{\mathbf{X}}_i^{\mathcal{T}} \leftarrow$ Eq. (6) for each $G_i \in \{G^{\mathcal{T}}\}$;

    `# Semantic-wise feature alignment`
    Initialize the assignment weight vector: $\mathbf{S} = \text{Softmax}(\phi(\mathbf{X}^{\mathcal{T}}))$;
    Assemble source domain aligners (experts) to facilitate semantic alignment: $\widehat{\mathbf{X}}_i^{\mathcal{T}\prime} \leftarrow$ Eq. (7);
    Optimize the MoE network: $\mathcal{L}_{\text{MoE}}(\mathcal{A}_s; \mathbf{S}) \leftarrow$ Eq. (8);

    `# Down-prompting classification objective`
    Encode target domain node or graph embeddings with graph prompts: $\mathbf{H}_i^{\mathcal{T}} \leftarrow$ Eq. (9);
    Calculate down-prompting classification loss: $\mathcal{L}_{\text{cls}}(\Omega; \mathbf{H}^{\mathcal{T}}) \leftarrow$ Eq. (10);

    `# Generalization bound as spectral regularizer`
    Calculate the spectral regularize: $\mathcal{R}_{\text{spec}}(\Omega; \widehat{\mathbf{X}}^{\mathcal{T}}; \mathbf{H}^{\mathcal{T}}) \leftarrow$ Eq. (20);

    `# Optimization`
    Calculate overall loss: $\lceil \mathcal{L}_{\text{down}} \rceil \leftarrow \mathcal{L}_{\text{cls}}(\Omega; \mathbf{H}^{\mathcal{T}}) + \beta \mathcal{L}_{\text{MoE}}(\mathcal{A}_s; \mathbf{S}) + \gamma \mathcal{R}_{\text{spec}}(\Omega; \widehat{\mathbf{X}}^{\mathcal{T}}; \mathbf{H}^{\mathcal{T}})$;
    Update $\Omega$ by minimizing $\lceil \mathcal{L}_{\text{down}} \rceil$ and back-propagation.
**end for**

---

**Complexity Analysis of Algorithm 2.** We analyze the computational complexity of each part in Algorithm 2 as follows.

- **Dimension-wise feature alignment:** We apply the same $\mathcal{A}_d$ in pre-training, which is implemented by a two-layer MLP ($\mathbb{R}^{d_i} \mapsto \mathbb{R}^{d^{\mathcal{T}}}$). Its computational complexity takes $\mathcal{O}(N_i d_i d^{\mathcal{T}})$ for node classification and $\mathcal{O}(\sum_{i=1}^{m} N_i d_i d^{\mathcal{T}})$ for graph classification, where $N_i$ represents node numbers in target graph $G_i$, and $d_i$ is its initial feature dimension.

- **Routing network initialization:** The learnable routing network $\phi(\cdot)$ is implemented by a single-layer MLP, which is initialized by $\mathbf{X}^{\mathcal{T}}$. The initialization takes the computation complexity of $\mathcal{O}(N_i d^{\mathcal{T}} n)$ for node classification and $\mathcal{O}(\sum_{i=1}^{m} N_i d^{\mathcal{T}} n)$ for graph classification.

- **Selective knowledge assembly:** The assignment weight vector takes the $\text{Softmax}$ operation on output of $\phi(\cdot)$, and then assemble $\mathcal{A}_s^{\top}$ from source domains to guide dimension selection on $\widehat{\mathbf{X}}^{\mathcal{T}}$, which takes the computational complexity of $\mathcal{O}(nmd^{\mathcal{S}}d^{\mathcal{T}})$ for node classification and $\mathcal{O}(nd^{\mathcal{S}}d^{\mathcal{T}}\sum_{i=1}^{m} N_i)$ for graph classification.

- **MoE optimization by uncertainty:** We applied the self-supervised optimization by minimizing the uncertainty of the assignment weight vector, where the uncertainty calculation takes the computational complexity of $\mathcal{O}(n)$ for both node and graph classification.

- **Graph prompt insertion:** We initialize graph prompt with learnable vector $\Omega$ over the aligned features $\widehat{\mathbf{X}}_i^{\mathcal{T}\prime}$, which takes the computational complexity of $\mathcal{O}(N_i d^{\mathcal{T}})$ for node classification and $\mathcal{O}(\sum_{i=1}^{m} N_i d^{\mathcal{T}})$ for graph classification.

- **Feature encoding:** We encode target graph(s) by the pre-trained two-layer GNN, which takes the complexity of $\mathcal{O}(|\mathcal{E}_i|(d^{\mathcal{T}} + d_h) + N_i d_h^2)$ for node classification and $\mathcal{O}(\sum_{i=1}^{m}(|\mathcal{E}_i|(d^{\mathcal{T}} + d_h) + N_i d_h^2))$ for graph classification, where $|\mathcal{E}_i|$ is the number of edges, $d^{\mathcal{T}}$ is the aligned feature dimension, and $h$ is the hidden dimension.

- **Down-prompting classification loss:** Suppose there are $n_c$ classes, the contrastive loss takes the computational complexity of $\mathcal{O}(mn_c d^{\mathcal{T}})$ for both node and graph classification.

- **Spectral regularizer:** Extracting spectral signals with eigenvalue decomposition (EVD) requires $\mathcal{O}(N^3)$ for both node and graph classification.

The overall computational complexity for the down-prompting (fine-tuning) phase of node classification takes,

$$\mathcal{O}(N_i d_i d^{\mathcal{T}}) + \mathcal{O}(N_i d^{\mathcal{T}} n) + \mathcal{O}(nmd^{\mathcal{S}} d^{\mathcal{T}}) + \mathcal{O}(n) + \mathcal{O}(N_i d^{\mathcal{T}}) + \mathcal{O}(|\mathcal{E}_i|(d^{\mathcal{T}} + d_h) + N_i d_h^2) + \mathcal{O}(mn_c d^{\mathcal{T}}) + \mathcal{O}(mN_i^3). \quad \text{(B.3)}$$

Suppose every target graph has an average of $N$ nodes, $E$ edges, with $d_0$ initial feature dimension, the overall computational complexity of node classification is simplified to ($d^{\mathcal{S}} = d^{\mathcal{T}} = d$),

$$\mathcal{O}(E(d + d_h) + Nd_h^2) + \mathcal{O}(Nd(d_0 + n)) + \mathcal{O}(m(nd^3 + n_c d + N^3)) + \mathcal{O}(n). \quad \text{(B.4)}$$

As $m$, $n$, $n_c$, $d$, $d_0$ are all small constants compared with $N$, beyond that of a regular two-layer GCN, the complexity primarily increases due to the $\mathcal{O}(N^3)$ complexity required for eigenvalue decomposition.

The overall computational complexity for the down-prompting (fine-tuning) phase of graph classification takes,

$$\mathcal{O}\Big(\sum_{i=1}^{m} N_i d_i d^{\mathcal{T}}\Big) + \mathcal{O}\Big(\sum_{i=1}^{m} N_i d^{\mathcal{T}} n\Big) + \mathcal{O}\Big(nd^{\mathcal{S}} d^{\mathcal{T}} \sum_{i=1}^{m} N_i\Big) + \mathcal{O}(n) + \mathcal{O}\Big(\sum_{i=1}^{m} N_i d^{\mathcal{T}}\Big)$$
$$+ \mathcal{O}\Big(\sum_{i=1}^{m}(|\mathcal{E}_i|(d^{\mathcal{T}} + d_h) + N_i d_h^2)\Big) + \mathcal{O}(mn_c d^{\mathcal{T}}) + \mathcal{O}(N^3). \quad \text{(B.5)}$$

Similarly, the overall computational complexity of graph classification can also be simplified to ($d^{\mathcal{S}} = d^{\mathcal{T}} = d$),

$$\mathcal{O}(m(E(d + d_h) + Nd_h^2)) + \mathcal{O}(mNd(d_0 + n + nd)) + \mathcal{O}(mdn_c) + \mathcal{O}(n) + \mathcal{O}(N^3). \quad \text{(B.6)}$$

In the same way, since $m$, $n$, $n_c$, $d$, and $d_0$ are all small constants relative to $N$, the additional complexity compared to a standard two-layer GCN is mainly attributed to the $\mathcal{O}(N^3)$ complexity introduced by the eigenvalue decomposition.

To mitigate the $\mathcal{O}(N^3)$ complexity of eigenvalue decomposition, several effective strategies are implemented (You et al., 2023). Mini-batch training is utilized to reduce computational overhead by processing only a small subset of nodes in each batch rather than the entire graph, ensuring that each batch is completed in tolerable time (Hamilton et al., 2017; Zeng et al., 2020). Additionally, when EVD is indispensable, it is performed only once for the entire graph, and the results are reused across subsequent training steps to avoid unnecessary computations. These optimizations keep the computational burden manageable, as efficiency experiments demonstrate that the total down-prompting time remains in line with other competitive baselines, striking a balance between computational efficiency and scalability for large-scale graphs.

It is worth noting that we temporarily set aside the theoretical analysis of space complexity and instead focus on evaluating the actual computational time and space requirements through efficiency analysis experiments. Experimental results indicate that, under the hardware and software configurations detailed in Appendix F.3, the pre-training time and space complexity of our proposed BRIDGE are comparable to those of other competitive baselines. While the fine-tuning phase incurs slightly higher time and space complexity than other baselines, this can be effectively mitigated by adopting optimized training strategies to achieve a balance between performance and efficiency.

In summary, the computational complexity of the BRIDGE framework consists of two phases: pre-training and fine-tuning. During the pre-training phase, the overall complexity is on par with other state-of-the-art pre-training frameworks, as the additional computations for feature alignment and risk regularization remain manageable. In the fine-tuning phase, although both node classification and graph classification are influenced by the $\mathcal{O}(N^3)$ complexity introduced by eigenvalue decomposition (EVD), the total down-prompting time remains in line with other competitive baselines, striking a balance between computational efficiency and scalability for large-scale graphs.

# C. Proofs

## C.1. Proof of Proposition 4.2: Achievable Assumption

We first restate Assumption 4.1 and Proposition 4.2 for reference.

---

**Assumption 4.1** (**Domain Invariance**). *For each source domain $D_i \in \{D^S\}$, there exist a domain-specific invariant aligner $\mathcal{A}_s^{\mathcal{I}}$ and variant aligner $\mathcal{A}_s^{\mathcal{V}}$ that lead to generalized graph representation under any target domains. Domain-invariant features $\widehat{\mathbf{X}}_i^{\mathcal{I}} = \mathcal{A}_s^{\mathcal{I}} \odot \widehat{\mathbf{X}}_i^S$ should satisfy,*

- ***Invariance Property:** for any target domain $D_j \in \{D^S\} \cup \{D^T\}$, the equality $\mathbb{P}(Y_i \mid \widehat{\mathbf{X}}_i^{\mathcal{I}}, D_j) = \mathbb{P}(Y_i \mid \widehat{\mathbf{X}}_i^{\mathcal{I}})$ holds.*

- ***Sufficient Condition:** given any independent and random noise $\epsilon$, $Y_i = g(f_{\boldsymbol{\Theta}}(\widehat{\mathbf{X}}_i^{\mathcal{I}}, \mathbf{A}_i)) + \epsilon$, i.e., $Y_i \perp \mathcal{A}_s^{\mathcal{V}} \mid \mathcal{A}_s^{\mathcal{I}}$, where $f_{\boldsymbol{\Theta}}(\cdot)$ is the pre-trained graph encoder, $g(\cdot)$ is a discriminator. The logic structure syntaxes $\perp$ and $\mid$ represents label $Y_i$ is independent of $\mathcal{A}_s^{\mathcal{V}}$ but rely on $\mathcal{A}_s^{\mathcal{I}}$.*

---

**Proposition 4.2** (**Achievable Assumption**). *Minimizing Eq. (4) encourages pre-trained $f_{\boldsymbol{\Theta}}^{\star}$ to satisfy Assumption 4.1.*

---

*Proof.* We begin by introducing the following lemma to reformulate the Invariance Property and the Sufficient Condition in Assumption 4.1 via the mutual information theory (Kraskov et al., 2004).

---

**Lemma C.1** (**Mutual Information Equivalence**). *The Mutual Information $I(\cdot; \cdot)$ provides an equivalent representation of the Invariance Property and the Sufficient Condition in Assumption 4.1, which takes the following form,*
- ***Invariance Property:** $\mathbb{P}(Y_i \mid \widehat{\mathbf{X}}_i^{\mathcal{I}}, D_j) = \mathbb{P}(Y_i \mid \widehat{\mathbf{X}}_i^{\mathcal{I}}) \Leftrightarrow I(Y_i; D_j \mid \widehat{\mathbf{X}}_i^{\mathcal{I}}) = 0;$*
- ***Sufficient Condition:** $Y_i = g(f_{\boldsymbol{\Theta}}(\widehat{\mathbf{X}}_i^{\mathcal{I}}, \mathbf{A}_i)) + \epsilon \Leftrightarrow I(Y_i; \widehat{\mathbf{X}}_i^{\mathcal{I}})$ is maximized.*

---

*Proof.* We establish Lemma C.1 by independently proving the validity of both the Invariance Property and the Sufficient Condition (Yuan et al., 2023).

**(1) Proofs of Invariance Property.** Given three random variables $\mathbf{x}$, $\mathbf{y}$ and $\mathbf{z}$, the mutual information between $\mathbf{x}$ and $\mathbf{y}$, conditioned on $\mathbf{z}$, is defined by,

$$I(\mathbf{x}; \mathbf{y}|\mathbf{z}) = \mathbb{E}_{\mathbb{P}(\mathbf{z})} \left[ \mathcal{D}_{\mathrm{KL}}(\mathbb{P}(\mathbf{x}, \mathbf{y}|\mathbf{z}) \parallel \mathbb{P}(\mathbf{x}|\mathbf{z})\mathbb{P}(\mathbf{y}|\mathbf{z})) \right]. \tag{C.1}$$

The joint conditional distribution $\mathbb{P}(\mathbf{x}, \mathbf{y}|\mathbf{z})$ can be written as,

$$\mathbb{P}(\mathbf{x}, \mathbf{y}|\mathbf{z}) = \mathbb{P}(\mathbf{x}|\mathbf{z}, \mathbf{y})\mathbb{P}(\mathbf{y}|\mathbf{z}). \tag{C.2}$$

Substituting Eq. (C.2) into Eq. (C.1),

$$I(\mathbf{x}; \mathbf{y}|\mathbf{z}) = \mathbb{E}_{\mathbb{P}(\mathbf{z})} \left[ \log \frac{\mathbb{P}(\mathbf{x}|\mathbf{z}, \mathbf{y})}{\mathbb{P}(\mathbf{x}|\mathbf{z})} \right]. \tag{C.3}$$

Eq. (C.3) is exactly the Kullback-Leibler divergence (Joyce, 2011) between the distributions $\mathbb{P}(\mathbf{x}|\mathbf{z}, \mathbf{y})$ and $\mathbb{P}(\mathbf{x}|\mathbf{z})$, weighted by $\mathbb{P}(\mathbf{z}, \mathbf{y})$. Hence, we can conclude,

$$I(\mathbf{x}; \mathbf{y}|\mathbf{z}) = \mathcal{D}_{\mathrm{KL}}(\mathbb{P}(\mathbf{x}|\mathbf{z}, \mathbf{y}) \parallel \mathbb{P}(\mathbf{x}|\mathbf{z})). \tag{C.4}$$

Then, substitute $Y_i$ into $\mathbf{x}$, $D_j$ into $\mathbf{y}$, and $\widehat{\mathbf{X}}_i^{\mathcal{I}}$ into $\mathbf{z}$ in Eq (C.4), we can readily obtain the following equation,

$$I(Y_i; D_j \mid \widehat{\mathbf{X}}_i^{\mathcal{I}}) = \mathcal{D}_{\mathrm{KL}}(\mathbb{P}(Y_i \mid \widehat{\mathbf{X}}_i^{\mathcal{I}}, D_j) \parallel \mathbb{P}(Y_i \mid \widehat{\mathbf{X}}_i^{\mathcal{I}})). \tag{C.5}$$

When $\mathbb{P}(Y_i \mid \widehat{\mathbf{X}}_i^{\mathcal{I}}, D_j) = \mathbb{P}(Y_i \mid \widehat{\mathbf{X}}_i^{\mathcal{I}})$, the Kullback-Leibler divergence between the distributions $\mathbb{P}(Y_i \mid \widehat{\mathbf{X}}_i^{\mathcal{I}}, D_j)$ and $\mathbb{P}(Y_i \mid \widehat{\mathbf{X}}_i^{\mathcal{I}})$ equals 0, such that $I(Y_i; D_j \mid \widehat{\mathbf{X}}_i^{\mathcal{I}}) = 0$. Similarly, it is also straightforward to derive the left-hand side from the right-hand side of the equivalent condition of the Invariance Property. This will not be elaborated further here.

**(2) Proofs of Sufficient Condition.** We establish the sufficiency and necessity through the following two steps.

**Step 1:** We first prove for $Y_i$, $\widehat{\mathbf{X}}_i^{\mathcal{I}}$, and $\epsilon$ satisfying $Y_i = g(f_{\boldsymbol{\Theta}}(\widehat{\mathbf{X}}_i^{\mathcal{I}}, \mathbf{A}_i)) + \epsilon$, condition $\widehat{\mathbf{X}}_i^{\mathcal{I}} = \arg\max_{\widehat{\mathbf{X}}_i^{\mathcal{I}}} I(Y_i; \widehat{\mathbf{X}}_i^{\mathcal{I}})$ holds. We proceed by contradiction. Assume that $\widehat{\mathbf{X}}_i^{\mathcal{I}} \neq \arg\max_{\widehat{\mathbf{X}}_i^{\mathcal{I}}} I(Y_i; \widehat{\mathbf{X}}_i^{\mathcal{I}})$, and suppose that there exists another $\widehat{\mathbf{X}}_i^{\mathcal{I}\prime}$ satisfies

$\widehat{\mathbf{X}}_i^{\mathtt{I}\prime} = \arg\max_{\widehat{\mathbf{X}}_i^{\mathtt{I}}} I(Y_i; \widehat{\mathbf{X}}_i^{\mathtt{I}})$, such that $\widehat{\mathbf{X}}_i^{\mathtt{I}\prime} \neq \widehat{\mathbf{X}}_i^{\mathtt{I}}$. We can always construct a permutation-invariant mapping function $\Pi$ such that $\widehat{\mathbf{X}}_i^{\mathtt{I}\prime} = \Pi(\widehat{\mathbf{X}}_i^{\mathtt{I}}, \boldsymbol{\pi})$, where $\boldsymbol{\pi}$ is a random variable. This leads to the following sequence of equalities,

$$I(Y_i; \widehat{\mathbf{X}}_i^{\mathtt{I}\prime}) = I(Y_i; \widehat{\mathbf{X}}_i^{\mathtt{I}}, \boldsymbol{\pi}) = I(g(f_{\boldsymbol{\Theta}}(\widehat{\mathbf{X}}_i^{\mathtt{I}}, \mathbf{A}_i)); \widehat{\mathbf{X}}_i^{\mathtt{I}}, \boldsymbol{\pi}) = I(g(f_{\boldsymbol{\Theta}}(\widehat{\mathbf{X}}_i^{\mathtt{I}}, \mathbf{A}_i)); \widehat{\mathbf{X}}_i^{\mathtt{I}}) = I(Y_i; \widehat{\mathbf{X}}_i^{\mathtt{I}}). \tag{C.6}$$

Since $\widehat{\mathbf{X}}_i^{\mathtt{I}\prime}$ and $\widehat{\mathbf{X}}_i^{\mathtt{I}}$ yield the same mutual information with $Y_i$, this contradicts the assumption that $\widehat{\mathbf{X}}_i^{\mathtt{I}\prime} \neq \widehat{\mathbf{X}}_i^{\mathtt{I}}$. Therefore, the condition $\widehat{\mathbf{X}}_i^{\mathtt{I}} = \arg\max_{\widehat{\mathbf{X}}_i^{\mathtt{I}}} I(Y_i; \widehat{\mathbf{X}}_i^{\mathtt{I}})$ must hold.

**Step 2:** Next, we prove that for $Y_i$, $\widehat{\mathbf{X}}_i^{\mathtt{I}}$, and $\epsilon$ satisfying $\widehat{\mathbf{X}}_i^{\mathtt{I}} = \arg\max_{\widehat{\mathbf{X}}_i^{\mathtt{I}}} I(Y_i; \widehat{\mathbf{X}}_i^{\mathtt{I}})$, the condition $Y_i = g(f_{\boldsymbol{\Theta}}(\widehat{\mathbf{X}}_i^{\mathtt{I}}, \mathbf{A}_i)) + \epsilon$ must also hold. Again, we use proof by contradiction. Assume $Y_i \neq g(f_{\boldsymbol{\Theta}}(\widehat{\mathbf{X}}_i^{\mathtt{I}}, \mathbf{A}_i)) + \epsilon$, and instead suppose $Y_i = g(f_{\boldsymbol{\Theta}}(\widehat{\mathbf{X}}_i^{\mathtt{I}\prime}, \mathbf{A}_i)) + \epsilon$, where $\widehat{\mathbf{X}}_i^{\mathtt{I}\prime} \neq \widehat{\mathbf{X}}_i^{\mathtt{I}}$. Under this assumption, we have the following inequality,

$$I(g(f_{\boldsymbol{\Theta}}(\widehat{\mathbf{X}}_i^{\mathtt{I}\prime}, \mathbf{A}_i)); \widehat{\mathbf{X}}_i^{\mathtt{I}}) \leqslant I(g(f_{\boldsymbol{\Theta}}(\widehat{\mathbf{X}}_i^{\mathtt{I}\prime}, \mathbf{A}_i)); \widehat{\mathbf{X}}_i^{\mathtt{I}\prime}), \tag{C.7}$$

which implies that $\widehat{\mathbf{X}}_i^{\mathtt{I}\prime} = \arg\max_{\widehat{\mathbf{X}}_i^{\mathtt{I}}} I(Y_i; \widehat{\mathbf{X}}_i^{\mathtt{I}})$. This contradicts the original assumption $\widehat{\mathbf{X}}_i^{\mathtt{I}} = \arg\max_{\widehat{\mathbf{X}}_i^{\mathtt{I}}} I(Y_i; \widehat{\mathbf{X}}_i^{\mathtt{I}})$. Hence, we conclude that $Y_i = g(f_{\boldsymbol{\Theta}}(\widehat{\mathbf{X}}_i^{\mathtt{I}}, \mathbf{A}_i)) + \epsilon$ must hold. $\qquad\square$

Next, we establish Proposition 4.2 from two perspectives. Step 1 proves minimizing the expectation term in Eq. (4) corresponds to maximizing the lower bound of $I(Y_i; \widehat{\mathbf{X}}_i^{\mathtt{I}})$), thereby encouraging the pre-trained model to satisfy the Sufficient Condition. Step 2 proves minimizing the variance term in Eq. (4) minimizes the upper bound of $I(Y_i; D_j \mid \widehat{\mathbf{X}}_i^{\mathtt{I}})$, which ultimately leads the model satisfies the Invariance Property. The respective proofs are as follows.

**Step 1:** To begin, we show that minimizing the **expectation term** in Eq. (4) ($\mathbb{E}\mathcal{L}_{\mathrm{pre}}(\boldsymbol{\Theta}; \mathbf{H}^{\mathcal{S}})$) ensures that the pre-trained model satisfies the Sufficient Condition stated in Assumption 4.1.

Referring to the Structural Causal Model (SCM) (Pearl & Judea, 2009; Pearl, 2010; Peters et al., 2017), given the dependency relationship $\widehat{\mathbf{X}}_i^{\mathtt{I}} \leftarrow G_i \rightarrow Y_i$, we can analyze that $\max_{\mathbb{Q}(\widehat{\mathbf{X}}_i^{\mathtt{I}} | G_i)} I(Y_i; \widehat{\mathbf{X}}_i^{\mathtt{I}})$ is equivalent to $\min_{\mathbb{Q}(\widehat{\mathbf{X}}_i^{\mathtt{I}} | G_i)} I(Y_i; G_i \mid \widehat{\mathbf{X}}_i^{\mathtt{I}})$, as we remove the spurious correlations embedded in the dependencies between $Y_i$ and $G_i$ by the set of domain-invariant aligners. By considering $\mathbb{Q}(\widehat{\mathbf{X}}_i^{\mathtt{I}} \mid G_i)$ as the variational distribution, the following upper bound can be derived, *i.e.*,

$$I(Y_i; G_i \mid \widehat{\mathbf{X}}_i^{\mathtt{I}}) = \mathcal{D}_{\mathrm{KL}}(\mathbb{P}(Y_i \mid G_i, D_j) \| \mathbb{P}(Y_i \mid \widehat{\mathbf{X}}_i^{\mathtt{I}}, D_j)) \tag{C.8}$$

$$= \mathcal{D}_{\mathrm{KL}}(\mathbb{P}(Y_i \mid G_i, D_j) \| \mathbb{P}(Y_i \mid \widehat{\mathbf{X}}_i^{\mathtt{I}})) - \mathcal{D}_{\mathrm{KL}}(\mathbb{P}(Y_i \mid \widehat{\mathbf{X}}_i^{\mathtt{I}}, D_j) \| \mathbb{Q}(Y_i \mid \widehat{\mathbf{X}}_i^{\mathtt{I}})) \tag{C.9}$$

$$\leqslant \mathcal{D}_{\mathrm{KL}}(\mathbb{P}(Y_i \mid G_i, D_j) \| \mathbb{Q}(Y_i \mid \widehat{\mathbf{X}}_i^{\mathtt{I}})) \tag{C.10}$$

$$\leqslant \min_{\mathbb{Q}(Y_i | \widehat{\mathbf{X}}_i^{\mathtt{I}})} \mathcal{D}_{\mathrm{KL}}(\mathbb{P}(Y_i \mid G_i, D_j) \| \mathbb{Q}(Y_i \mid \widehat{\mathbf{X}}_i^{\mathtt{I}})). \tag{C.11}$$

Additionally, the following expression holds,

$$\mathcal{D}_{\mathrm{KL}}(\mathbb{P}(Y_i \mid G_i, D_j) \| \mathbb{Q}(Y_i \mid \widehat{\mathbf{X}}_i^{\mathtt{I}})) = \mathbb{E}_{D_j}\mathbb{E}_{(G_i, Y_i)\sim\mathbb{P}(\mathcal{G}, \mathcal{Y}|D_j)}\mathbb{E}_{\widehat{\mathbf{X}}_i^{\mathtt{I}}\sim\mathbb{Q}(\widehat{\mathbf{X}}_i^{\mathtt{I}}|G_i)}\left[\log \frac{\mathbb{P}(Y_i \mid G_i, D_j)}{\mathbb{Q}(Y_i \mid \widehat{\mathbf{X}}_i^{\mathtt{I}})}\right] \tag{C.12}$$

$$\leqslant \mathbb{E}_{D_j}\mathbb{E}_{(G_i, Y_i)\sim\mathbb{P}(\mathcal{G}, \mathcal{Y}|D_j)}\left[\log \frac{\mathbb{P}(Y_i \mid G_i, D_j)}{\mathbb{E}_{\widehat{\mathbf{X}}_i^{\mathtt{I}}\sim\mathbb{Q}(\widehat{\mathbf{X}}_i^{\mathtt{I}}|G_i)}\mathbb{Q}(Y_i \mid \widehat{\mathbf{X}}_i^{\mathtt{I}})}\right]. \tag{C.13}$$

Here, Eq. (C.13) relies on Jensen's Inequality (Jensen, 1906; Wu et al., 2022), and requires the GNN encoder $f_{\boldsymbol{\Theta}}(\cdot)$ is a Dirac delta distribution ($\delta$-distribution). Therefore, we reach,

$$\min_{\mathbb{Q}(Y_i|\widehat{\mathbf{X}}_i^{\mathtt{I}})} \mathcal{D}_{\mathrm{KL}}(\mathbb{P}(Y_i \mid G_i, D_j) \| \mathbb{Q}(Y_i \mid \widehat{\mathbf{X}}_i^{\mathtt{I}})) \Leftrightarrow \min_{\boldsymbol{\Theta}} \mathbb{E}_{D_j\sim\mathbb{Q}(\{D^{\mathcal{S}}\}),(G_i, Y_i)\sim\mathbb{P}(\mathcal{G}, \mathcal{Y}|D_j)}\mathcal{L}_{\mathrm{pre}}(\boldsymbol{\Theta}; \mathbf{H}^{\mathcal{S}}). \tag{C.14}$$

Thus, we have demonstrated that minimizing the expectation term in Eq. (4) corresponds to minimizing the upper bound of $I(Y_i; G_i \mid \widehat{\mathbf{X}}_i^{\mathtt{I}})$ (or equivalently maximizing the lower bound of $I(Y_i; \widehat{\mathbf{X}}_i^{\mathtt{I}})$), thereby encouraging the pre-trained model to satisfy the Sufficient Condition.

**Step 2:** We now demonstrate that minimizing the **variance term** in Eq. (4) ($\mathbb{V}([\mathbb{E}\mathcal{L}_{\text{pre}}(\boldsymbol{\Theta}; \widetilde{\mathbf{H}}_i^{\mathcal{S}})]_{i=1}^K)$) guides the model to satisfy the Invariance Property in Assumption 4.1.

By considering $\mathbb{Q}(\widehat{\mathbf{X}}_i^{\mathcal{I}} \mid G_i)$ as the variational distribution, the following upper bound can be derived, *i.e.*,

$$I(Y_i; D_j \mid \widehat{\mathbf{X}}_i^{\mathcal{I}}) = \mathcal{D}_{\text{KL}}(\mathbb{P}(Y_i \mid \widehat{\mathbf{X}}_i^{\mathcal{I}}, D_j) \parallel \mathbb{P}(Y_i \mid \widehat{\mathbf{X}}_i^{\mathcal{I}})) \tag{C.15}$$

$$= \mathcal{D}_{\text{KL}}(\mathbb{P}(Y_i \mid \widehat{\mathbf{X}}_i^{\mathcal{I}}, D_j) \parallel \mathbb{E}_{D_j}\mathbb{P}(Y_i \mid \widehat{\mathbf{X}}_i^{\mathcal{I}}, D_j)) \tag{C.16}$$

$$= \mathcal{D}_{\text{KL}}(\mathbb{Q}(Y_i \mid \widehat{\mathbf{X}}_i^{\mathcal{I}}) \parallel \mathbb{E}_{D_j}\mathbb{Q}(Y_i \mid \widehat{\mathbf{X}}_i^{\mathcal{I}})) - \mathcal{D}_{\text{KL}}(\mathbb{Q}(Y_i \mid \widehat{\mathbf{X}}_i^{\mathcal{I}}) \parallel \mathbb{P}(Y_i \mid \widehat{\mathbf{X}}_i^{\mathcal{I}}, D_j)) \tag{C.17}$$

$$- \mathcal{D}_{\text{KL}}(\mathbb{E}_{D_j}\mathbb{P}(Y_i \mid \widehat{\mathbf{X}}_i^{\mathcal{I}}, D_j) \parallel \mathbb{E}_{D_j}\mathbb{Q}(Y_i \mid \widehat{\mathbf{X}}_i^{\mathcal{I}})) \tag{C.18}$$

$$\leqslant \mathcal{D}_{\text{KL}}(\mathbb{Q}(Y_i \mid \widehat{\mathbf{X}}_i^{\mathcal{I}}) \parallel \mathbb{E}_{D_j}\mathbb{Q}(Y_i \mid \widehat{\mathbf{X}}_i^{\mathcal{I}})) \tag{C.19}$$

$$\leqslant \min_{\mathbb{Q}(Y_i|\widehat{\mathbf{X}}_i^{\mathcal{I}})} \mathcal{D}_{\text{KL}}(\mathbb{Q}(Y_i \mid \widehat{\mathbf{X}}_i^{\mathcal{I}}) \parallel \mathbb{E}_{D_j}\mathbb{Q}(Y_i \mid \widehat{\mathbf{X}}_i^{\mathcal{I}})). \tag{C.20}$$

Additionally, we have,

$$\mathcal{D}_{\text{KL}}(\mathbb{Q}(Y_i \mid \widehat{\mathbf{X}}_i^{\mathcal{I}}) \parallel \mathbb{E}_{D_j}\mathbb{Q}(Y_i \mid \widehat{\mathbf{X}}_i^{\mathcal{I}})) = \mathbb{E}_{D_j}\mathbb{E}_{(G_i,Y_i)\sim\mathbb{P}(\mathcal{G},\mathcal{Y}|D_j)}\mathbb{E}_{\widehat{\mathbf{X}}_i^{\mathcal{I}}\sim\mathbb{Q}(\widehat{\mathbf{X}}_i^{\mathcal{I}}|G_i)}\left[\log\frac{\mathbb{Q}(Y_i \mid \widehat{\mathbf{X}}_i^{\mathcal{I}})}{\mathbb{E}_{D_j}\mathbb{Q}(Y_i \mid D_j)}\right]. \tag{C.21}$$

Derived from Jensen's Inequality (Jensen, 1906; Wu et al., 2022), we have,

$$\mathcal{D}_{\text{KL}}(\mathbb{Q}(Y_i \mid \widehat{\mathbf{X}}_i^{\mathcal{I}}) \parallel \mathbb{E}_{D_j}\mathbb{Q}(Y_i \mid \widehat{\mathbf{X}}_i^{\mathcal{I}})) \leqslant \mathbb{E}_{D_j}\left[\left|\mathcal{L}_{\text{pre}}(\boldsymbol{\Theta}; \mathbf{H}^{\mathcal{S}}) - \mathbb{E}_{D_j}\mathcal{L}_{\text{pre}}(\boldsymbol{\Theta}; \widetilde{\mathbf{H}}^{\mathcal{S}})\right|\right], \tag{C.22}$$

where $\widetilde{\mathbf{H}}$ denotes embeddings bound with misaligned features under any domain $D_j \in \{D^S\}$. Finally, minimizing,

$$\min_{\mathbb{Q}(Y_i|\widehat{\mathbf{X}}_i^{\mathcal{I}})} \mathcal{D}_{\text{KL}}(\mathbb{Q}(Y_i \mid \widehat{\mathbf{X}}_i^{\mathcal{I}}) \parallel \mathbb{E}_{D_j}\mathbb{Q}(Y_i \mid \widehat{\mathbf{X}}_i^{\mathcal{I}})) \Leftrightarrow \mathbb{V}\left[\mathbb{E}_{D_j\sim\mathbb{Q}(\{D^S\}),(G_i,Y_i)\sim\mathbb{P}(\mathcal{G},\mathcal{Y}|D_j)}\mathbb{E}\mathcal{L}_{\text{pre}}(\boldsymbol{\Theta}; \widetilde{\mathbf{H}}_i^{\mathcal{S}})\right]_{i=1}^K. \tag{C.23}$$

Thus, we have proven that minimizing the variance term in Eq. (4) minimizes the upper bound of $I(Y_i; D_j \mid \widehat{\mathbf{X}}_i^{\mathcal{I}})$, which ultimately leads the model satisfies the Invariance Property.

We conclude the proof for Proposition 4.2. $\qquad\square$

Proposition 4.2 avoids making overly stringent assumptions regarding the Sufficient Condition and Invariance Property.

### C.2. Proof of Proposition 4.3: Equivalent Optimization

We first restate Eq. (3), Eq. (4), and Proposition 4.3 for reference.

**Eq. (3):**
$$\min_{\boldsymbol{\Theta}} \max_{D\in\{D^S\}\cup\{D^{\mathcal{T}}\}} \mathbb{E}_{(G,Y)\sim\mathbb{P}(\mathcal{G},\mathcal{Y}|D)}\left[\mathcal{L}_{\text{pre}}(\boldsymbol{\Theta}; \mathbf{H}^{\mathcal{S}})|D\right].$$

**Eq. (4):**
$$\min_{\boldsymbol{\Theta}} \mathbb{E}\mathcal{L}_{\text{pre}}(\boldsymbol{\Theta}; \mathbf{H}^{\mathcal{S}}) + \alpha\mathbb{V}([\mathbb{E}\mathcal{L}_{\text{pre}}(\boldsymbol{\Theta}; \widetilde{\mathbf{H}}_k^{\mathcal{S}})]_{k=1}^K), \quad \text{where } \widetilde{\mathbf{H}}_k^{\mathcal{S}} = f_{\boldsymbol{\Theta}}(\mathcal{A}_{\text{s}}^{\mathcal{I}} \odot \widehat{\mathbf{X}}^{\mathcal{S}} + (\mathcal{A}_{\text{s}}^{\mathcal{V}} \odot \widehat{\mathbf{X}}^{\mathcal{S}})\mathbf{W}_k, \mathbf{A}).$$

**Proposition 4.3** (**Equivalent Optimization**). *Optimizing Eq. (4) is equivalent to minimizing the upper bound of multi-domain semantic alignment risk in Eq. (3).*

*Proof.* As stated in Eq. (3), the intrinsic generation process of graph $(G_i, Y_i)$ from any domain $D_j \in \{D^S\}$ is determined by the distribution $\mathbb{P}(\mathcal{G}, \mathcal{Y} \mid D_j)$, where the semantic misalignment difference in $D_j \sim \mathbb{P}(\mathcal{D})$ between domains introduces

graph pre-training variations. Let $\mathbb{Q}(Y_i \mid G_i)$ denote the inferred variational distribution of the true posterior $\mathbb{Q}(Y_i \mid G_i, D_j)$, then the multi-domain semantic alignment risk can be expressed as the KL divergence between these two distributions, *i.e.*,

$$\mathcal{D}_{\text{KL}}(\mathbb{P}(Y_i \mid G_i, D_j) \parallel \mathbb{Q}(Y_i \mid G_i)) = \mathbb{E}_{D_j}\mathbb{E}_{(G_i,Y_i)\sim\mathbb{P}(\mathcal{G},\mathcal{Y}|D_j)}\left[\log \frac{\mathbb{P}(Y_i \mid G_i, D_j)}{\mathbb{Q}(Y_i \mid G_i)}\right] \quad (C.24)$$

Following Federici et al. (2021); Wu et al. (2022), we adopt an information-theoretic perspective to analyze this problem. To simplify the multi-domain semantic alignment risk in Eq. (C.24), we propose the following lemma.

**Lemma C.2** (**Multi-Domain Semantic Alignment Risk**). *The multi-domain semantic alignment risk satisfies the following upper bound,*

$$\mathcal{D}_{KL}(\mathbb{P}(Y_i \mid G_i, D_j) \parallel \mathbb{Q}(Y_i \mid G_i)) \leqslant \mathcal{D}_{KL}(\mathbb{P}(Y_i \mid G_i, D_j) \parallel \mathbb{Q}(Y_i \mid \widehat{\mathbf{X}}_i^{\mathcal{I}})), \quad (C.25)$$

*where can be interpreted as the variational distribution for the label predictor (if two nodes are connected in Eq. (2)).*

*Proof.* We start from Eq. (C.24).

$$\mathcal{D}_{\text{KL}}(\mathbb{P}(Y_i \mid G_i, D_j) \parallel \mathbb{Q}(Y_i \mid G_i)) = \mathbb{E}_{D_j}\mathbb{E}_{(G_i,Y_i)\sim\mathbb{P}(\mathcal{G},\mathcal{Y}|D_j)}\left[\log \frac{\mathbb{P}(Y_i \mid G_i, D_j)}{\mathbb{Q}(Y_i \mid G_i)}\right] \text{ (Eq. (C.24))} \quad (C.26)$$

$$= \mathbb{E}_{D_j}\mathbb{E}_{(G_i,Y_i)\sim\mathbb{P}(\mathcal{G},\mathcal{Y}|D_j)}\left[\log \frac{\mathbb{P}(Y_i \mid G_i, D_j)}{\mathbb{E}_{\widehat{\mathbf{X}}_i^{\mathcal{I}}\sim\mathbb{Q}(\widehat{\mathbf{X}}_i^{\mathcal{I}}|G_i)}\mathbb{Q}(Y_i \mid \widehat{\mathbf{X}}_i^{\mathcal{I}})}\right] \quad (C.27)$$

$$\leqslant \mathbb{E}_{D_j}\mathbb{E}_{(G_i,Y_i)\sim\mathbb{P}(\mathcal{G},\mathcal{Y}|D_j)}\mathbb{E}_{\widehat{\mathbf{X}}_i^{\mathcal{I}}\sim\mathbb{Q}(\widehat{\mathbf{X}}_i^{\mathcal{I}}|G_i)}\left[\log \frac{\mathbb{P}(Y_i \mid G_i, D_j)}{\mathbb{Q}(Y_i \mid \widehat{\mathbf{X}}_i^{\mathcal{I}})}\right] \quad (C.28)$$

$$= \mathcal{D}_{\text{KL}}(\mathbb{P}(Y_i \mid G_i, D_j) \parallel \mathbb{Q}(Y_i \mid \widehat{\mathbf{X}}_i^{\mathcal{I}})) \text{ (Jensen's Inequality)}, \quad (C.29)$$

where the upper bound (Eq. (C.29)) emerges from the assumption that $\mathbb{Q}(\widehat{\mathbf{X}}_i^{\mathcal{I}} \mid G_i)$ represents a deterministic distribution (*i.e.*, a Dirac delta function), as the GNN encoder $f_{\boldsymbol{\Theta}}(\cdot)$ holds the assumption. This completes the proof for Lemma C.2. □

From Lemma C.1, we can reframe Eq. (4) as,

$$\min_{\mathbb{Q}(\widehat{\mathbf{X}}_i^{\mathcal{I}}|G_i),\mathbb{Q}(Y_i,\widehat{\mathbf{X}}_i^{\mathcal{I}})} \mathcal{D}_{\text{KL}}(\mathbb{P}(Y_i \mid G_i, D_j) \parallel \mathbb{Q}(Y_i \mid \widehat{\mathbf{X}}_i^{\mathcal{I}})) + I(Y_i; D_j \mid \widehat{\mathbf{X}}_i^{\mathcal{I}}). \quad (C.30)$$

Consequently, using Lemma C.2, minimizing Eq. (4) reduces to minimizing the upper bound of multi-domain semantic alignment risk in Eq. (3), *i.e.*,

$$\min_{\boldsymbol{\Theta}} \mathbb{E}\mathcal{L}_{\text{pre}}(\boldsymbol{\Theta}; \mathbf{H}^{\mathcal{S}}) + \alpha\mathbb{V}([\mathbb{E}\mathcal{L}_{\text{pre}}(\boldsymbol{\Theta}; \widetilde{\mathbf{H}}_k^{\mathcal{S}})]_{k=1}^K) \Leftrightarrow \min_{\mathbb{Q}(\widehat{\mathbf{X}}_i^{\mathcal{I}}|G_i),\mathbb{Q}(Y_i,\widehat{\mathbf{X}}_i^{\mathcal{I}})} \mathcal{D}_{\text{KL}}(\mathbb{P}(Y_i \mid G_i, D_j)\|\mathbb{Q}(Y_i \mid \widehat{\mathbf{X}}_i^{\mathcal{I}})) + \underbrace{I(Y_i; D_j \mid \widehat{\mathbf{X}}_i^{\mathcal{I}})}_{\text{non-negative}}$$

$$(C.31)$$

$$\geqslant \min_{\mathbb{Q}(\widehat{\mathbf{X}}_i^{\mathcal{I}}|G_i),\mathbb{Q}(Y_i,\widehat{\mathbf{X}}_i^{\mathcal{I}})} \mathcal{D}_{\text{KL}}(\mathbb{P}(Y_i \mid G_i, D_j)\|\mathbb{Q}(Y_i \mid \widehat{\mathbf{X}}_i^{\mathcal{I}})) \quad (C.32)$$

$$\geqslant \min_{\mathbb{Q}(\widehat{\mathbf{X}}_i^{\mathcal{I}}|G_i),\mathbb{Q}(Y_i,\widehat{\mathbf{X}}_i^{\mathcal{I}})} \mathcal{D}_{\text{KL}}(\mathbb{P}(Y_i \mid G_i, D_j)\|\mathbb{Q}(Y_i \mid G_i). \quad (C.33)$$

It is evident that as $K$ increases (to positive infinity), it approaches the theoretical upper bound more closely. A formal proof is omitted for brevity. This concludes the proof for Proposition 4.3. □

Proposition 4.3 declares optimizing the pre-training optimization goal is equivalent to minimizing the upper bound of multi-domain semantic alignment risk in Eq. (3), which is intractable. It also highlights the role that the set of feature aligners play during pre-training, which not only aligns multi-domain graph features dimension-wise and semantic-wise, but also guarantees the learned knowledge is domain-specific and invariant, which enhances the generalization potential and model robustness during the in-context pre-training phase, laying a better and solid foundation for multi-domain knowledge transfer by down-prompting.

## C.3. Proof of Eq. (8): Why Decrease Uncertainty Lead to Optimization?

Let the routing vector be $S = \mathrm{Softmax}(\phi(\mathbf{X}^{\mathcal{T}}))$, where $\phi(\cdot)$ is a learnable routing network initialized by the target feature $\mathbf{X}^{\mathcal{T}}$, and $\mathbf{S} \in \mathbb{R}^n$ represents the assignment weight distribution over $n$ source domain experts (aligners). The uncertainty of $S$ is quantified by its entropy,

$$H(\mathbf{S}) = -\sum_{i=1}^{n} \mathbf{S}_i \log(\mathbf{S}_i), \tag{C.34}$$

where $\mathbf{S}_i$ is the $i$-th component of $\mathbf{S}$, representing the weight of the $i$-th expert. The goal is to minimize the entropy $H(\mathbf{S})$ to reduce the uncertainty of the routing vector.

We answer why decreasing uncertainty leads to optimization from four perspectives.

**(1) The relationship between uncertainty and optimization.** The entropy $H(\mathbf{S})$ measures the uncertainty of the distribution $\mathbf{S}$. High entropy indicates that the distribution is more **uniform**, implying that the model does not prioritize any specific expert. This leads to an averaging effect across experts, which dilutes the influence of the most relevant ones. Conversely, low entropy corresponds to a **sharper** distribution, where the model assigns higher weights to a few critical experts, improving its ability to adapt to target domain features. Thus, minimizing $H(\mathbf{S})$ aligns with the goal of encouraging the model to focus on selecting high-quality experts while ignoring irrelevant ones, thereby improving feature adaptation.

**(2) Lower entropy implies a sharper distribution.** We give the following lemma without proof.

---

**Lemma C.3** (**Relation between Distribution and Entropy**).
*If $\mathbf{S}$ is a uniform distribution (i.e., $\mathbf{S}_i = \frac{1}{n}$ for all i), the entropy reaches its maximum,*

$$H(\mathbf{S}) = -\sum_{i=1}^{n} \frac{1}{n} \log \frac{1}{n} = \log(n). \tag{C.35}$$

*In this case, all components are equally weighted, indicating no preference for any expert. If $\mathbf{S}$ is concentrated on a single component (i.e., $\mathbf{S}_j \to 1$ for some j, and $\mathbf{S}_i \to 0$ for $i \neq j$), the entropy reaches its minimum value of 0,*

$$H(\mathbf{S}) = -(1 \cdot \log 1 + \sum_{i \neq j} 0 \cdot \log 0) = 0. \tag{C.36}$$

*This indicates a fully deterministic selection, where only one expert is prioritized.*

---

By analyzing the behavior of the entropy function, we can quantitatively demonstrate that, by minimizing entropy, the routing vector $\mathbf{S}$ transitions from a uniform distribution to a sharper one, emphasizing a few key experts while suppressing others. This sharpness ensures the model focuses on relevant experts, improving feature adaptation efficiency.

**(3) Entropy minimization as implicit regularization.** Minimizing $H(\mathbf{S})$ introduces an implicit regularization that encourages $\mathbf{S}$ to focus on the most relevant experts. The mathematical intuition behind this process is as follows.

- **Entropy gradient:** The gradient of $H(\mathbf{S})$ encourages larger $\mathbf{S}_i$, thereby concentrating the routing distribution: $\frac{\partial H(\mathbf{S})}{\partial \mathbf{S}_i} = -(1 + \log \mathbf{S}_i)$.
- **Routing distribution convergence:** Minimizing $H(\mathbf{S})$ ensures that $\mathbf{S}_i \to 1$ for a few relevant experts, while $\mathbf{S}_i \to 0$ for irrelevant ones. This leads to effective feature selection, where the model emphasizes only the most critical experts.
- **Explicit objective:** By reducing $\mathcal{L}_{\mathrm{MoE}}$, the model suppresses noise from irrelevant experts, improving feature adaptation to the target domain.

**(4) Self-supervised mechanism.** Minimizing $H(\mathbf{S})$ operates as a self-supervised mechanism. As the routing vector $\mathbf{S}$ is derived entirely from the target feature $\mathbf{X}^{\mathcal{T}}$, implicitly encoding the relationship between the target domain and source domain experts. This process does not require additional labeled data, relying instead on the internal structure of $S$ to guide optimization. As a result, the model automatically learns to select the most relevant experts, achieving effective domain adaptation without external supervision.

In summary, minimizing the uncertainty of the routing vector $\mathbf{S}$ by reducing its entropy $H(\mathbf{S})$ leads to optimization because it concentrates the distribution on a few key experts, reducing noise and improving feature adaptation. Lower entropy implies a sharper distribution, which aligns the model's focus on the most relevant source domain experts. This self-supervised mechanism ensures efficient domain adaptation and enhanced performance in downstream tasks.

## C.4. Proof of Proposition 4.5: Generalization Error Upper Bound

We first restate Proposition 4.5 for reference.

---

**Proposition 4.5** (**Generalization Error Upper Bound** (Redko et al., 2017; Li et al., 2021; You et al., 2023)) *Let $\mathcal{H}$ = $\{h : \mathcal{G} \mapsto \mathcal{Y} | h = g \circ f\}$ represent the hypothesis space of bounded real-valued functions, where $\mathrm{VCdim}(\mathcal{H}) = d$. Suppose $h^{\mathcal{S}} \neq h^{\mathcal{T}}$ with the Lipschitz constant $L_h$ satisfies,*

$$L_h \triangleq \|h\|_{\mathsf{Lip}} = \sup_{G_i, G_j} \frac{\|h^D(G_i) - h^D(G_j)\|_2}{d_W(G_i, G_j)} \leqslant L_f L_g,$$

*where $D \in \{\mathcal{S}, \mathcal{T}\}$. With probability no less than $1 - \delta$, the following inequality holds for target domain error $\epsilon^{\mathcal{T}}(h^{\mathcal{T}})$,*

$$\epsilon^{\mathcal{T}}(h^{\mathcal{T}}) \leqslant {}^{[a]} \frac{m}{n+m} \widehat{\epsilon}^{\mathcal{T}}(h^{\mathcal{T}}) + {}^{[b]} \frac{n}{n+m} \widehat{\epsilon}^{\mathcal{S}}(h^{\mathcal{S}}) + {}^{[c]} 2 L_f L_g d_W\left(\mathbb{P}(G^{\mathcal{S}}), \mathbb{P}(G^{\mathcal{T}})\right) + {}^{[d]} \Delta_h$$

$$+ \mathcal{O}\left[\left(\frac{1}{m} + \frac{1}{n}\right)\log\frac{1}{\delta} + \frac{d}{n}\log\frac{n}{d} + \frac{d}{m}\log\frac{m}{d}\right]^{\frac{1}{2}},$$

*where $\widehat{\epsilon}$ indicates empirical error, and $\Delta_h$ satisfies,*

$$\Delta_h = \min\left(\left|\epsilon^{\mathcal{S}}(h^{\mathcal{S}}) - \epsilon^{\mathcal{S}}(h^{\mathcal{T}})\right|, \left|\epsilon^{\mathcal{T}}(h^{\mathcal{S}}) - \epsilon^{\mathcal{T}}(h^{\mathcal{T}})\right|\right).$$

---

*Proof.* We first introduce the following lemma that introduces Wasserstein distance to relate the source and target domain generalization errors.

---

**Lemma C.4** (**Wasserstein Distance to Relate Errors** (Shen et al., 2018)). *Let $\mu^{\mathcal{S}}$, $\mu^{\mathcal{T}} \in \mathbb{P}(\mathcal{X})$ be two probability measures. Assume the hypotheses $h \in \mathcal{H}$ are all $L_K$-Lipschitz continuous for some $K$. Then the following holds,*

$$\epsilon^{\mathcal{T}}(h, h') \leqslant \epsilon^{\mathcal{S}}(h, h') + 2 L_K d_W(\mu^{\mathcal{S}}, \mu^{\mathcal{T}}), \tag{C.37}$$

*for every hypothesis $h, h' \in \mathcal{H}$.*

---

*Proof.* We first prove that for every $L_K$-Lipschitz continuous hypotheses, $h, h' \in \mathcal{H}$, $|h - h'|$ is $2L_K$-Lipschitz continuous. Using the triangle inequality, we have,

$$|h(x) - h'(x)| \leqslant |h(x) - h(y)| + |h(y) - h'(x)| \tag{C.38}$$

$$\leqslant |h(x) - h(y)| + |h(y) - h'(y)| + |h'(x) - h'(y)|, \tag{C.39}$$

and thus for every $x, y \in \mathcal{X}$,

$$\frac{|h(x) - h'(x)| - |h(y) - h'(y)|}{\rho(x, y)} \leqslant \frac{|h(x) - h(y)| + |h'(x) - h'(y)|}{\rho(x, y)} \tag{C.40}$$

$$\leqslant 2 L_K, \tag{C.41}$$

where $\rho(\cdot, \cdot)$ is some distance measurement. Then, for every hypothesis $h, h'$, we have,

$$\epsilon^{\mathcal{T}}(h, h') - \epsilon^{\mathcal{S}}(h, h') = \mathbb{E}_{\mu^{\mathcal{T}}}\left[|h(x) - h'(x)|\right] - \mathbb{E}_{\mu^{\mathcal{S}}}\left[|h(x) - h'(x)|\right] \tag{C.42}$$

$$\leqslant \sup_{\|f\|_{\mathsf{Lip}} \leqslant 2 L_K} \mathbb{E}_{\mu^{\mathcal{T}}}[f(x)] - \mathbb{E}_{\mu^{\mathcal{S}}}[f(x)] \tag{C.43}$$

$$= 2 L_K d_W(\mu^{\mathcal{S}}, \mu^{\mathcal{T}}). \tag{C.44}$$

$$\square$$

Derived by Lemma C.4 and You et al. (2023) with the $L_f$-Lipschitz continuous assumption,

$$L_h \triangleq \|h\|_{\mathsf{Lip}} = \sup_{G_i, G_j} \frac{\|h^D(G_i) - h^D(G_j)\|_2}{d_W(G_i, G_j)} \leqslant L_f L_g, \tag{C.45}$$

we reach the following inequality,

$$|\epsilon^{\mathcal{S}}(h^{\mathcal{S}}) - \epsilon^{\mathcal{T}}(h^{\mathcal{T}})| = |\epsilon^{\mathcal{S}}(h^{\mathcal{S}}) - \epsilon^{\mathcal{S}}(h^{\mathcal{T}}) + \epsilon^{\mathcal{S}}(h^{\mathcal{T}}) - \epsilon^{\mathcal{T}}(h^{\mathcal{T}})| \tag{C.46}$$

$$\leqslant |\epsilon^{\mathcal{S}}(h^{\mathcal{S}}) - \epsilon^{\mathcal{S}}(h^{\mathcal{T}})| + |\epsilon^{\mathcal{S}}(h^{\mathcal{T}}) - \epsilon^{\mathcal{T}}(h^{\mathcal{T}})| \tag{C.47}$$

$$\leqslant |\epsilon^{\mathcal{S}}(h^{\mathcal{S}}) - \epsilon^{\mathcal{S}}(h^{\mathcal{T}})| + 2L_f L_g d_{\mathrm{W}}(\mathbb{P}(G^{\mathcal{S}}), \mathbb{P}(G^{\mathcal{T}})). \tag{C.48}$$

Similarly,

$$|\epsilon^{\mathcal{S}}(h^{\mathcal{S}}) - \epsilon^{\mathcal{T}}(h^{\mathcal{T}})| \leqslant |\epsilon^{\mathcal{T}}(h^{\mathcal{S}}) - \epsilon^{\mathcal{T}}(h^{\mathcal{T}})| + 2L_f L_g d_{\mathrm{W}}(\mathbb{P}(G^{\mathcal{S}}), \mathbb{P}(G^{\mathcal{T}})). \tag{C.49}$$

We then combine Eq. (C.48) and Eq. (C.49) into,

$$|\epsilon^{\mathcal{S}}(h^{\mathcal{S}}) - \epsilon^{\mathcal{T}}(h^{\mathcal{T}})| \leqslant 2L_f L_g d_{\mathrm{W}}(\mathbb{P}(G^{\mathcal{S}}), \mathbb{P}(G^{\mathcal{T}})) + \Delta_h, \tag{C.50}$$

where,

$$\Delta_h = \min\left(\left|\epsilon^{\mathcal{S}}(h^{\mathcal{S}}) - \epsilon^{\mathcal{S}}(h^{\mathcal{T}})\right|, \left|\epsilon^{\mathcal{T}}(h^{\mathcal{S}}) - \epsilon^{\mathcal{T}}(h^{\mathcal{T}})\right|\right). \tag{C.51}$$

Such that, the following upper bound of target domain generalization error holds,

$$\epsilon^{\mathcal{T}}(h^{\mathcal{T}}) \leqslant \epsilon^{\mathcal{S}}(h^{\mathcal{S}}) + 2L_f L_g d_{\mathrm{W}}(\mathbb{P}(G^{\mathcal{S}}), \mathbb{P}(G^{\mathcal{T}})) + \Delta_h. \tag{C.52}$$

We next present the following corollary with proof given in Mohri (2018) Theorem 11.8.

---

**Corollary C.5** (VC-Dimension Generalization Bounds (Mohri, 2018) Corollary 3.19). *Let $\mathcal{H}$ be a family of functions with* $\mathrm{VCdim}(\mathcal{H}) = d$. *Then, for any $\delta > 0$, with probability at least $1 - \delta$, the following holds for all $h \in \mathcal{H}$,*

$$\epsilon(h) \leqslant \widehat{\epsilon}(h) + \left(\frac{2d}{m} \log \frac{em}{d}\right)^{\frac{1}{2}} + \left(\frac{1}{2m} \log \frac{1}{\delta}\right)^{\frac{1}{2}}. \tag{C.53}$$

*Thus, the form of this generalization bound is,*

$$\epsilon(h) \leqslant \widehat{\epsilon}(h) + \mathcal{O}\left(\frac{d}{m} \log \frac{m}{d}\right)^{\frac{1}{2}}, \tag{C.54}$$

*which emphasizes the importance of the ratio $m/d$ for generalization. The theorem provides another instance of Occam's razor principle where simplicity is measured in terms of a smaller VC-dimension.*

---

By replacing $\epsilon(h)$ with $\epsilon^{\mathcal{T}}(h^{\mathcal{T}})$, $\widehat{\epsilon}(h)$ with $\widehat{\epsilon}^{\mathcal{T}}(h^{\mathcal{T}})$ in Corollary C.5, with probability at least $1 - \delta$, the following holds,

$$\epsilon^{\mathcal{T}}(h^{\mathcal{T}}) \leqslant \widehat{\epsilon}^{\mathcal{T}}(h^{\mathcal{T}}) + \left(\frac{2d}{m} \log \frac{em}{d}\right)^{\frac{1}{2}} + \left(\frac{1}{2m} \log \frac{1}{\delta}\right)^{\frac{1}{2}}. \tag{C.55}$$

Upon the reference of Corollary C.5, we transform Eq. (C.52) into, with probability at least $1 - \delta$, the following holds,

$$\epsilon^{\mathcal{T}}(h^{\mathcal{T}}) \leqslant \widehat{\epsilon}^{\mathcal{S}}(h^{\mathcal{S}}) + 2L_f L_g d_{\mathrm{W}}(\mathbb{P}(G^{\mathcal{S}}), \mathbb{P}(G^{\mathcal{T}})) + \Delta_h + \left(\frac{2d}{n} \log \frac{en}{d}\right)^{\frac{1}{2}} + \left(\frac{1}{2n} \log \frac{1}{\delta}\right)^{\frac{1}{2}}, \tag{C.56}$$

Inspired by Li et al. (2021); You et al. (2023), we apply a union bound to combine Eq. (C.55) and Eq. (C.56) with coefficients $m/(n+m)$ and $n/(n+m)$, respectively, we reach,

$$\epsilon^{\mathcal{T}}(h^{\mathcal{T}}) \leqslant \frac{m}{n+m}\left(\widehat{\epsilon}^{\mathcal{T}}(h^{\mathcal{T}}) + \left(\frac{2d}{m} \log \frac{em}{d}\right)^{\frac{1}{2}} + \left(\frac{1}{2m} \log \frac{1}{\delta}\right)^{\frac{1}{2}}\right)$$

$$+ \frac{n}{n+m}\left(\widehat{\epsilon}^{\mathcal{S}}(h^{\mathcal{S}}) + 2L_f L_g d_{\mathrm{W}}(\mathbb{P}(G^{\mathcal{S}}), \mathbb{P}(G^{\mathcal{T}})) + \Delta_h + \left(\frac{2d}{n} \log \frac{en}{d}\right)^{\frac{1}{2}} + \left(\frac{1}{2n} \log \frac{1}{\delta}\right)^{\frac{1}{2}}\right) \tag{C.57}$$

$$\leqslant \frac{m}{n+m}\left(\widehat{\epsilon}^{\mathcal{T}}(h^{\mathcal{T}}) + \left(\frac{2d}{m} \log \frac{em}{d}\right)^{\frac{1}{2}} + \left(\frac{1}{2m} \log \frac{1}{\delta}\right)^{\frac{1}{2}}\right) + \frac{n}{n+m}\left(\widehat{\epsilon}^{\mathcal{S}}(h^{\mathcal{S}}) + \left(\frac{2d}{n} \log \frac{en}{d}\right)^{\frac{1}{2}} + \left(\frac{1}{2n} \log \frac{1}{\delta}\right)^{\frac{1}{2}}\right)$$

$$+ \frac{n}{n+m}(2L_f L_g d_{\mathrm{W}}(\mathbb{P}(G^{\mathcal{S}}), \mathbb{P}(G^{\mathcal{T}})) + \Delta_h). \tag{C.58}$$

**Definition C.6** (**Simplfied Cauchy-Schwarz Inequality**). *For any real numbers $x$ and $y$, the Cauchy-Schwarz inequality in its alternative additive form states that,*

$$x + y \leqslant [2(x^2 + y^2)]^{\frac{1}{2}}. \tag{C.59}$$

*This inequality is derived directly from the standard form of the Cauchy-Schwarz inequality and can be used to simplify expressions involving sums or to bound the magnitude of quadratic terms.*

We applied the simplified Cauchy-Schwarz Inequality to the two square root terms in Eq. (C.58), then we reach,

$$
\epsilon^{\mathcal{T}}(h^{\mathcal{T}}) \leqslant \frac{m}{n+m}\left(\widehat{\epsilon}^{\mathcal{T}}(h^{\mathcal{T}}) + \left(\frac{4d}{m}\log\frac{em}{d} + \frac{1}{m}\log\frac{1}{\delta}\right)^{\frac{1}{2}}\right) + \frac{n}{n+m}\left(\widehat{\epsilon}^{\mathcal{S}}(h^{\mathcal{S}}) + \left(\frac{4d}{n}\log\frac{en}{d} + \frac{1}{n}\log\frac{1}{\delta}\right)^{\frac{1}{2}}\right)
$$
$$
+ \frac{n}{n+m}(2L_f L_g d_{\mathsf{W}}(\mathbb{P}(G^{\mathcal{S}}), \mathbb{P}(G^{\mathcal{T}})) + \Delta_h) \tag{C.60}
$$
$$
\leqslant \frac{m}{n+m}\widehat{\epsilon}^{\mathcal{T}}(h^{\mathcal{T}}) + \frac{n}{n+m}\widehat{\epsilon}^{\mathcal{S}}(h^{\mathcal{S}}) + \frac{n}{n+m}(2L_f L_g d_{\mathsf{W}}(\mathbb{P}(G^{\mathcal{S}}), \mathbb{P}(G^{\mathcal{T}})) + \Delta_h)
$$
$$
+ \left(\frac{m^2}{(n+m)^2}\left(\frac{8d}{m}\log\frac{em}{d} + \frac{2}{m}\log\frac{1}{\delta}\right) + \frac{n^2}{(n+m)^2}\left(\frac{8d}{n}\log\frac{en}{d} + \frac{2}{n}\log\frac{1}{\delta}\right)\right)^{\frac{1}{2}}. \tag{C.61}
$$

As $m \ll n$, we have,

$$
\epsilon^{\mathcal{T}}(h^{\mathcal{T}}) \leqslant \frac{m}{n+m}\widehat{\epsilon}^{\mathcal{T}}(h^{\mathcal{T}}) + \frac{n}{n+m}\widehat{\epsilon}^{\mathcal{S}}(h^{\mathcal{S}}) + 2L_f L_g d_{\mathsf{W}}(\mathbb{P}(G^{\mathcal{S}}), \mathbb{P}(G^{\mathcal{T}})) + \Delta_h
$$
$$
+ \mathcal{O}\left[\left(\frac{1}{m} + \frac{1}{n}\right)\log\frac{1}{\delta} + \frac{d}{n}\log\frac{n}{d} + \frac{d}{m}\log\frac{m}{d}\right]^{\frac{1}{2}}. \tag{C.62}
$$

We conclude the proof of Proposition 4.5. $\qquad\square$

### C.5. Proof of Proposition 4.6: The Lower Bound of $L_f$

We first restate Proposition 4.6 here.

**Proposition 4.6** (**The Lower Bound of $L_f$** (Gama et al., 2020; You et al., 2023)) *Rewrite the first Wasserstein distance (optimal transport distance) between $G_i$ and $G_j$,*

$$d_W(G_i, G_j) = \inf_{\mathbf{P}\in\Pi}\left(\|\mathbf{X}_i - \mathbf{P}\mathbf{X}_j\|_F + \|\mathbf{A}_i - \mathbf{P}\mathbf{A}_j\mathbf{P}^\top\|_F\right),$$

*where $\mathbf{P}$ is the optimized permutation matrix by minimizing $d_W(G_i, G_j)$, $\Pi$ is matrix set. Such that, the lower bound of the Lipschitz constant of the encoder $f$ is,*

$$L_f \geqslant L_\lambda\left(1 + \Gamma_{ij}\sqrt{N_\Delta}\right) + \mathcal{O}(\Gamma^2),$$
$$\Gamma = \|\mathbf{A}_i - \mathbf{P}^\star\mathbf{A}_j\mathbf{P}^{\star\top}\|_F, \Gamma_{ij} = (\|\mathbf{U}_i - \mathbf{U}_j\|_F + 1)^2 - 1,$$

*where $\mathbf{P}^\star$ is the optimal permutation, $\mathbf{A}_i = \mathbf{U}_i\mathbf{\Lambda}_i\mathbf{U}_i^\top$ is the eigenvalue decomposition for both the adjacency matrices. $L_\lambda$ is the spectral Lipschitz constant, defined such that for any eigenvalues $\lambda_1$, $\lambda_2$, the inequality $|r(\lambda_1) - r(\lambda_2)| \leqslant L_\lambda|\lambda_1 - \lambda_2|$, where $r(\cdot)$ is the frequency response. $N_\Delta$ is graph size after padding isolated nodes (Zhu et al., 2021).*

*Proof.* Derived by Theorem 1 in Gama et al. (2020), Lemma 1 and Lemma 2 in You et al. (2023), we have the inequality,

$$\|f(\mathbf{X}_i, \mathbf{A}_i) - f(\mathbf{X}_j, \mathbf{A}_j)\|_2 \leqslant L_\lambda\Gamma\left(1 + \Gamma_{ij}\sqrt{N_\Delta}\right) + \mathcal{O}(\Gamma^2) + \lambda\Gamma, \tag{C.63}$$

where $\Gamma = \|\mathbf{A}_i - \mathbf{P}^\star\mathbf{A}_j\mathbf{P}^{\star\top}\|_F$ is the structure perturbations, and $\Gamma_{ij}$ is the eigenvector misalignment. Since,

$$L_f \triangleq \|f\|_{\mathsf{Lip}} = \sup_{G_i, G_j}\frac{\|f(\mathbf{X}_i, \mathbf{A}_i) - f(\mathbf{X}_j, \mathbf{A}_j)\|_2}{d_W(G_i, G_j)}, \tag{C.64}$$

for any $G_i$ and $G_j \in \{G^{\mathcal{S}} \cup G^{\mathcal{T}}\}$, $\|f(\mathbf{X}_i, \mathbf{A}_i) - f(\mathbf{X}_j, \mathbf{A}_j)\|_2 \leqslant L_f d_{\mathsf{W}}(G_i, G_j)$. Then, Eq. (C.63) is transformed to,

$$L_\lambda \Gamma \left(1 + \Gamma_{ij}\sqrt{N_\Delta}\right) + \mathcal{O}(\Gamma^2) + \lambda \Gamma \leqslant L_f d_{\mathsf{W}}(G_i, G_j) \ (L_f \text{ is the tight supremum}) \tag{C.65}$$

$$\leqslant L_f \Gamma. \tag{C.66}$$

As $\lambda$ and $\Gamma$ are both non-negative, then we reach,

$$L_f \Gamma \geqslant L_\lambda \Gamma \left(1 + \Gamma_{ij}\sqrt{N_\Delta}\right) + \mathcal{O}(\Gamma^2). \tag{C.67}$$

Dividing both sides of the inequality by $\Gamma$, we obtain,

$$L_f \geqslant L_\lambda \left(1 + \Gamma_{ij}\sqrt{N_\Delta}\right) + \mathcal{O}(\Gamma). \tag{C.68}$$

Since elements in adjacency matrics $\mathbf{A}_i$ and $\mathbf{A}_j$ belongs to $[0, 1]$, such that $\Gamma = \|\mathbf{A}_i - \mathbf{P}^\star \mathbf{A}_j \mathbf{P}^{\star\top}\|_{\mathsf{F}} \in [0, 1]$. Consequently, $\mathcal{O}(\Gamma^2) \leqslant \mathcal{O}(\Gamma)$. Then, we reach,

$$L_f \geqslant L_\lambda \left(1 + \Gamma_{ij}\sqrt{N_\Delta}\right) + \mathcal{O}(\Gamma^2). \tag{C.69}$$

We conclude the proof of Proposition 4.6. □

## D. Experiment Details

In this section, we provide additional experiment details. Statistic of datasets is illustrated in Table D.1.

*Table D.1.* Statistic of the multi-domain graph dataset.

| Dataset | Domain | # Nodes | # Edges | # Feature Dimensions | # Classes | Avg. # Deg. |
|---------|--------|---------|---------|---------------------|-----------|-------------|
| Cora | Academic | 2,708 | 10,556 | 1,433 | 7 | 3.90 |
| CiteSeer | Academic | 3,327 | 9,104 | 3,703 | 6 | 2.77 |
| PubMed | Academic | 19,717 | 88,648 | 500 | 3 | 4.50 |
| Photo | E-Commerce | 7,650 | 238,162 | 745 | 8 | 31.13 |
| Computers | E-Commerce | 13,752 | 491,722 | 767 | 10 | 35.76 |
| Reddit | Social Network | 232,965 | 114,615,892 | 602 | 41 | 492.00 |

### D.1. Datasets Details

To highlight multi-domain pre-training, we select **six** benchmark graph datasets from **three** different domains, which differs from conventional settings that consider any single dataset as one independent domain. These datasets span varying structures, scales, and feature distributions, offering a diverse set of challenges for pre-trained graph models. Specifically,

**(1) Academic Domain.** This domain includes **three** prominent citation networks, which are widely used as benchmarks in graph representation learning. Each dataset represents a single graph where nodes correspond to documents, edges correspond to citation links, and node labels indicate the category or topic of the corresponding paper.

- Cora (McCallum et al., 2000): consists of 2,708 machine learning-related papers organized into 7 research categories, including "Neural Networks" and "Probabilistic Methods". Each paper is represented by a 1,433-dimensional sparse bag-of-words feature vector, where each dimension corresponds to the presence of a specific word in the document. The graph contains 10,556 citation links, making it moderately sparse. The dataset is frequently used for semi-supervised node classification tasks due to its small size and well-defined class boundaries.

- CiteSeer (Giles et al., 1998): comprising 3,327 scientific publications grouped into 6 predefined categories, including "Artificial Intelligence" and "Robotics". Nodes are represented by 3,703-dimensional sparse feature vectors, capturing word occurrences in the documents. The graph has 9,104 citation edges, forming a sparse structure with fewer connections per node compared to Cora. CiteSeer is challenging for its noisy class labels and relatively low connectivity, presenting obstacles in capturing structural information.

- `PubMed` (Sen et al., 2008): a biomedical citation network includes 19,717 research articles on diabetes, categorized into 3 classes. Each node is represented by a dense 500-dimensional feature vector derived from Term Frequency-Inverse Document Frequency (TF-IDF) statistics of medical terms. The graph contains 88,648 citation edges, resulting in a denser structure compared to the other two citation datasets. Due to its size and real-world biomedical context, `PubMed` is often considered a benchmark for scalable graph learning algorithms.

**(2) E-Commerce Domain.** This domain includes **two** product networks derived from Amazon, capturing relationships between items frequently bought together. The two datasets are widely used for evaluating recommendation and product-related tasks. Nodes denote products, edges for co-purchase relationships, and node labels correspond to product categories.

- `Photo` (Shchur et al., 2018): is constructed from the "Electronics" category on Amazon, focusing specifically on camera and photo products. It consists of 7,650 products (nodes) and 238,162 co-purchase relationships (edges). Each node is represented by a 745-dimensional feature vector capturing product reviews and descriptions as word embeddings. Nodes are categorized into 8 classes based on product types. The graph exhibits a moderately sparse structure and serves as a benchmark for semi-supervised node classification.
- `Computers` (Shchur et al., 2018): is also derived from the Amazon "Electronics" category but focuses on computer-related products. It comprises 13,752 products (nodes) and 491,722 co-purchase relationships (edges), forming a denser graph compared to `Photo`. Each node is characterized by the same 767-dimensional feature vector format, based on product reviews and descriptions. The products are categorized into 10 classes. `Computers` is often used to test the scalability and robustness of graph learning models on larger and denser graphs.

**(3) Social Network Domain.** This domain includes `Reddit` (Hamilton et al., 2017), a large-scale social network constructed from the Reddit platform. In this dataset, nodes represent individual posts, and edges signify interactions between posts, such as comments, replies, or votes. The graph comprises 232,965 nodes and 114,15,892 edges, making it one of the largest datasets commonly used in graph representation learning. Each node is associated with a 602-dimensional feature vector generated from the textual content of the posts using pre-trained text embeddings. Posts are grouped into 41 distinct categories, corresponding to specific subreddits or discussion communities. Due to its size and dynamic nature, `Reddit` is frequently used as a benchmark for evaluating models on inductive learning tasks, particularly to test their scalability and generalization ability on unseen nodes and large-scale graphs.

### D.2. Baseline Details

We compare the proposed BRIDGE with **four** primary categories, **15** state-of-the-art baselines, including traditional graph neural networks, self-supervised pre-training methods, prompt-based fine-tuning approaches, and multi-domain pre-training frameworks. These baselines provide a comprehensive benchmark to assess the effectiveness and versatility of our approach.

**(1) Vanilla Graph Neural Networks.** This group includes traditional graph neural networks trained directly on downstream tasks without any pre-training. These models serve as foundational baselines for graph representation learning.

- `GCN` (Kipf & Welling, 2022) (backbone): The Graph Convolutional Network (`GCN`) is a spectral-based approach that aggregates neighborhood information using a layer-wise propagation rule, effectively capturing local graph structure. It is applied to both node and graph classifications.
- `GAT` (Velickovic et al., 2017): The Graph Attention Network (`GAT`) incorporates attention mechanisms to assign learnable weights to neighbors, allowing the model to focus on the most relevant nodes in a graph during the aggregation process. It is applied in both node and graph classifications.

**(2) Graph Self-Supervised Pre-training.** These methods leverage self-supervised learning objectives to pre-train graph representations, aiming at enhancing performance across various downstream tasks.

- `GCC` (Qiu et al., 2020): This framework focuses on learning transferable structural representations by using subgraph instance discrimination across multiple networks. It employs contrastive learning to capture universal topological properties, enabling better performance on diverse downstream tasks and out-of-domain generalization. It is applied to node classifications.
- `DGI` (Veličković et al., 2019): `DGI` maximizes mutual information between local patch representations and global graph summaries, capturing meaningful node-level representations. Unlike prior approaches, it does not rely on random walks

and works well in both transductive and inductive settings, showing strong performance on node classification tasks. It is applied to node classifications.

- `InfoGraph` (Sun et al., 2020): `InfoGraph` learns graph-level representations by maximizing mutual information between graph-level and multi-scale substructure representations (*e.g.*, nodes, edges, triangles). This approach encodes shared structural and semantic information, achieving superior performance on graph classification and molecular property prediction tasks. It is applied to graph classifications.

- `GraphCL` (You et al., 2020): This method introduces graph contrastive learning with various graph augmentations, enabling the generation of robust and generalizable graph representations. By systematically exploring augmentation strategies, `GraphCL` achieves strong results in semi-supervised, unsupervised, and transfer learning scenarios. It is applied to both node and graph classifications.

- `DSSL` (Xiao et al., 2022): `DSSL` addresses the limitations of homophily assumptions in existing methods by decoupling semantic information in self-supervised learning. Through a latent variable modeling framework, it captures diverse neighborhood semantics and achieves improved performance across homophilic and heterophilic graph benchmarks. It is applied to both node and graph classifications.

- `GraphACL` (Xiao et al., 2024): `GraphACL` introduces asymmetric contrastive learning to better handle heterophilic graphs. It eliminates the need for prefabricated augmentations and homophily assumptions, effectively capturing local and monophily similarities. This simple yet powerful approach outperforms existing contrastive and self-supervised methods in both homophilic and heterophilic graph scenarios. It is applied to both node and graph classifications.

**(3) Graph Prompt Fine-tuning.** Prompt-based methods introduce learnable task-specific prompts to guide the fine-tuning of pre-trained models. These approaches are designed to improve transferability and adaptability to downstream tasks.

- `GPPT` (Sun et al., 2022): `GPPT` introduces a novel paradigm combining pre-training and prompt tuning. It uses masked edge prediction as the pre-training task and reformulates downstream tasks (*e.g.*, node classification) into edge prediction tasks via token-pair prompts. This approach minimizes the objective gap between pre-training and downstream tasks, enabling efficient knowledge elicitation from pre-trained models without extensive fine-tuning. It is applied to node classifications.

- `GraphPrompt` (Liu et al., 2023b): `GraphPrompt` unifies pre-training and downstream tasks into a shared template using a learnable prompt. By narrowing the gap between pre-training and downstream objectives, it improves task-specific knowledge extraction and reduces labeling requirements. It provides a generalizable framework suitable for diverse downstream tasks. It is applied to both node and graph classifications.

- `GraphPrompt+` (Yu et al., 2024b): An enhanced version of `GraphPrompt`, this method expands compatibility with pre-training tasks and introduces hierarchical prompt vectors across graph encoder layers. These advancements improve representation learning, enabling better utilization of multi-scale structural information and enhancing performance across tasks. It is applied to both node and graph classifications.

- `GPF` (Fang et al., 2024): `GPF` proposes a universal prompt-based tuning method applicable to pre-trained GNNs under any pre-training strategy. Operating in the input feature space, it adapts pre-trained models to downstream tasks flexibly and effectively, outperforming traditional fine-tuning and specialized prompt-tuning methods across diverse scenarios. It is applied to both node and graph classifications.

- `ProNoG` (Yu et al., 2025): `ProNoG` focuses on non-homophilic graphs, recognizing their mixed structural patterns. It incorporates a conditional network to adaptively model node-specific characteristics during downstream tasks. This approach generalizes well to both homophilic and heterophilic settings, providing robust performance on real-world non-homophilic graphs. It is applied to both node and graph classifications.

**(4) Multi-Domain Graph Pre-training.** This category focuses on learning graph representations that generalize across different domains, which are the most relative baselines with ours. By leveraging shared structural or semantic patterns, these methods aim to improve performance in multi-domain scenarios.

- `GCOPE` (Zhao et al., 2024): `GCOPE` introduces a unification framework to aggregate diverse graph datasets during pre-training, effectively distilling shared structural knowledge. By addressing the negative transfer issue often encountered in cross-domain learning, `GCOPE` enhances few-shot learning performance on target tasks. It leverages the synergistic potential of multiple graph datasets, pioneering the development of graph foundational models for multi-domain pre-training. It is applied to both node and graph classifications.

- MDGPT (Yu et al., 2024c): MDGPT proposes a text-free multi-domain graph pre-training framework designed to align and integrate knowledge from diverse graph domains without relying on textual attributes. It introduces domain tokens to unify features across source domains and employs dual prompts (unifying and mixing prompts) to adapt the model to target domains. This approach effectively mitigates domain conflicts and enhances the relevance of source domain knowledge, achieving significant performance improvements across various datasets. It is applied to both node and graph classifications.

### D.3. Experimental Setting Details

In this section, we introduce detailed experimental settings.

**Detailed Settings for Section 5.2 (*RQ1*).** This section evaluates BRIDGE's ability to transfer knowledge from source domains to unseen target datasets or domains under few-shot learning settings, specifically ($C$-way-)one-shot, five-shot, and $m$-shot ($1 \leqslant m \leqslant 10$), which goal is to evaluate the model's generalization ability by pre-training BRIDGE on source domain graphs and then fine-tuning it with only a few labeled target samples. ❶ $m$-**shot settings.** In all cases, only the $m$ randomly selected labeled samples per class are used for training (supporting set), and the remaining samples in the dataset are used for testing (query set). ❷ **Generalization settings.** Two types of generalization settings are considered, *i.e.*, cross-dataset setting (the target dataset belongs to the same domain as the source datasets but was unseen during pre-training) and cross-domain setting (the target dataset belongs to a different domain from the source datasets). It is noteworthy that, some prior cross-domain graph pre-training and fine-tuning works (Zhao et al., 2024; Yu et al., 2024c) consider different datasets as different domains. However, in this paper, we argue that treating datasets as domains is too coarse-grained, and instead, the distinction between domains should be based on the broader fields to which the datasets belong. Therefore, the "cross-dataset" setting in this section aligns with what other works refer to as "cross-domain", while the "cross-domain" setting in this paper refers to a higher-level distinction between broader domains. ❸ **Downstream Task settings.** We evaluate both the node classification and graph classification. For graph classification, we sample 2-hop ego-graphs for each node to synthesize the corresponding datasets for graph-level tasks. Note that, GCC, DGI, and GPPT are only for node-level tasks comparison, while InfoGraph is only for graph-level tasks comparison (as an extension of DGI).

**Detailed Settings for Section 5.3 (*RQ2*).** To analyze the effectiveness of BRIDGE's key components, we further elaborate on the following three variants, each with a specific module removed. The full BRIDGE is also included as a baseline to compare against the variants. ❶ **BRIDGE** (*w/o* $\mathcal{A}_{\mathrm{s}}$). Removes the semantic-wise aligner $\mathcal{A}_{\mathrm{s}}$ responsible for feature alignment across domains. This variant only applies dimension-wise alignment ($\mathcal{A}_{\mathrm{s}}$) but does not ensure feature consistency across domains at the semantic level. Specifically, directly apply $\widehat{\mathbf{X}}^{\mathcal{S}}$ in Eq. (1) with adjacencies $\mathbf{A}$ to encode and stack $\mathbf{H}^{\mathcal{S}}$, which is then be integrated in $\mathcal{L}_{\mathrm{pre}}(\boldsymbol{\Theta}; \mathbf{H}^{\mathcal{S}})$ in Eq. (2) as the ultimate goal of pre-training phase. Consequently, the alignment risk regularization objectives in Eq. (3), Eq. (4), and Eq. (5) are abandoned. This variant tests the necessity of ensuring semantic consistency between different graph domains. ❷ **BRIDGE** (*w/o* MoE). Removes the lightweight Mixture of Experts (MoE) network, which is responsible for selectively assembling source domain knowledge during prompting. Instead of selectively assembling knowledge from different source domains, this variant uses a fixed, non-adaptive knowledge transfer mechanism (treats all pre-trained features as equally relevant for the target task with averaged routing). Specifically, the constant routing weight vector $\mathbf{S} = [1/n]^n$ is applied on $\mathcal{A}_{\mathrm{s}}^{\mathrm{T}}$) to obtain $\widehat{\mathbf{X}}^{\mathcal{T}'}$ in Eq. (7). This variant tests whether a static prompt learning strategy can be as effective as an adaptive expert routing mechanism. ❸ **BRIDGE** (*w/o* $\mathcal{R}_{\mathrm{spec}}$): Removes the spectral regularizer $\mathcal{R}_{\mathrm{spec}}$, which constrains the upper bound of generalization error. Without the spectral regularizer, the model still follows the "pretrain-then-prompt" paradigm but lacks an explicit regularization mechanism to guarantee the transferability of knowledge. Specifically, we directly omit $\mathcal{R}_{\mathrm{spec}}$ term in Eq. (21), which only leaves $\mathcal{L}_{\mathrm{down}}$ in Eq. (11) as the prompt objectives. This variant tests the role of theoretical constraints on knowledge transfer and whether they significantly impact downstream performance.

**Detailed Settings for Section 5.4 (*RQ3*).** This section aims to analyze how efficiently BRIDGE performs downstream adaptation (prompt-based fine-tuning) compared to other baselines, which represents a challenging real-world scenario where labeled samples are limited, making efficient adaptation crucial. ❶ **Datasets and Baselines.** The evaluation is conducted on the Photo dataset under a five-shot node classification setting (cross-dataset). Results are compared to ProNoG, GCOPE, and MDGPT, which represent state-of-the-art graph pre-training and adaptation approaches. ❷ **Efficiency Metrics.** We report results with the loss curvature, and accuracy curvature, where the reduction in training loss over epochs is tracked to measure how quickly each model stabilizes, the accuracy is recorded at different epochs to show how rapidly the model reaches its peak performance, and the total number of fine-tuning epochs needed for convergence is measured and compared.

**Detailed Settings for Section 5.5 (*RQ4*).** Hyperparameter sensitivity analysis aims to analyze how different key hyperparameters in BRIDGE affect its performance. We track the fluctuation led by four hyperparameters. ❶ $K$**.** Controls the number of domain-specific variations (times of causal interventions) generated during the multi-domain semantic alignment process in Eq. (4). It determines how diverse the source domain embeddings become before knowledge transfer. Higher $K$ simulates more varied domain diversity, which potentially brings better generalization performance, but heavier computational burdens. ❷ $\alpha$**.** Balances the expectation term and variation term in the multi-domain semantic alignment objective in Eq. (4). It influences how strictly domain-specific features are aligned across different datasets. ❸ $\beta$**.** Controls the weight between the classification loss and the self-supervised MoE loss in Eq. (11). Higher values give more importance to expert routing in knowledge selection. ❹ $\gamma$**.** Regulates the effect of spectral regularization on controlling generalization error in Eq. (21). It determines the extent to which the model constrains feature transferability.

**Detailed Settings for Section 5.6 (*RQ5*).** This section investigates whether Large Language Models (LLMs) can improve the alignment of graph features across different domains. Specifically, this experiment evaluates whether leveraging LLM-generated textual descriptions enhances the cross-domain generalization of BRIDGE. Since node features in the benchmark dataset are considered as"secondary data" derived from raw text through feature engineering, we follow Liu et al. (2024a); Li et al. (2024b); Liu et al. (2024b); Chen et al. (2024b) and establish a correspondence between the raw text and the processed features. Specifically, for a given raw sentence text $\mathbf{s}$, we encode it using SentenceBERT (Reimers, 2019) to obtain a textual feature representation $\mathbf{X_s}$ with the same dimension as the original node feature $\mathbf{X}$. The two types of features are then concatenated as $[\mathbf{X}\|\mathbf{X_s}]$ to form the new initial node feature, enabling the enhancement of node representations by leveraging the semantic understanding capabilities of LLMs.

# E. Additional Results

In this section, we provide additional experiment results.

### E.1. Additional Results: Five-Shot Node and Graph Classification

We evaluate our BRIDGE against 15 state-of-the-art baselines under the challenging one-shot setting for both node and graph classification, assessing its effectiveness in transferring source knowledge across cross-dataset and cross-domain scenarios. The results, presented in Table E.1 and Table E.2, highlight the following key insights. ❶ Results of the five-shot node classification graph classification demonstrate significant performance improvements for all models, including BRIDGE and baselines, compared to the one-shot results in Table 1 and Table 2. For example, BRIDGE achieves notable results such as 63.85% on `Cora` and 71.14% on `Computers` in node classification, significantly higher than the runner-up methods. In graph classification, BRIDGE attains accuracies such as 70.53% on `PubMed` and 84.42% on `Reddit`, showcasing its robustness. ❷ Compared to the one-shot setting, BRIDGE's performance improvement in one-shot tasks is more substantial, highlighting its capability to effectively leverage additional labeled samples for fine-tuning. For example, BRIDGE achieves an accuracy gain of nearly 4% on `Reddit` compared to the best baseline in the five-shot setting, whereas the gain was less pronounced in the one-shot setting. ❸ We also witnessed the improved generalization in cross-domain scenarios, *i.e.*, BRIDGE's superior performance is particularly evident in cross-domain tasks, where domain shifts are more significant. For instance, BRIDGE achieves 80.12% (node classification) and 84.42% (graph classification) on `Reddit`, far exceeding the performance of baselines such as `MDGPT` and `ProNoG`.

### E.2. Additional Results: $m$-Shot Classification on Diverse Datasets

To further evaluate BRIDGE's performance under different few-shot settings varied in $m$, we selected three competitive baselines in *RQ1*, including `ProNoG`, `GCOPE` and `MDGPT` for comparison. Results on more diverse dataset are shown in Figure E.1, Figure E.2, Figure E.3, and Figure E.4. The results demonstrate how accuracy evolves as the number of labeled samples per class increases, key observations include, ❶ BRIDGE consistently outperforms the other three baselines across nearly all $m$-shot settings. Its superiority is particularly evident in both node and graph classification tasks, where it achieves state-of-the-art performance at higher shot levels. ❷ While BRIDGE shows only a slight advantage over the baselines when $m$ is small, this advantage grows significantly as $m$ increases. This trend can be attributed to the spectral regularizer, which enables more effective fine-tuning under increased supervision by leveraging down prompts with a theoretical generalization error upper bound guarantee. For example, in the graph classification task on `PubMed` dataset, BRIDGE steadily outperforms baselines such as `MDGPT` and `GCOPE` as $m$ increases, showcasing its ability to scale with additional labeled data. ❸ Similar trends are observed across both cross-dataset and cross-domain scenarios, suggesting that

*Table E.1.* Accuracy (% ± standard deviation for five runs) of **five-shot node** classification. `CR` =Cora, `CS` =CiteSeer, `PM` =PubMed, `Ph` =Photo, `Com` =Computers, `Rdt` =Reddit. The best results are shown in **bold** and the runner-ups are underlined.

| Source (Cross-Dataset \| Cross-Domain) | CS PM Ph Com | CR PM Ph Com | CR CS Ph Com | CR CS PM Com | CR CS PM Ph | CR CS PM Ph Com |
|---|---|---|---|---|---|---|
| **Model / Target** | CR | CS | PM | Ph | Com | Rdt |
| GCN (bb.) (Kipf & Welling, 2022) | 46.53 ± 5.36 | 46.59 ± 6.31 | 51.07 ± 5.89 | 56.17 ± 5.75 | 51.82 ± 6.93 | 66.65 ± 6.89 |
| GAT (Velickovic et al., 2017) | 46.47 ± 7.10 | 46.09 ± 5.91 | 50.47 ± 7.26 | 53.11 ± 7.80 | 49.78 ± 5.68 | 66.59 ± 5.07 |
| GCC (Qiu et al., 2020) | 47.05 ± 7.88 | 45.12 ± 6.78 | 52.45 ± 7.67 | 59.76 ± 7.15 | 51.96 ± 5.31 | 68.10 ± 5.12 |
| DGI (Veličković et al., 2019) | 47.77 ± 6.85 | 47.22 ± 5.66 | 52.43 ± 5.67 | 62.60 ± 6.41 | 54.66 ± 7.03 | 67.64 ± 5.52 |
| GraphCL (You et al., 2020) | 49.61 ± 6.67 | 48.76 ± 6.72 | 52.81 ± 5.27 | 59.55 ± 7.50 | 57.88 ± 7.36 | 69.94 ± 5.43 |
| DSSL (Xiao et al., 2022) | 46.28 ± 7.61 | 48.44 ± 7.26 | 52.42 ± 7.57 | 62.35 ± 6.94 | 54.15 ± 7.58 | 68.22 ± 5.40 |
| GraphACL (Xiao et al., 2024) | 51.38 ± 7.06 | 49.71 ± 7.12 | 54.48 ± 6.01 | 65.87 ± 6.54 | 59.72 ± 5.36 | 70.93 ± 7.41 |
| GPPT (Sun et al., 2022) | 48.24 ± 5.27 | 47.51 ± 7.54 | 52.86 ± 6.98 | 63.13 ± 6.78 | 52.11 ± 5.16 | 71.05 ± 5.59 |
| GraphPrompt (Liu et al., 2023b) | 52.48 ± 7.67 | 52.01 ± 5.54 | 56.85 ± 5.65 | 68.19 ± 7.09 | 60.05 ± 7.30 | 72.79 ± 6.23 |
| GraphPrompt+ (Yu et al., 2024b) | 52.16 ± 7.84 | 52.76 ± 7.09 | 55.82 ± 6.16 | 66.31 ± 5.90 | 61.07 ± 5.37 | 72.63 ± 5.77 |
| GPF (Fang et al., 2024) | 56.16 ± 5.27 | 53.97 ± 7.45 | 55.88 ± 5.77 | 67.65 ± 6.58 | 55.69 ± 6.05 | 74.00 ± 5.46 |
| ProNoG (Yu et al., 2025) | 58.06 ± 5.00 | 55.53 ± 7.58 | 56.53 ± 7.29 | 74.23 ± 5.90 | 63.54 ± 6.12 | 76.48 ± 6.82 |
| GCOPE (Zhao et al., 2024) | 52.87 ± 7.15 | 55.93 ± 6.32 | 53.05 ± 7.86 | 68.80 ± 6.63 | 60.46 ± 5.56 | 73.61 ± 5.27 |
| MDGPT (Yu et al., 2024c) | 59.51 ± 4.54 | 55.28 ± 4.47 | 56.25 ± 6.76 | 75.73 ± 4.54 | 64.38 ± 4.13 | 75.29 ± 4.25 |
| **BRIDGE (ours)** | **63.85 ± 5.87** | **60.20 ± 7.72** | **62.35 ± 5.92** | **81.95 ± 5.38** | **71.14 ± 4.86** | **80.12 ± 2.66** |

*Table E.2.* Accuracy (% ± standard deviation for five runs) of **five-shot graph** classification. `CR` =Cora, `CS` =CiteSeer, `PM` =PubMed, `Ph` =Photo, `Com` =Computers, `Rdt` =Reddit. The best results are shown in **bold** and the runner-ups are underlined.

| Source (Cross-Dataset \| Cross-Domain) | CS PM Ph Com | CR PM Ph Com | CR CS Ph Com | CR CS PM Com | CR CS PM Ph | CR CS PM Ph Com |
|---|---|---|---|---|---|---|
| **Model / Target** | CR | CS | PM | Ph | Com | Rdt |
| GCN (bb.) (Kipf & Welling, 2022) | 51.93±4.55 | 44.15±5.81 | 56.88±5.54 | 59.29±4.78 | 51.67±5.38 | 75.70±5.88 |
| GAT (Velickovic et al., 2017) | 48.95±6.08 | 43.65±5.66 | 55.89±5.86 | 57.18±4.92 | 50.91±6.22 | 74.82±6.16 |
| InfoGraph (Sun et al., 2020) | 52.81±4.84 | 45.51±5.44 | 60.47±6.25 | 62.18±5.24 | 53.40±5.49 | 76.34±4.97 |
| GraphCL (You et al., 2020) | 53.92±4.65 | 48.47±4.91 | 61.84±6.48 | 62.74±5.19 | 57.73±6.27 | 78.59±5.40 |
| DSSL (Xiao et al., 2022) | 55.03±5.57 | 45.58±6.20 | 62.45±4.95 | 63.04±6.13 | 57.26±6.33 | 78.82±4.91 |
| GraphACL (Xiao et al., 2024) | 55.52±5.83 | 47.26±5.24 | 61.44±5.90 | 63.69±6.33 | 56.54±6.46 | 79.75±6.20 |
| GraphPrompt (Liu et al., 2023b) | 55.24±6.39 | 52.31±5.04 | 63.09±6.22 | 64.27±5.86 | 58.12±5.19 | 79.48±4.56 |
| GraphPrompt+ (Yu et al., 2024b) | 59.53±5.07 | 51.22±5.88 | 62.81±5.83 | 65.73±4.60 | 58.61±6.24 | 78.77±6.15 |
| GPF (Fang et al., 2024) | 56.19±5.28 | 53.47±5.33 | 61.39±5.93 | 64.08±5.24 | 57.47±5.96 | 79.66±5.26 |
| ProNoG (Yu et al., 2025) | 61.07±5.31 | 51.25±5.98 | 64.73±6.12 | 66.10±5.14 | 60.55±5.17 | 80.26±5.05 |
| GCOPE (Zhao et al., 2024) | 57.29±4.77 | 53.97±6.13 | 62.61±4.94 | 65.77±5.35 | 58.20±4.65 | 79.87±6.22 |
| MDGPT (Yu et al., 2024c) | 61.77±4.10 | 55.37±6.33 | 62.70±6.20 | 67.06±3.72 | 59.76±8.56 | 80.29±5.46 |
| **BRIDGE (ours)** | **67.21±3.78** | **59.80±5.04** | **67.17±3.77** | **70.53±6.53** | **64.91±6.24** | **84.42±1.43** |

BRIDGE not only excels in domain-specific tasks but also demonstrates strong generalization in transferring knowledge across diverse domains. This robustness is further validated in challenging scenarios such as the `Reddit` dataset, where BRIDGE achieves significant accuracy gains compared to its competitors.

### E.3. Additional Results: Ablation Studies

The additional ablation results in Table E.3 and Table E.4 provide a detailed analysis of the impact of three key components in BRIDGE on node classification and graph classification across various datasets and shot settings ($m = 1$ and $m = 5$). We find that, ❶ Removing the semantic-wise aligner ($\mathcal{A}_s$) results in consistent accuracy drops across all datasets and tasks, particularly in node classification. For example, in `Cora` ($m = 5$), the accuracy decreases by 4.9%, demonstrating the importance of semantic alignment for effective knowledge transfer. ❷ The removal of the Mixture of Experts (MoE) module has a significant impact, especially in more challenging cross-domain datasets like `Reddit`. For instance, in node classification on `Reddit` ($m = 5$), accuracy drops by 10.1%, underscoring the role of selective knowledge assembly in

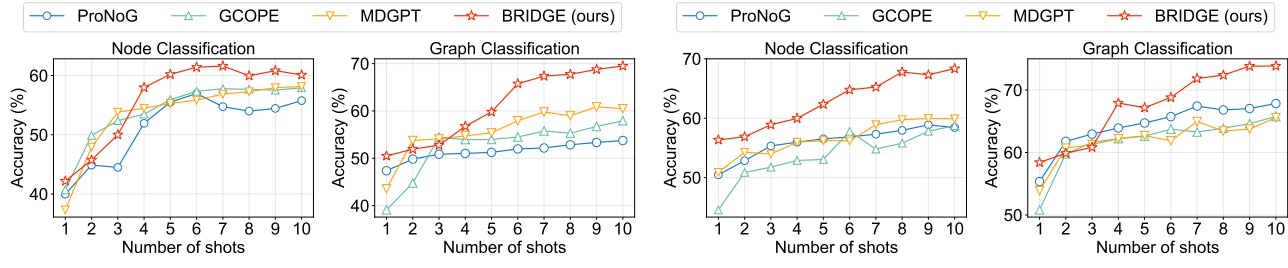

*Figure E.1.* $m$-shot evaluation on `CiteSeer`.

*Figure E.2.* $m$-shot evaluation on `PubMed`.

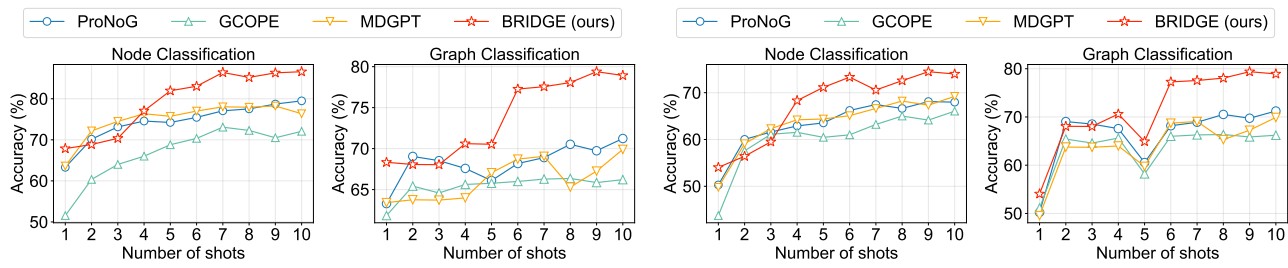

*Figure E.3.* $m$-shot evaluation on `Photo`.

*Figure E.4.* $m$-shot evaluation on `Computers`.

*Table E.3.* Accuracy (%) with relative decrease ($\Delta$%) of ablation on **node** classification. `CR`=Cora, `CS`=CiteSeer, `PM`=PubMed, `Ph`=Photo, `Com`=Computers, `Rdt`=Reddit. The biggest decrease is shown in **bold** and the runner-ups are underlined.

| **Source** | CS PM Ph Com | | | | CR PM Ph Com | | | | CR CS Ph Com | | | | CR CS PM Com | | | | CR CS PM Ph | | | | CR CS PM Ph Com | | | |
|---|---|---|---|---|---|---|---|---|---|---|---|---|---|---|---|---|---|---|---|---|---|---|---|---|
| **Target** | CR | | | | CS | | | | PM | | | | Ph | | | | Com | | | | Rdt | | | |
| **Variants** | $m=1$ | | $m=5$ | | $m=1$ | | $m=5$ | | $m=1$ | | $m=5$ | | $m=1$ | | $m=5$ | | $m=1$ | | $m=5$ | | $m=1$ | | $m=5$ | |
| **BRIDGE** | 46.4 | $\Delta$ | 63.9 | $\Delta$ | 42.2 | $\Delta$ | 60.2 | $\Delta$ | 56.4 | $\Delta$ | 62.4 | $\Delta$ | 67.9 | $\Delta$ | 62.4 | $\Delta$ | 54.0 | $\Delta$ | 71.1 | $\Delta$ | 68.5 | $\Delta$ | 80.1 | $\Delta$ |
| - (*w/o* $\mathcal{A}_s$) | 44.2 | **4.9** | 61.2 | 4.2 | 40.2 | 4.8 | 57.3 | 4.8 | 55.0 | 2.5 | 60.1 | 3.7 | 65.7 | **3.3** | 61.9 | 0.8 | 53.4 | 1.2 | 70.7 | 0.6 | 68.3 | 0.4 | 78.9 | 1.6 |
| - (*w/o* MoE) | 44.3 | 4.6 | 61.0 | 4.5 | 39.9 | **5.3** | 58.3 | 3.2 | 52.2 | **7.4** | 57.6 | 7.6 | 65.7 | 3.2 | 58.3 | 6.6 | 53.1 | **1.8** | 68.4 | 3.9 | 64.3 | **6.2** | 72.1 | **10.1** |
| - (*w/o* $\mathcal{R}_{spec}$) | 45.6 | 1.8 | 59.9 | **6.2** | 41.3 | 2.0 | 54.6 | **9.4** | 55.3 | 1.8 | 55.9 | **10.3** | 66.5 | 2.0 | 56.0 | **10.2** | 53.6 | 0.9 | 66.7 | **6.2** | 67.7 | 1.2 | 72.4 | 9.7 |

addressing complex cross-domain tasks. ❸ The spectral regularizer ($\mathcal{R}_{spec}$) has the most critical impact on performance. Its removal causes the largest accuracy drops, particularly in tasks with higher supervision ($m = 5$). For example, in `PubMed` for node classification, accuracy decreases by 10.3%, highlighting $\mathcal{R}_{spec}$'s effectiveness in enhancing fine-tuning with robust generalization guarantees. ❹ Overall, the effects of ablation are more pronounced in the five-shot setting, as the increased supervision enables the full BRIDGE to leverage its components more effectively. This highlights the importance of these modules in scenarios with more labeled data. Among the three key components, the spectral regularizer ($\mathcal{R}_{spec}$) consistently demonstrates the most critical role in both node and graph classification tasks. Its ability to constrain generalization error provides substantial benefits for cross-dataset and cross-domain learning. The Mixture of Experts (MoE) is essential for selectively transferring domain-specific knowledge, particularly in challenging cross-domain scenarios. The semantic-wise aligner ($\mathcal{A}_s$), while less impactful than $\mathcal{R}_{spec}$ and MoE, still contributes to stable improvements across tasks.

### E.4. Additional Results: Embedding Visualizations

We visualize node embeddings after pre-training and down-prompting (fine-tuning) in Figure E.5 and Figure E.6, respectively. During pre-training, we simultaneously take six datasets (belonging to three different domains) as the source knowledge. Note that, "BRIDGE Ablated" of the left subfigure in Figure E.6 refers to the removal of both the lightweight MoE network and the spectral regularize, as the comparison with the right subfigure. We find that, ❶ The pre-trained embeddings for nodes from six datasets, spanning three distinct domains (academic, e-commerce, and social networks), exhibit well-clustered structures. Nodes belonging to the same dataset are grouped tightly, demonstrating the ability of BRIDGE's pre-training phase to align domain-specific features effectively. Different domains remain distinguishable (forming into three

*Table E.4.* Accuracy (%) with relative decrease (Δ%) of ablation on **graph** classification. `CR`=Cora, `CS`=CiteSeer, `PM`=PubMed, `Ph`=Photo, `Com`=Computers, `Rdt`=Reddit. The biggest decrease is shown in **bold** and the runner-ups are underlined.

| Source | CS PM Ph Com | | | | CR PM Ph Com | | | | CR CS Ph Com | | | | CR CS PM Com | | | | CR CS PM Ph | | | | CR CS PM Ph Com | | | |
|---|---|---|---|---|---|---|---|---|---|---|---|---|---|---|---|---|---|---|---|---|---|---|---|---|
| Target | CR | | | | CS | | | | PM | | | | Ph | | | | Com | | | | Rdt | | | |
| Variants | $m=1$ | | $m=5$ | | $m=1$ | | $m=5$ | | $m=1$ | | $m=5$ | | $m=1$ | | $m=5$ | | $m=1$ | | $m=5$ | | $m=1$ | | $m=5$ | |
| **BRIDGE** | 53.8 | Δ | 67.2 | Δ | 50.5 | Δ | 59.8 | Δ | 58.4 | Δ | 67.2 | Δ | 68.3 | Δ | 70.5 | Δ | 54.1 | Δ | 64.9 | Δ | 75.8 | Δ | 84.4 | Δ |
| - (*w/o* $\mathcal{A}_s$) | 51.3 | 4.7 | 64.9 | 3.5 | 49.2 | 2.6 | 58.8 | 1.6 | 57.3 | 1.9 | 66.6 | 0.8 | 68.0 | 0.5 | 68.8 | 2.4 | 53.9 | 0.4 | 63.8 | 1.7 | 75.0 | 1.0 | 83.0 | 1.7 |
| - (*w/o* MoE) | 51.3 | **4.8** | 63.8 | 5.1 | 46.9 | **7.1** | 57.0 | 4.6 | 55.3 | **5.3** | 64.3 | 4.3 | 66.4 | **2.8** | 65.2 | **7.5** | 52.1 | **3.6** | 61.7 | 4.9 | 72.7 | **4.1** | 80.6 | 4.5 |
| - (*w/o* $\mathcal{R}_{spec}$) | 53.0 | 1.6 | 61.2 | **8.9** | 47.3 | 6.4 | 55.2 | **7.7** | 57.1 | 2.2 | 62.2 | **7.4** | 67.5 | 1.2 | 66.3 | 6.0 | 53.9 | 0.3 | 60.1 | **7.4** | 74.9 | 1.1 | 78.6 | **7.0** |

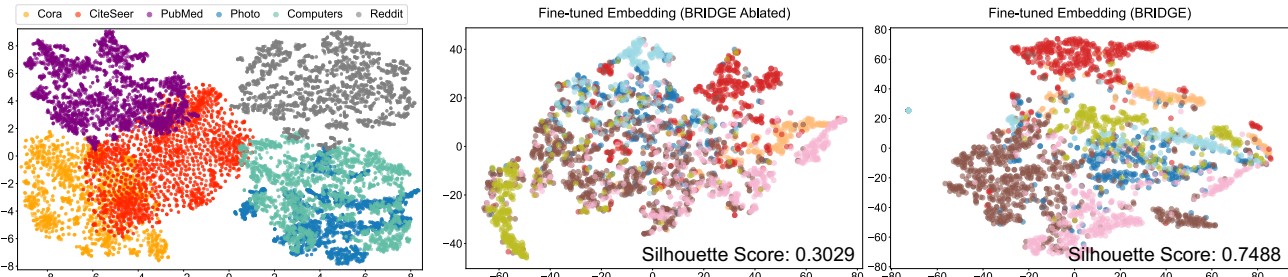

*Figure E.5.* Pre-trained Embeddings.     *Figure E.6.* Fine-tuned Embeddings on `Cora` ($m=5$). Colors denote class labels.

obvious domain clusters), reflecting the preservation of inter-domain characteristics while ensuring intra-domain consistency. This well-separated clustering highlights the success of BRIDGE's domain-invariant feature alignment mechanism in the pre-training stage, which is critical for effective transfer learning. ❷ With ablation (left), the silhouette score of 0.3029 indicates relatively poor class separability, reflecting the challenges posed by the absence of MoE and $\mathcal{R}_{spec}$. For full BRIDGE (right), the embeddings show significantly improved clustering, with clear boundaries between nodes of different classes. The silhouette score increases to 0.7488, demonstrating enhanced class separability. ❸ Overall, the additional visualization results provide strong evidence of the synergy between BRIDGE's pre-training and fine-tuning mechanisms, further validating the effectiveness of its key components.

## F. Implementation Details

In this section, we provide implementation details.

### F.1. Implementation Details of BRIDGE

**Pre-training.** The number of pre-training epochs for optimizing our proposed BRIDGE and all baselines is set to 10,000. An early stopping strategy is employed, where training is terminated if the pre-training loss does not decrease for 50 consecutive epochs. This ensures computational efficiency without sacrificing performance. The aligned feature dimension is set to 50, and the encoded hidden dimension is set to 256. For BRIDGE, the hyperparameters $\alpha$, $\beta$, and $\gamma$ are chosen from the range of 1e-3 to 5e-1. The optimal values are determined using Weights & Biases (wandb)[2], a framework for hyperparameter optimization and experiment tracking. For the optimization process, we utilize the Adam optimizer (Kingma & Ba, 2015), carefully tuning the learning rate and weight decay. The learning rate is chosen from the range of 1e-4 to 1e-2, and weight decay is selected from 1e-5 to 1e-2 through grid search on the validation set. All parameters are initialized randomly to avoid any bias from pre-trained weights.

### F.2. Implementation Details of Baselines

We provide the baseline methods implementations with their respective licenses as follows.

[2] https://wandb.ai

- GCN (Kipf & Welling, 2022): [MIT License] https://github.com/pyg-team/pytorch_geometric.
- GAT (Velickovic et al., 2017): [MIT License] https://github.com/pyg-team/pytorch_geometric.
- GCC (Qiu et al., 2020): [MIT License] https://github.com/THUDM/GCC.
- DGI (Veličković et al., 2019): [MIT License] https://github.com/PetarV-/DGI.
- InfoGraph (Sun et al., 2020): [Unspecified] https://github.com/sunfanyunn/InfoGraph.
- GraphCL (You et al., 2020): [MIT License] https://github.com/Shen-Lab/GraphCL.
- DSSL (Xiao et al., 2022): [Apache-2.0 License] https://github.com/BUPT-GAMMA/OpenHGNN/blob/main/openhgnn/models/DSSL.py.
- GraphACL (Xiao et al., 2024): [Unspecified] https://github.com/tengxiao1/GraphACL.
- GPPT (Sun et al., 2022): [Unspecified] https://github.com/MingChen-Sun/GPPT.
- GraphPrompt (Liu et al., 2023b): [Unspecified] https://github.com/Starlien95/GraphPrompt.
- GraphPrompt+ (Yu et al., 2024b): [Unspecified] https://github.com/gmcmt/graph_prompt_extension.
- GPF (Fang et al., 2024): [MIT License] https://github.com/zjunet/GPF.
- ProNoG (Yu et al., 2025): [Unspecified] https://github.com/Jaygagaga/ProNoG.
- GCOPE (Zhao et al., 2024): [Unspecified] https://github.com/cshhzhao/GCOPE.
- MDGPT (Yu et al., 2024c): [Unspecified] https://anonymous.4open.science/r/MDGPT.

For all baselines, we follow the recommended hyperparameter settings from the original papers or official implementations to ensure their optimal performance. In cases where recommended configurations are unavailable or suboptimal, we carefully fine-tune the hyperparameters to achieve the best possible performances. To ensure a fair comparison, we align other experimental settings such as hidden dimensions, number of model layers, *etc.*, with those of our proposed method. This consistent setup guarantees that performance differences arise solely from the model designs and not from external factors.

### F.3. Hardware and Software Configurations

We conduct the experiments with the following hardware and software configurations.

- Operating System: Ubuntu 20.04 LTS.
- CPU: Intel(R) Xeon(R) Platinum 8358 CPU@2.60GHz with 1TB DDR4 of Memory.
- GPU: NVIDIA Tesla A100 SMX4 with 80GB of Memory.
- Software: CUDA 10.1, Python 3.8.12, PyTorch[3] 1.9.1, PyTorch Geometric[4] 2.0.1.

## G. Discussions

To make BRIDGE easy-to-follow to readers, we discuss through the following **"How to Understand?"** questions.

### G.1. How to Understand Domain-Invariant Aligner?

The domain-invariant aligner in BRIDGE facilitates robust cross-domain knowledge transfer by combining **dimension-wise** and **semantic-wise** alignment. Dimension-wise alignment ensures basic dimensionality-level consistency by aligning the statistical distributions of features across domains, addressing simple discrepancies such as scale and distribution differences. However, this alone is insufficient for handling deeper semantic mismatches that arise in **complex domain shifts**. Semantic-wise alignment complements this by focusing on aligning features with similar semantic meanings across domains, even when their original representations differ significantly.

Semantic-wise alignment disentangles domain-invariant semantics from domain-specific noise by isolating shared factors that are consistent across domains. This process helps BRIDGE avoid relying on spurious correlations or domain-specific

---

[3] https://github.com/pytorch/pytorch
[4] https://github.com/pyg-team/pytorch_geometric

confounders, but **causal relations**. Additionally, it enforces consistency through principles of **invariant learning**, ensuring that the predictive relationships between features and labels remain stable regardless of the domain. By integrating these two alignment strategies, BRIDGE creates representations that are not only robust to domain shifts but also capable of preserving meaningful semantic relationships, enabling effective knowledge transfer and strong generalization across diverse domains.

### G.2. How to Understand the Objective of Pre-training?

The objective of pre-training in BRIDGE is to create a robust and transferable representation that bridges the gap between **heterogeneous tasks** encountered during pre-training and fine-tuning. To achieve this, a **self-supervised loss** is employed, which serves as a unifying mechanism for multi-task learning on graphs.

The self-supervised loss focuses on aligning tasks by framing them as **similarity computations** (Liu et al., 2023b). Regardless of the specific downstream task, whether it maybe node classification, link prediction, or graph classification, these tasks can be reformulated into the **shared objective** of learning embeddings through similarity maximization. By doing so, the pre-training process effectively reduces the task gap between pre-training and fine-tuning, ensuring that the representations learned during pre-training are directly applicable to a wide variety of graph tasks. This unified approach not only enhances knowledge transfer across domains and tasks but also provides a flexible framework for handling the diverse objectives inherent in graph-based learning.

### G.3. How to Understand the Alignment Risk Regularization?

The alignment risk regularization in BRIDGE is designed to **mitigate spurious correlations** and **enhance the learning of invariant dimensions** across domains. This is achieved by leveraging a **causal** intervention-inspired mechanism that systematically targets and reduces the influence of domain-specific variations in the embeddings.

Specifically, the regularization introduces **stochasticity** into the embeddings by randomly replacing the variant dimensions $K$ times, achieved by applying different projection weight matrices. For each replacement, the embeddings are perturbed to simulate domain-specific variations, and the model computes the downstream loss for each perturbed version. The regularization objective minimizes the **variance** of these $K$ times loss values. By forcing the model to remain robust under these perturbations, it effectively learns to **focus on the invariant dimensions** that are consistent across domains, while reducing reliance on domain-specific, spurious correlations. This approach ensures that the learned representations capture the essential causal factors shared across domains, improving generalization and transferability.

### G.4. How to Understand Proposition 4.2 and Proposition 4.3?

Proposition 4.2 establishes that minimizing Eq. (4) encourages the pre-trained model $f_{\widehat{\Theta}}^{\star}$ to satisfy Assumption 4.1, which defines the conditions for effective multi-domain alignment. Essentially, it avoids relying on overly strong assumptions by explicitly integrating domain alignment into the optimization process. This ensures that the model is capable of capturing domain-invariant features without requiring unrealistic assumptions about perfect alignment between source and target.

Proposition 4.3 formalizes that optimizing Eq. (4) is equivalent to minimizing the upper bound of the multi-domain semantic alignment risk defined in Eq. (2). This guarantees that the optimization process is not only aligning features dimension- and semantic-wise but also systematically reducing alignment risks. By transforming the problem into an equivalent optimization of alignment risk, it provides a more direct way to enhance generalization and robustness.

Proposition 4.2 highlights the practicality of the alignment process by avoiding overly strong assumptions, while Proposition 4.3 bridges the gap between theoretical alignment risk and its empirical optimization. Together, they demonstrate how the aligner in BRIDGE contributes to both feature alignment across domains and robustness during pre-training. The propositions solidify the theoretical foundation for BRIDGE's ability to generalize effectively in multi-domain scenarios.

### G.5. How to Understand the Selective Knowledge Assembly?

Selective knowledge assembly is a mechanism designed to bridge the gap between pre-training and downstream tasks by dynamically integrating domain-specific knowledge. During pre-training, BRIDGE aligns graph features across domains to extract invariant representations, ensuring these representations retain core knowledge while filtering out domain-specific noise. This alignment lays the foundation for selective transfer.

In downstream tasks, the lightweight Mixture of Experts (MoE) network facilitates this assembly process by **assigning**

**weights** to the pre-trained aligners based on the input graph's characteristics. Instead of uniformly applying all pre-trained knowledge, the MoE selectively activates the most relevant components, tailoring the knowledge transfer to the target domain. This targeted assembly improves generalization to unseen tasks while maintaining computational efficiency, especially in few-shot settings. By focusing only on what is essential, BRIDGE ensures the effective adaptation of multi-domain graph knowledge to diverse tasks.

### G.6. How to Understand the Downstream Classification Objective?

The downstream classification objective **unifies the pre-training and down-prompting** stages by adopting a **universal** task template based on **subgraph similarity**. This objective ensures consistency across tasks such as node and graph classification, enabling efficient knowledge transfer from pre-trained embeddings to downstream tasks.

Formally, the classification loss aims to maximize the similarity between embeddings of correctly matched subgraphs while minimizing the similarity with mismatched ones. Specifically, the model utilizes the pre-trained graph encoder and a learnable prompt to adapt the embeddings of target graphs. The classification loss is formulated as a cross-entropy over **pairwise embedding similarities**, where few-shot labeled samples serve as the supervision to guide the model in distinguishing classes. This design not only aligns with the self-supervised pre-training task but also enables effective fine-tuning under limited labeled data, ensuring robust performance even in few-shot settings.

### G.7. How to Understand the Derivation Process of Generalization Error Upper Bound?

The derivation of the generalization error upper bound quantifies the extent to which knowledge can be transferred between domains. It decomposes the target domain error into **four components**: ❶ (term $[a]$ and term $[b]$) empirical errors on source and target domains, which can be optimized through classification objectives. ❷ (term $[c]$) domain divergence, measured using the Wasserstein distance between source and target distributions. ❸ (term $[d]$) hypothesis mismatch, capturing the gap between source and target models' predictions. ❹ (Term $\mathcal{O}(\cdot)$) residual terms related to sample complexity and model capacity. To ensure smooth generalization, Lipschitz continuity is introduced for the graph encoder and classifier, bounding how much the output can change with respect to input variations.

To further constrain domain divergence, the bound is refined using **graph spectral theory**. The Wasserstein distance is related to spectral differences in adjacency matrices, leading to a spectral regularization term that stabilizes knowledge transfer. This regularizer ensures that down-prompting effectively preserves task-relevant features while mitigating overfitting to domain-specific noise. By incorporating this bound into the optimization process, BRIDGE explicitly balances knowledge transfer and generalization, making it theoretically sound and practically efficient for adapting to new domains.

### G.8. How to Understand Proposition 4.6?

Proposition 4.6 provides a lower bound for the Lipschitz constant $L_f$, which measures how much the graph encoder's output changes in response to variations in input graphs. A smaller $L_f$ suggests a smoother model that generalizes better across domains, while a larger $L_f$ implies that small input differences can lead to large output variations, potentially reducing transferability. The bound is derived by linking $L_f$ to the **spectral properties of graphs**, specifically the eigenvalues of their adjacency matrices. Since graphs from different domains have different structures, their adjacency matrices vary, and the optimal transport (Wasserstein) distance between them helps quantify these differences.

Mathematically, the bound shows that $L_f$ depends on both the **spectral gap** (differences in eigenvalues) and the **structural distance** between source and target graphs. If two graphs have highly similar adjacency matrices (*i.e.*, small spectral differences), the bound indicates that the knowledge transferability is high, leading to better generalization. Conversely, if the graphs are structurally different, the transferability is limited, and adaptation becomes harder. This result is crucial for BRIDGE because it provides a theoretical guarantee that minimizing spectral differences during training enhances the model's ability to generalize across domains.

### G.9. How to Understand How the Spectral Regularizer Takes Effect?

The spectral regularizer in BRIDGE ensures that knowledge transfer remains stable by controlling **how much the graph representations change** when adapting from source to target domains. It works by enforcing **smoothness** in the spectral space, meaning that graph embeddings should not fluctuate too much between adjacent frequencies. This is done by minimizing deviations in spectral components of node features and their transformed embeddings, reducing the risk of

overfitting to domain-specific noise. The regularizer acts as a constraint during fine-tuning, preventing drastic shifts in feature distributions while ensuring the model retains transferable knowledge.

Mathematically, the regularizer is formulated based on **spectral differences** in graph signals, ensuring that variations in embeddings align with the inherent structure of the target domain. By penalizing excessive spectral changes, it helps maintain consistency in feature representations, leading to better generalization. In practice, this means that BRIDGE can adapt to new domains while preserving key structural properties, ensuring efficient and stable knowledge transfer even with limited labeled data.

**Notably,** if pre-training and fine-tuning are **jointly optimized** rather than conducted separately, the spectral regularizer is expected to have an even greater impact. In this setting, the model can dynamically adjust feature alignment while simultaneously learning a more generalizable representation space. This synergy allows the regularizer to directly influence both phases, reinforcing domain-invariant structures early in training and further stabilizing adaptation.

### G.10. How to Understand Why LLM Can Enhance Knowledge Transfer?

LLMs enhance knowledge transfer by improving feature alignment across different domains through **semantic enrichment**. In BRIDGE, node features are often derived from raw text via feature engineering, which may lead to information loss. By leveraging LLMs to generate textual descriptions and encode them into numerical representations, BRIDGE integrates **richer semantic information** into its graph embeddings. This augmented representation bridges the gap between structurally different graphs by introducing a **shared semantic space**, making it easier for the model to generalize across domains.

Specifically, the experiment constructs node representations by concatenating original graph features with LLM-generated textual embeddings. This hybrid approach allows the model to retain domain-specific graph structures while benefiting from the broader contextual understanding provided by LLMs. As a result, the cross-domain adaptation process becomes more effective, as semantically similar nodes from different domains are better aligned. This method is particularly useful in scenarios where domain shifts are significant, enabling BRIDGE to achieve stronger generalization with minimal supervision.

## H. Additional Related Work

### H.1. Multi-Domain Graph Pre-training

Pre-training methods on graphs (Liu et al., 2022; Hu et al., 2020a; Xie et al., 2022) capitalize on the intrinsic properties of graph structures and feature semantics via self-supervised learning. These approaches leverage optimization objectives such as node masking, edge prediction, and contrastive learning to learn transferable representations from massive graph data, such as GCC (Qiu et al., 2020), GPT-GNN (Hu et al., 2020b), GPPT (Sun et al., 2022), L2P-GNN (Lu et al., 2021), APT (Xu et al., 2023), CPT-HG (Jiang et al., 2021), Att-HGNN (Yang et al., 2022), PT-DGNN (Chen et al., 2022), W2PGNN (Cao et al., 2023), $\pi$-GNN[3] (Yin et al., 2023), *etc.* Pre-trained knowledge can be transferred to downstream domains and tasks through parameter fine-tuning or prompting. However, these methods typically assume that the pre-training and downstream graphs originate from the same or very similar domains. For instance, they may involve different subgraphs of a large graph (Huang et al., 2022) or a collection of similar graphs within the same field (Zhang et al., 2021). Consequently, these methods struggle to generalize across multi-domain graphs due to domain-specific biases inherent in the pre-training process.

Another line of research in cross-domain learning addresses the challenge of transferring knowledge from a single source domain to a different target domain, such as GCOPE (Zhao et al., 2024), OMOG (Liu et al., 2024b), PGPRec (Yi et al., 2023), UniAug (Tang et al., 2024c), UniGraph (He & Hooi, 2024), *etc.* These studies focus on exploiting domain-invariant representations to bridge the gap between disparate domains. While effective for single-source to single-target scenarios, they fall short in utilizing comprehensive multi-domain knowledge. Additionally, many of these methods are tailored to specific tasks or domains and often require domain expertise, which limits their applicability to diverse domains and tasks. Towards addressing multi-domain graph pre-training, methods such as OFA (Liu et al., 2024b) and HiGPT (Tang et al., 2024b) employ large language models to derive and align node features from different domains through the medium of natural language. While innovative, these methods are constrained to text-attributed graphs (Wang et al., 2024; Tan et al., 2024), thereby limiting their applicability to a broader range of more general graph types. For non-text-attributed graphs, GCOPE (Zhao et al., 2024) introduces domain-specific, interconnecting virtual nodes to link across domains. Though it facilitates some level of connection, they do not explicitly align node features, thereby failing to overcome domain conflicts effectively. MDGPT (Yu et al., 2024c) proposes a set of domain tokens to explicitly align multi-domain features for

synergistic pre-training. However, it simply utilizes SVD to align features dimension-wise without considering semantics.

Unlike existing multi-domain graph pre-training methods that often struggle with domain-specific biases or lack explicit feature alignment, BRIDGE introduces a principled approach by aligning multi-domain features with a set of domain-invariant aligners. Additionally, BRIDGE offers a theoretical framework that quantifies transferable knowledge through an optimizable upper bound on the generalization error. This combination of alignment, selective transfer, and theoretical grounding leads to improved performance across various tasks, setting it apart from prior work in the domain.

## H.2. Graph Fine-tuning and Prompt Learning

Fine-tuning pre-trained graph models for specific downstream tasks has become a widely adopted approach to adapting general representations to particular applications. Traditional fine-tuning techniques involve modifying the entire graph model or a subset of its parameters to optimize performance on the target task, such as GTOT (Zhang et al., 2022), G-Tuning (Sun et al., 2024), S2PGNN (Zhili et al., 2024), IGAP (Yan et al., 2024), WalkLM (Tan et al., 2024), NEAT (Ying et al., 2024), Bridge-Tune (Huang et al., 2024b), GSP (Jiang et al., 2024), GRAPH-BERT (Zhang et al., 2020), *etc.* While effective, these methods can be computationally expensive and prone to overfitting, especially when labeled data is limited. To mitigate these issues, parameter-efficient fine-tuning (PEFT) methods have emerged, adjusting only a small portion of the model's parameters (Zhu et al., 2024a; Tian et al., 2024; Gui et al., 2024; Xue et al., 2023) or introducing lightweight modules (Lin et al., 2024; Zhu et al., 2024b; Yang et al., 2023; 2024) to adapt the model to new tasks without extensive retraining. PEFT-based methods significantly reduce computational cost and the risk of overfitting while maintaining strong performance on downstream tasks. This allows for more efficient adaptation to new domains, especially in scenarios with limited labeled data. However, these methods still face challenges in handling complex multi-domain scenarios, where domain-specific biases and task heterogeneity can affect their effectiveness.

Prompt learning has emerged as an alternative approach, initially popularized in natural language processing (Liu et al., 2023a; Zhou et al., 2022b;a; Du et al., 2022). For graph learning, prompt-based methods aim to guide the behavior of pre-trained models with minimal parameter updates. Techniques such as graph prompt tuning (Sun et al., 2022; Zhu et al., 2023; Shirkavand & Huang, 2023; Zhang et al., 2024) involve designing task-specific prompts that interact with pre-trained embeddings to achieve the desired outcomes. These methods provide a more parameter-efficient adaptation strategy, reducing the risk of overfitting and improving generalization across diverse tasks. Recent advancements in graph prompt learning have introduced various strategies to improve adaptability and performance, such as GPPT (Sun et al., 2022), All in One (Sun et al., 2023), GraphPrompt (Liu et al., 2023b), Self-Pro (Gong et al., 2023), SGL-PT (Zhu et al., 2023), GPF-Plus (Fang et al., 2024), PRODIGY (Huang et al., 2024a), MDGPT (Yu et al., 2024c), HGPROMPT (Yu et al., 2024a), ProG (Zi et al., 2024), *etc.* These methods integrate learnable prompts into the graph neural networks, enabling the model to focus on task-relevant features or structural semantics, which captures key patterns in the downstream graph data, facilitating transfer learning across different graph tasks, enhancing the model's ability to handle complex graph structures.

Despite these advancements, existing prompt learning methods for graphs are often limited in their ability to handle multi-domain scenarios effectively. Many approaches are designed with a focus on single-domain tasks, requiring domain-specific prompt engineering that restricts their scalability and generalization. Furthermore, there is a lack of theoretical insights into how and why prompt learning works in the graph domain, which constrains the development of more robust and principled methods. BRIDGE, however, not only integrates multi-domain graph feature alignment but provides a theoretical foundation for knowledge transfer, addressing these challenges and offering a more scalable solution for multi-domain graph learning.

## H.3. Towards Graph Foundation Models

Graph Foundation Models (GFMs) extend the idea of foundation models, which have shown remarkable success in fields, like natural language processing (Myers et al., 2024; Bai et al., 2024) and computer vision (Singh et al., 2022; Wang et al., 2023), to graphs (Mao et al., 2024b; Shi et al., 2024a; Zi et al., 2024; Guo et al., 2025b). The concept of GFMs revolves around pre-training models on large-scale, diverse graphs to learn generalizable representations that can then be fine-tuned for a wide variety of downstream tasks (Shi et al., 2024b; Bommasani et al., 2021). These models aim to combine the flexibility and generalization ability seen in large language models with structural precision required for graph-based tasks.

While traditional graph deep models have made significant progress in tasks like node classification, link prediction, and graph classification, their generalization across different domains remains limited. Pre-training GNNs on domain-specific graphs has been a common approach, but such methods often face challenges in transferring learned representations across diverse domains due to domain-specific biases and the heterogeneous nature of graph data. This limitation motivates the

exploration of GFMs, which aim to learn domain-invariant representations through multi-domain pre-training. Existing multi-domain graph pre-training methods, such as GCOPE (Zhao et al., 2024) and MDGPT (Liu et al., 2024b), have attempted to bridge the gap by aligning features from different domains. These approaches focus on aligning node and edge features across domains, but they are typically restricted to text-attributed graphs or rely on shallow domain alignment techniques. Methods like OFA (Liu et al., 2024b) and TSGFM (Chen et al., 2024b) leverage language models to align features via language descriptions, yet they still struggle with non-textual graphs and fail to fully capture semantic similarities.

GFMs, however, aim to provide a more unified approach by leveraging large-scale pre-training on graph data, followed by task-specific adaptation. These models are expected to exhibit both emergence and homogenization (Shi et al., 2024a), similar to LLMs, enabling them to generalize across tasks and domains while maintaining their ability to perform graph-specific reasoning. By aligning features across domains in a principled way and using techniques like contrastive pre-training or prompt learning, GFMs have the potential to handle diverse graph data types and tasks more effectively. A growing line of inquiry is into the theoretical underpinnings of GFMs. Some works have begun to explore the potential existence of scaling laws in graph models (Ma et al., 2024; Tang et al., 2024c; Chen et al., 2024a; Liu et al., 2024c), analogous to those found in large language models (Kaplan et al., 2020). These studies hypothesize that increasing the scale of graph models may yield emergent properties, such as enhanced transferability and generalization across tasks. However, empirical validation is still limited, and the theoretical foundation for scaling GFMs remains underdeveloped.

Several key challenges remain for GFMs. First, the heterogeneous nature of graph data ranging from node features to edge types poses significant difficulties for creating unified representations. Second, the scalability of GFMs remains a concern, as large-scale graph data requires substantial computational resources for pre-training. Third, while prompt learning techniques show promise for fine-tuning pre-trained graph models, their application across multiple domains and tasks is still an open challenge. Overall, GFMs present an exciting opportunity to revolutionize graph learning by combining the strengths of pre-trained models with the precision of graph-based reasoning.

Our work, BRIDGE, addresses these gaps by aligning multi-domain graph features during pre-training and using a Mixture of Experts (MoE) network for efficient prompting. Moreover, we provide a theoretical upper bound on the generalization error, offering new insights into the amount of knowledge that can be transferred across domains. This foundational work paves the way for more robust and scalable GFMs.

## I. Limitations

While BRIDGE demonstrates notable advancements in multi-domain graph pre-training and prompt learning, certain areas remain open for further exploration to maximize its utility and versatility towards Graph Foundation Models (GFMs).

**Evaluation on Wider Domains.** The current evaluation focuses on well-known benchmark datasets across three domains (academic, e-commerce, and social networks). However, niche domains, such as chemical reaction networks (where nodes represent molecules and edges denote reactions) or financial fraud detection graphs (with complex temporal and hierarchical patterns), may pose unique challenges due to their specialized data distributions. For example, highly sparse or asymmetric adjacency matrices prevalent in these graphs may affect feature alignment or generalization capabilities. Additionally, multimodal graphs, such as those combining image features with graph structure (*e.g.*, protein folding graphs with 3D spatial data), remain unexplored.

**Assumptions on Feature Alignment.** BRIDGE relies on domain-invariant aligners to unify graph features across domains. However, this assumes that such alignments can adequately capture shared characteristics across all graphs. In graphs with extreme heterogeneity, such as heterogeneous information networks (HINs) where nodes and edges have diverse types (*e.g.*, author-paper-venue graphs), this assumption may not always hold. Similarly, in hypergraphs, where edges connect multiple nodes, or dynamic graphs with time-evolving structures, fixed aligners may struggle to generalize effectively. Developing adaptive or context-aware aligners could mitigate these issues by dynamically reconfiguring based on the data.

**Hyperparameter Exploration.** The sensitivity analysis indicates that BRIDGE's performance depends on hyperparameters to some extent, such as trade-off weights for the spectral regularizer and MoE components. Extending this exploration to cover interactions between hyperparameters and task-specific factors (*e.g.*, dataset size, graph sparsity) could yield practical guidelines. Automated hyperparameter optimization techniques, such as Bayesian optimization or reinforcement learning, might also reduce manual tuning efforts.

**Impact of Fine-tuning Epochs.** BRIDGE's few-shot fine-tuning demonstrates strong results with limited labeled data

and epochs. However, its performance under prolonged fine-tuning remains unexplored. For example, understanding the trade-offs between overfitting and incremental gains with additional epochs on larger labeled datasets would provide practical guidance for deploying BRIDGE in scenarios with varied data availability.

## J. Future Directions

Building on the advancements presented in this work, several promising directions can further enhance and expand the impact of BRIDGE.

**Enhancing Generalization for Extreme Few-Shot Scenarios.** Although BRIDGE demonstrates strong performance in one-shot and five-shot tasks, real-world scenarios often involve domains where labeled data is either completely unavailable (zero-shot) or extremely sparse (*e.g.*, fewer than 5 samples per class). Future research could explore unsupervised prompt initialization techniques that leverage unlabelled target domain data. Additionally, meta-learning approaches could be integrated to pre-train models with better generalization capabilities across unseen domains, enabling more effective transfer with minimal supervision.

**Integration with Graph Agents.** Future research could explore integrating BRIDGE with graph-based agents that leverage graph-structured data for decision-making in interactive environments. For example, combining BRIDGE with reinforcement learning or multi-agent systems could enable applications such as autonomous traffic management, supply chain optimization, or real-time network monitoring. This would involve adapting BRIDGE's pre-training and prompt learning mechanisms to dynamically update graph representations based on agent interactions and feedback. Additionally, policy-guided graph prompting could be introduced to optimize agents' actions by focusing on domain-relevant graph substructures, enhancing both efficiency and effectiveness in graph-driven environments.

**Extending to Dynamic and Temporal Graphs.** Real-world graphs, such as social networks, traffic systems, and financial transactions, often evolve over time. Future extensions of BRIDGE could incorporate temporal graph neural networks (*e.g.*, temporal attention mechanisms) to capture dynamic relationships. Designing time-sensitive feature aligners that adapt to evolving graph structures and align temporal features across domains would further enhance the framework's utility for applications like anomaly detection or forecasting.

**Incorporating Multi-Modal Graph Data.** Many practical applications involve graphs enriched with multi-modal data, such as images, textual descriptions, or sensor readings. Future research could focus on joint feature extraction mechanisms that integrate these modalities into the BRIDGE framework. For example, leveraging pre-trained language or vision models to embed auxiliary information and align it with graph features could improve performance on tasks requiring a holistic understanding of entities and their relationships, such as biomedical knowledge graphs or e-commerce recommendation systems.

**Exploration of Domain-Specific Variations.** Although BRIDGE aligns features across domains, certain specialized graphs, such as biological pathways, chemical reaction networks, or transportation systems, have unique properties that may benefit from tailored techniques. Future research could investigate domain-specific pre-training objectives, such as link prediction for protein-protein interaction graphs or path optimization for logistics networks. Additionally, customizable aligners that leverage domain knowledge, such as hierarchical clustering or molecular graph fingerprints, could improve transferability in these specialized contexts.

**Expanding the Range of Downstream Tasks.** While this study focuses on node and graph classification tasks, BRIDGE has the potential to address a broader range of graph-related problems. Future work could adapt the framework for link prediction (*e.g.*, for knowledge graph completion or social network analysis), graph generation (*e.g.*, molecule generation in drug discovery), or graph clustering (*e.g.*, community detection in social graphs). This expansion would require task-specific prompts and evaluation protocols to demonstrate adaptability across diverse use cases.

