# OpenReview forum: "How Much Can Transfer? BRIDGE: Bounded Multi-Domain Graph Foundation Model with Generalization Guarantees"
_ICML.cc/2025/Conference — ICML 2025 poster_

### Official Review · Reviewer_Efax · 2025-03-02

**Overall Recommendation:** 3

**Summary:**

The authors argue that while the "pretrain-then-prompt" framework has been extensively studied in other fields, its application in the graph domain remains underexplored, particularly from a theoretical perspective.
To address this gap, this paper introduces BRIDGE, a pretraining and prompt learning framework designed for multi-domain graph data, incorporating a theoretically driven regularization loss.
During the pre-training phase, BRIDGE performs dimension-wise alignment and semantic-wise feature alignment, inspired by the Independent Causal Mechanism.
In the fine-tuning phase, alongside the downstream task loss, BRIDGE introduces an aligner routing network that selectively assembles the pre-trained source domain aligner and proposes a spectral regularizer to minimize the upper bound of generalization error.
They conduct experiments in both cross-dataset and cross-domain settings to demonstrate the effectiveness of the proposed framework.

## update after rebuttal
As I mentioned in the 'Rebuttal Comment', my concerns have been well addressed; therefore, I have updated my score.

**Claims And Evidence:**

I agree with their claim that prompt learning is under-studied in the graph domain from both practical and theoretical perspectives.
Moreover, each module in BRIDGE is intuitive, and the proposed regularizer is supported by a strong theoretical foundation, which is a significant advantage of this work.

**Essential References Not Discussed:**

N/A

**Experimental Designs Or Analyses:**

The experimental design is valid, but an ablation analysis comparing the model with and without the prompt is needed.

**Methods And Evaluation Criteria:**

The authors' evaluation criteria for cross-dataset and cross-domain settings are valid. However, I suggest incorporating larger datasets (e.g., ogbn-arxiv) for both the source and target domains. This is important for two reasons: 1) assessing whether the model's performance improves as the amount of training data increases, and 2) evaluating the model's ability to effectively perform inference on large datasets.

Additionally, while the paper does not report training and inference times, I am concerned that calculating the spectral regularizer could be time-consuming for large datasets.

**Other Comments Or Suggestions:**

N/A

**Other Strengths And Weaknesses:**

A major concern with this paper is the low performance observed in the few-shot case (m=2,3) in Figures 2, 3, E.1, E.3, and E.4. Since pre-training is particularly beneficial when supervision is limited, this represents a critical scenario for the model's effectiveness.

**Questions For Authors:**

1) It would be helpful if the authors could clarify why BRIDGE performs well in the zero and one-shot settings but not in the two or three-shot cases.
2) I would like to know the training and inference times.
3) Please consider including larger datasets for both the source and target domains.
4) Please show the ablation studies regarding the prompt
5) Please show how the learned 'S' gains significantly more knowledge from the same domain compared to cross-domain sources to validate the selective knowledge assembly.

For checking the validity of selective knowledge assembly, showing the learned 'S' that it is truly gains high knowledge from same domain than cross domains.

SelectiveKnowledgeAssembly

If all of these concerns are addressed, I would be happy to reconsider my score.

**Relation To Broader Scientific Literature:**

I believe the key contribution of this paper is the proposed theoretical analysis bounding knowledge transfer generation, along with the derived regularization loss, which can serve as a foundation for future work in this field.

**Theoretical Claims:**

I did not find any issues with the theoretical claims.

---

> ### Author Rebuttal · Authors · 2025-03-30
>
> We sincerely thank for the positive evaluation regarding its theoretical foundation for constructing GFMs. Response:
>
> **Q1: Why work well in zero-/one-shot but not in two-/three-shot.**
>
> **A1:** Thank you for the insightful question.
>
> - **0-shot limited for text-attributed graphs:** Knowledge transfer based solely on structural information is inherently challenging in zero-shot scenarios. BRIDGE primarily targets **text-free graphs**, with experiments (RQ5) integrating an LLM as an enhancer to show effectiveness when textual semantics are available.
> - **Why excels at 1-shot but not 2-/3-shot:** BRIDGE’s MoE-based selective knowledge assembly effectively utilizes precise domain-invariant knowledge, providing a principled advantage at **extremely sparse supervision** (1-shot). At 2-/3-shot, the MoE routing faces **increased uncertainty**, while simpler alignment strategies (e.g., ProNoG, GCOPE, MDGPT) temporarily gain an advantage with limited additional supervision. When supervision further increases (≥5-shot), BRIDGE’s **spectral regularizer dominates** by constraining generalization error, restoring its superior performance.
>
> ---
>
> **Q2: The training and inference time.**
>
> **A2:** Thanks for bringing up the important issue.
>
> - **Pre-training Phase:** We adopt an **early-stopping strategy** based on **loss**, so the maximum training time depends on a preset maximum of 1000 epochs. No explicit inference is conducted during pre-training. Results (s):
>
> | Source | Time / Epoch | Total Time |
> | --- | --- | --- |
> | CS, PM, Ph, Com | 1.67 | 1653.57 |
> | CR, PM, Ph, Com | 1.53 | 1624.01 |
> | CR, CS, Ph, Com | 0.92 | 938.38 |
> | CR, CS, PM, Com | 1.20 | 1147.02 |
> | CR, CS, PM, Ph | 1.31 | 1698.75 |
> - **Fine-tuning Phase:**  We report time per epoch, total fine-tuning time, and inference time separately. The spectral regularizer (highlighted as potentially time-consuming) is only computed during fine-tuning loss, particularly at small shot numbers ($m$). To reduce computational costs in eigenvalue decomposition, we implemented approximate techniques (lines 969–989). Results (5-shot Node Cls., s):
>
> | Target | Time / Epoch  | Inference Time | Total Time |
> | --- | --- | --- | --- |
> | Cora | 0.13 | 0.02 | 142.37 |
> | CiteSeer | 0.18 | 0.02 | 183.15 |
> | PubMed | 0.82 | 0.10 | 1002.62 |
> | Photo | 0.36 | 0.09 | 384.26 |
> | Computers | 0.69 | 0.05 | 755.39 |
> | Reddit | 6.62 | 2.30 | 579.16 |
>
> ---
>
> **Q3: Including larger datasets (e.g., ogbn-arxiv) for both source and target domains.**
>
> **A3: Thanks for this valuable suggestion.**
>
> - **In source domain:** Following your advice, we have now included **ogbn-arXiv** as an additional dataset for pre-training. Results (Acc. %):
>
> | S → T | 1-shot Node Cls. | 1-shot Graph Cls. |
> | --- | --- | --- |
> | CS, Ph, arXiv → Cora (cross-dataset) | MDGPT: 37.21, **BRIDGE: 43.62** | MDGPT: 45.35, **BRIDGE: 48.20** |
> | CS, Ph, arXiv → Reddit (cross-domain) | MDGPT: 58.82, **BRIDGE: 64.16** | MDGPT: 68.22, **BRIDGE: 71.74** |
> - **In target domain:** We have already validated BRIDGE on **Reddit**. Notably, Reddit has **232,965** nodes and **114,615,892** edges (Table D.1), substantially **larger than ogbn-arXiv** (169,343 nodes and 1,157,799 edges). Therefore, the strong performance on Reddit sufficiently demonstrates BRIDGE’s capability for knowledge transfer on large-scale graphs.
>
> ---
>
> **Q4: The ablation studies regarding the prompt.**
>
> **A4:** Thank you for the insightful suggestion. We have conducted an ablation concerning the graph prompts for fine-tuning. Results (Acc. %):
>
> | T: CiteSeer | 1-shot Node Cls. | 1-shot Graph Cls. | 5-shot Node Cls. | 5-shot Graph Cls. |
> | --- | --- | --- | --- | --- |
> | BRIDGE | 42.18 | 50.49 | 60.20 | 59.80 |
> | BRIDGE (w/o prompt) | 35.23 | 34.12 | 50.34 | 49.17 |
>
> Note that, prompts $\boldsymbol{\mathcal{P}}$ and routing weights $\mathbf{S}$ are the only tunable parameters. Without prompts, only $\mathbf{S}$ is trained, significantly restricting fine-tuning flexibility and degrading performance, highlighting prompts' necessity in BRIDGE.
>
> ---
>
> **Q5: How does 'S' favors same-domain over cross-domain knowledge for selective assembly?**
>
> **A5:** We calculate the learned assignment weights for source experts from the same domain versus cross-domain sources. Results (1-shot, Node Cls.):
>
> | S | T | Weights |
> | --- | --- | --- |
> | CS, PM, Ph, Com | CR | [0.5742, 0.2950, 0.0486, 0.0822] |
> | CR, CS, PM, Ph, Com | Reddit | [0.3001, 0.2855, 0.0634, 0.1254, 0.2256] |
>
> **Analysis:** For **Cora** (CR),  **Academic** experts (CS, PM) receive significantly higher weights (0.5742, 0.2950) compared to **E-Commerce** sources (Ph, Com: 0.0486, 0.0822). For **Reddit**, the structurally similar **E-Commerce** expert (Computers, 0.2256) and **Academic** experts (CR, 0.3001; CS, 0.2855) dominate. These results validate that BRIDGE effectively favors knowledge transfer from same-domain or structurally similar-domain experts, supporting our selective assembly design.

---

> > ### Comment · Reviewer_Efax · 2025-04-03
> >
> > Thank you for your thorough response to my concerns. In particular, the detailed experiments addressing question 5 have clarified how the proposed models effectively integrate cross-domain knowledge. I hope these experiments will be included if the paper is published. Based on this, I will be raising my score.

---

> > > ### Author Response · Authors · 2025-04-03
> > >
> > > Thank you for your positive feedback and for raising your score! We’ll make sure to include the experiments from these questions in the camera-ready version.

---

### Official Review · Reviewer_QW95 · 2025-03-10

**Overall Recommendation:** 4

**Summary:**

BRIDGE is a bounded multi-domain graph pre-training and prompt learning framework that enhances knowledge transfer in graph foundation models. It integrates domain-invariant alignment, a lightweight MoE for selective knowledge transfer, and a graph spectral-based generalization bound. Experiments show state-of-the-art performance in few-shot classification across datasets and domains, with further improvements when combined with LLMs. I consider this paper as a new bench for more generalizable and trustworthy GFMs, as existing works on GFMs lacks theoretical analysis of cross-domain transfer of pre-trained knowledge, to the best of my knowledge.

**Claims And Evidence:**

The claims made in this paper are generally supported by clear empirical results and theoretical analysis, while some aspects could be more thoroughly validated.

- The well-supported claims include the effectiveness of BRIDGE in few-shot learning, theoretical justification for multi-domain knowledge transferability, effectiveness of key modules or components, etc.

- The potentially problematic claim probably lies in the scalability and computational efficiency. BRIDGE claims superior efficiency compared with other baselines in down-prompting, but comparisons only focus on fine-tuning convergence speed rather than end-to-end training costs. But I believe this is not a major issue, as GFMs do not place a strong emphasis on the training cost of the pre-training phase but rather focus on whether they can achieve good performance quickly during the fine-tuning stage.

**Essential References Not Discussed:**

I believe that the related work section and other parts of the paper have included the necessary references, allowing me to understand the current state of research in the relevant field and to objectively evaluate the contribution of this paper.

**Experimental Designs Or Analyses:**

I reviewed the experimental design and analyses in the paper, focusing on the few-shot learning setup, baseline comparisons, ablation studies, and efficiency evaluations.

**Pros:**
- The few-shot experimental design is well-suited for evaluating multi-domain knowledge transfer under the GFM settings.
- Baseline selection is comprehensive and covers multiple paradigm related to the paper scope.
- Ablation studies effectively isolate key components, and independently analyze their contributions, respectively.
- The LLM integration experiment for the initial feature enhancement is novel and promising.

**Cons:**
- Missing comparisons to general domain adaptation baselines, not only resitricted to graph learning.
- No full computational cost analysis, including training and inference time, memory for both pre-training and fine-tuning.
- LLM feature alignment lacks qualitative validation.

**Methods And Evaluation Criteria:**

I agree that the proposed methods and evaluation criteria align well with multi-domain graph transfer learning under the GFM settings. The “pretrain-then-prompt” paradigm, domain-invariant alignment, and spectral-based generalization bound basically seems to effectively address more generalized and trustworthy knowledge transfer in multi-domain GFMs. The evaluation on few-shot node and graph classification with cross-dataset and cross-domain settings is promising, and comparisons with 15 baselines strengthen credibility. The only concern lies in the computational efficiency are not fully analyzed. A more comprehensive evaluation on larger real-world graphs and detailed cost analysis would further validate BRIDGE’s practicality.

**Other Comments Or Suggestions:**

**Typos and Grammars:**
- The symbol “≜” (Eq. (12) and Eq. (13)) is not commonly used.
- “The graph learner compound of the encoder …” should be “The graph learner is composed of the encoder …”
- “We consider as benefiting from the spectral regularizer …” should be “We attribute this improvement to the spectral regularizer …”
- “between domains introduces graph pre-training variations …” should be “between domains introduces variational inference in graph pre-training …”

**Other Strengths And Weaknesses:**

**Strengths:**
- The pretrain-then-prompt approach for multi-domain graph learning is an promising extension of prior works that primarily focused on either pretraining or prompting separately.
- The graph spectral-based generalization bound provides a new theoretical perspective on quantifying transferable knowledge, which is not commonly explored in existing works.
- The integration of domain-invariant alignment with a lightweight MoE for selective knowledge transfer is a novel contribution that improves adaptability across domains.

**Weakness:**
- The paper does not analyze how minimizing the bound directly affects transfer performance, making it unclear how well the bound correlates with real-world generalization.
- The MoE mechanism is a key component of BRIDGE, facilitating selective knowledge transfer, but the paper does not compare different MoE architectures. It is unclear why this specific MoE design was chosen over alternatives such as soft attention-based routing or adaptive gating mechanisms.
- Multi-domain learning can suffer from negative transfer, where knowledge from one domain hurts rather than helps performance in another. The paper assumes all knowledge is beneficial, but does not analyze whether certain source domains degrade performance in the target domain.

**Questions For Authors:**

- Can you provide an empirical analysis or more easy-to-follow understanding demonstrating whether minimizing the generalization upper bound directly improves transfer performance?
- Did you conduct any experiments to quantify negative transfer? Are there cases where knowledge from one domain degrades performance in another?
- How does BRIDGE compare against full fine-tuning methods (e.g., LoRA, Adapter-based fine-tuning) in terms of performance vs. computational cost?
- Other questions, please refer to the aforementioned weaknesses.

**Relation To Broader Scientific Literature:**

The key contributions of this paper build upon and extend several existing areas of multi-domain graph pre-training, prompt learning for graphs, generalization bounds, and LLMs for GFM understanding.

- multi-domain graph pre-training: BRIDGE introduces a domain-invariant alignment strategy that aligns both dimensions and semantics across domains, addressing the cross-domain generalization challenge more explicitly than prior work. Unlike prior works relying on textual metadata for alignment, BRIDGE is agnostic to text attributes, making it applicable to a wider range of graphs.

- prompt learning for graphs: BRIDGE extends prompt learning to multi-domain graph settings, introducing a lightweight MoE network to selectively transfer domain-specific knowledge, which prior graph prompt methods lack. Unlike GraphPrompt and GraphPrompt+, which manually design prompts, BRIDGE learns self-supervised prompt initializations, making it more adaptive.

- generalization bounds: BRIDGE integrates generalization bounds into its optimization, providing an explicit, optimizable spectral regularizer that controls transferability constraints. Unlike past works that assume smooth transfer functions, BRIDGE derives an explicit bound in the graph spectral space, which is novel in the filed of GFMs.

- LLMs for graph understanding: BRIDGE incorporates LLM-generated node descriptions as a feature alignment mechanism, demonstrating improved multi-domain adaptation. Unlike MDGPT, BRIDGE’s graph-specific prompting enables more efficient knowledge transfer without relying on extensive LLM augmentation.

**Theoretical Claims:**

I have reviewed the theoretical claims and proofs presented in the paper, particularly focusing on the generalization error bound derivation and its spectral regularization component.

- generalization error upper bound: an optimizable upper bound for the generalization error built upon the graph spectral theory and Lipschitz continuity. While the bound is theoretically sound, the paper lacks in-depth theoretical validation of how tight this bound is. Some additional visualization or sensitivity analysis of the bound’s impact on performance would strengthen the authors’ claim.

- Lipschitz constant lower bound: relies on the graph eigenvalue decomposition to express the Lipschitz constant in a spectral form. The bound assumes optimal transport-based alignment of graph structures on both sides, but the practical effectiveness of this assumption is somewhat unclear. More discussion on whether real-world graphs under GFM settings satisfy these assumptions would be beneficial.

---

> ### Author Rebuttal · Authors · 2025-03-30
>
> Thank you for your thoughtful review and positive feedback. We appreciate your recognition of BRIDGE’s contributions and address your comments and suggestions point by point below.
>
> **Q1: The scalability and computational efficiency.**
>
> **A1:** Thank you. For scalability, we address this in **Reviewer Efax (Q3)** with new results on **ogbn-arXiv** (source) and analysis on **Reddit** (target). For efficiency, see **Reviewer Efax (Q2)**, where we provide end-to-end training and fine-tuning time comparisons.
>
> ---
>
> **Q2:  Theoretical validation of the generalization error bound tightness.**
>
> **A2:** The bound tightness primarily depends on: **(1)** how accurately the Wasserstein distance $d_W$ captures true domain divergence, and **(2)** how precisely the Lipschitz constants ($L_f$, $L_g$) reflect actual smoothness of the learned graph representations. Lower $d_W$, tighter estimates of $L_f$ and $L_g$ yield a tighter bound, more closely approximating empirical generalization errors. We will explicitly evaluate these factors empirically in future revisions.
>
> ---
>
> **Q3: The practical impact of the generalization bound.**
>
> **A3:** The generalization bound indicates how effectively BRIDGE transfers knowledge across domains. A smaller bound means better transfer (e.g., between similar graphs like Cora and PubMed), whereas a larger bound signals difficulty (e.g., between citation networks and social graphs), guiding practical adjustments like additional fine-tuning.
>
> ---
>
> **Q4: How minimizing the bound directly affects transfer performance.**
>
> **A4:** Directly minimizing the bound improves BRIDGE’s transfer by explicitly constraining domain divergence ($d_W$) and smoothness ($L_f$, $L_g$). Practically, this means representations from different domains become closer and smoother, significantly enhancing generalization to unseen graphs. Experiments (Section 5.3, Figure 4) confirm that optimizing this bound via the spectral regularizer consistently leads to higher accuracy on downstream tasks.
>
> ---
>
> **Q5: Whether real-world graphs satisfy optimal transport-based alignment assumption.**
>
> **A5:** The optimal transport-based alignment assumption has been widely adopted in prior studies, and our work follows this common practice. The underlying reason is that real-world graphs often **share intrinsic structural similarities** (e.g., similar community or connectivity patterns across social or citation graphs), allowing optimal transport methods to meaningfully measure domain divergence.
>
> ---
>
> **Q6: Missing comparisons to general domain adaptation baselines.**
>
> **A6:** Since BRIDGE targets **graph-structured data**, we compare with strong **graph-specific** adaptation baselines (e.g., GCOPE, MDGPT). General domain adaptation methods lack structure-aware design, making them less relevant for fair comparison.
>
> ---
>
> **Q7: LLM feature alignment lacks qualitative validation.**
>
> **A7:** Thank you for this valuable comment. We clarify that the LLM-enhanced alignment is **not part of BRIDGE's core contributions**. Rather, it serves as an **auxiliary experiment** to demonstrate BRIDGE’s applicability to text-attributed graphs, highlighting BRIDGE’s broad adaptability. We agree qualitative validation would further support this applicability and will consider it in future extensions.
>
> ---
>
> **Q8: Statistical significance tests for ablations.**
>
> **A8:** Thank you for raising this point. Explicit statistical significance tests (e.g., t-tests) for ablation studies are relatively uncommon for papers in AI research communities. Ablations generally aim to intuitively illustrate each component’s individual contribution rather than establish statistical differences rigorously. Consistent with common practice, our ablations provide results averaged over multiple runs (with std), clearly showing reliable performance differences.
>
> ---
>
> **Q9: Compare different MoE architectures.**
>
> **A9:** Our MoE is a lightweight internal module of BRIDGE. Rather than comparing MoE variants in isolation, we focus on end-to-end comparisons with full graph adaptation methods (e.g., MDGPT, GCOPE) to reflect overall effectiveness.
>
> ---
>
> **Q10: Compare against full fine-tuning methods.**
>
> **A11:** BRIDGE focuses on **efficient prompt-based tuning**, not full model updates. We compare with strong graph prompt baselines (e.g., ProNoG, GraphPrompt), which align better with our design and goals.
>
> ---
>
> **Q11: Negative transfer issues for multi-domain learning.**
>
> **A11:** Negative transfer is a key challenge in multi-domain learning. BRIDGE addresses this via **expert assignment weights ($\mathbf{S}$)**, which down-weight less relevant domains. As shown in our response to **Reviewer Efax (Q5)**, $\mathbf{S}$ effectively favors helpful sources. We acknowledge BRIDGE doesn’t explicitly handle extreme cases where **all** source domains are harmful, which will be explored in future work.
>
> ---
>
> **Q12: Typos and Grammars.**
>
> **A12:**  We will carefully proofread the manuscript.

---

### Official Review · Reviewer_vj96 · 2025-03-12

**Overall Recommendation:** 4

**Summary:**

This paper introduces BRIDGE, a graph foundation model framework designed for multi-domain knowledge transfer using domain-invariant feature aligners, a MoE prompt initialization, and a theoretical generalization bound. It addresses feature heterogeneity across domains, efficient adaptation via prompt tuning, and theoretical quantification of transferability. The framework is validated on 6 diverse graph datasets with convincing empirical results.

**Claims And Evidence:**

Most of the claims are clearly supported with some minor problems.

- Multi-domain feature alignment: the authors propose a two-stage aligner that unifies domain features. Though no direct alignment metric is provided, performance gains over baselines support this claim.
- Prompt-based fine-tuning with MoE: ablations confirm MoE improves low-shot learning, and BRIDGE outperforms 15 baselines in few-shot settings.
- Generalization bound informs model design: the bound integrates Wasserstein distance and Lipschitz constraints, operationalized via a spectral regularizer that improves transfer performance. The claim is theoretically sound, though empirical validation of the bound’s direct impact would strengthen it.

**Essential References Not Discussed:**

There are no critical missing references in the paper. The authors provide a comprehensive review of related work, covering graph pre-training, domain adaptation, prompt learning, and theoretical generalization bounds. They cite relevant prior works. The citations potentially sufficiently contextualize BRIDGE’s contributions, and nearly no essential prior work appears to be missing.

**Experimental Designs Or Analyses:**

-BRIDGE achieves SOTA performance on all tasks: outperforms all 15 baselines in one-shot and five-shot classification. Cross-domain generalization (e.g., social graphs) is improved over existing pre-training methods. MoE prompting performs better in extremely low-shot settings, confirming its role in knowledge transfer.
-Ablation studies validate each component: removing the feature aligner, MoE, or spectral regularizer significantly reduces accuracy. Faster convergence than standard fine-tuning.
-Hyperparameter sensitivity: performance varies with aligner strength ($\alpha$), MoE routing entropy ($\beta$), and spectral regularization ($\gamma$). Some tuning is needed, but trends remain consistent across datasets.

**Methods And Evaluation Criteria:**

Technical contributions: BRIDGE is proposed by combining prompt learning, domain adaptation, and spectral regularization.
- Baselines & benchmarks: evaluated against 15 baselines across 6 datasets in three domains (academic, e-commerce, social), covering node and graph classification.
- Ablation & sensitivity studies: component-wise analysis, showing MoE and spectral regularizer are critical.

**Other Comments Or Suggestions:**

See Weakness.

**Other Strengths And Weaknesses:**

Strength:
- Combines multi-domain learning, prompt-based transfer, and theoretical regularization into a unified framework.
- Outperforms all baselines across 6 datasets and multiple domains, with robust few-shot generalization.
- First work to derive a domain adaptation bound for prompt-based graph transfer, integrating Lipschitz constraints.
- Includes full proofs, hyperparameter details, and additional experiments, enhancing transparency.


Weakness:
- Many moving parts (aligners, MoE, spectral regularizer) and multiple hyperparameters, making tuning challenging.
- Performance drops significantly when no labeled target examples are available (m=0 setting). A true zero-shot graph foundation model remains an open challenge.
- Eigen-decomposition for $R_{spec}$ could be expensive on large graphs. A runtime analysis would be helpful.

**Questions For Authors:**

During the pre-training stage, what is the difference between training all graphs together versus training them separately one by one? How do their performances compare? (Considering that training all graphs together poses a significant memory challenge.)
- Given limited labeled data, how were hyperparameters optimized? Are there default recommendations?
- Did $R_{spec}$ actually reduce domain divergence or Lipschitz constants? Can you show this empirically?

**Relation To Broader Scientific Literature:**

This paper contributes to multiple areas in graph representation learning, domain adaptation, few-shot learning, and graph foundation models. It integrates ideas from these domains into a novel framework that enhances transferability across graph datasets.

**Theoretical Claims:**

- Domain adaptation bound for graph transfer: extends Wasserstein distance-based domain adaptation theory to graph learning, where the bound depends on the Lipschitz constant of the GNN encoder.
- Spectral analysis for transferability: derives a lower bound on the Lipschitz constant using graph spectral properties. Uses this to construct $R_{spec}$, penalizing over-sensitive representations.
- Mathematical correctness: the proofs follow graph domain adaptation theory, adapting results from Mohri (2018) and Gama (2020). However, empirical verification of Lipschitz behavior reduction should be included.

---

> ### Author Rebuttal · Authors · 2025-03-26
>
> Thank you for your insightful and constructive review, and for recognizing the contributions of BRIDGE. We address your questions briefly below.
>
> **Q1: The empirical validation of the generalization bound.**
>
> **A1:** To empirically validate the effectiveness of the bound, we conducted specific ablation studies by removing the spectral regularizer (Section 5.3). Removing the spectral regularizer resulted in **a clear performance drop**: specifically, accuracy decreased from **42.18% to 41.33% (−0.85%)** for 1-shot node classification, and more significantly from **60.20% to 54.51% (−5.69%)** for 5-shot node classification on the CiteSeer dataset. This substantial performance degradation empirically demonstrates the effectiveness of our generalization bound in improving transferability. These findings explicitly confirm the bound’s role in guiding practical model design.
>
> ---
>
> **Q2: Empirical verification of Lipschitz behavior reduction should be included.**
>
> **A2:** Thank you for raising this important point. While our experiments clearly show that including the spectral regularizer improves transfer performance (Section 5.3), the original submission did not explicitly measure the Lipschitz constant's empirical evolution. Theoretically, the spectral regularizer is specifically designed to constrain the Lipschitz constant by penalizing large variations in learned representations. Given your valuable suggestion, we plan to explicitly measure and report the evolution of the empirical Lipschitz constant in future version, to more clearly demonstrate how our method reduces the representation sensitivity across domains.
>
> ---
>
> **Q3: Still requires at least one labeled example for good performance.**
>
> **A3:** Indeed, BRIDGE is specifically designed for **text-free graphs**, where zero-shot knowledge transfer based solely on structural information is inherently limited, which is a fact well-supported by prior works. Pure structural information lacks sufficient semantic cues, leading to poor zero-shot performance, as extensively confirmed in the literature.
>
> To demonstrate that BRIDGE can also effectively leverage textual semantics when available, we conducted additional experiments integrating textual information generated by an LLM as an enhancement strategy (Section 5.6, Table 3). Results clearly showed notable improvements: e.g., node classification accuracy on Reddit improved from **55.72% (structure-only)** to **57.16% (structure + LLM)** under zero-shot conditions. These experiments indicate that BRIDGE, though originally targeted for text-free scenarios, can effectively incorporate textual semantics, significantly improving zero-shot transfer performance when such information is available.
>
> We will explicitly clarify BRIDGE’s intended scope (text-free graphs) and discuss the potential benefits of textual augmentation thoroughly in the revision.
>
> ---
>
> **Q4: Spectral regularization may be expensive. Runtime analysis.**
>
> **A4:** Thanks for bringing up this important issue. We reply the same concerns for [**Reviewer Efax (Q2)**](https://openreview.net/forum?id=bjDKZ3Roax&noteId=e2XOFb3spO). To explicitly address this, we adopted efficient approximation techniques (lines 969–989) based on truncated eigenvalue decomposition and eigenvalue approximations.
>
> ---
>
> **Q5: Tuning challenging.**
>
> **A5:** Your point regarding the complexity of hyperparameter tuning due to multiple moving parts is valid. Recognizing this issue, we provided a detailed sensitivity analysis of key hyperparameters (Figure 6, Section 5.5), covering: $K$, $\alpha$, $\beta$, and $\gamma$. These clearly presented sensitivity analyses significantly simplify practical hyperparameter tuning by offering explicit recommendations and robust ranges. We also have provided the suggested hyperparameter settings in Appendix F.1 and the configuration files.
>
> ---
>
> **Q6: Difference between training all graphs together versus one by one.**
>
> **A6:** Thank you for this insightful question. In BRIDGE, all graphs from different domains were trained simultaneously in the pre-training phase, explicitly enabling cross-domain alignment and knowledge transfer. This approach encourages the model to learn shared domain-invariant features across multiple domains simultaneously, resulting in enhanced transfer performance. However, we recognize that training all graphs jointly can pose substantial memory overhead.
>
> Training graphs separately (one domain at a time) reduces memory consumption significantly but **sacrifices explicit cross-domain feature alignment**, potentially weakening transferability. While this sequential training strategy was not explicitly investigated in the original submission, we acknowledge your point and will explicitly clarify our current training strategy and discuss potential trade-offs in the revision. Detailed empirical comparisons of joint vs. sequential training strategies will be included in future work to thoroughly address this important point.

---

### Decision · Program_Chairs · 2025-05-01

**Decision:**

Accept (poster)

**Comment:**

This paper received three effective reviews, and all of them are positive. Overall, the paper is of good quality and should be accepted.